# Single-nucleotide m⁶A mapping uncovers redundant YTHDF function in planarian progenitor fate selection

Yarden Yesharim[1], Ophir Shwarzbard[1], Jenny Barboy-Smoliarenko [1], Prakash Varkey Cherian[1], Ran Shachar[2], Amrutha Palavalli [3], Hanh Thi-Kim Vu [3], Schraga Schwartz[2] & Omri Wurtzel [1,4✉]

## Abstract

Cell fate decisions require tight regulation of gene expression. In planarians, highly regenerative flatworms, the mRNA modification N⁶-methyladenosine (m⁶A) modulates progenitor production and fate. However, the mechanisms governing m⁶A deposition in the planarian transcriptome, and the role of their expanded family of YTHDF m⁶A reader proteins in orchestrating biological functions, remain unclear. Here, we generated the first single-nucleotide resolution map of m⁶A in planarians, and revealed that simple sequence rules guide m⁶A deposition, facilitating the flexible evolutionary gain and loss of these marks. Functional analyses of the five YTHDF planarian m⁶A readers revealed that while individual reader expression is dispensable, together, the planarian YTHDF proteins regulate the production of specific progenitor lineages and overall body size. Collectively, our findings uncover a robust, redundant regulatory architecture for cell fate control in planarians, characterized by multiple m⁶A sites per gene and coordinated m⁶A reader expression. This architecture is essential for proper lineage resolution and provides insights into the evolutionary dynamics of the m⁶A landscape.

**Keywords** m6A; YTHDF; Planarian; Differentiation; Stem Cells
**Subject Categories** Chromatin, Transcription & Genomics; Development; Stem Cells & Regenerative Medicine

## Introduction

Regeneration is a highly dynamic process that demands coordination of gene expression programs for new cell production and recovery of damaged tissues (Reddien, 2018; Wurtzel et al, 2015; Molina and Cebrià, 2021). In recent years, post-transcriptional modifications have emerged as important regulators of such dynamic gene expression changes (Dagan et al, 2022; Zhang et al,

2017). Among these modifications, N⁶-methyladenosine (m⁶A) is the most prevalent across the transcriptome (Dominissini et al, 2012; Patil et al, 2018), and has critical regulatory roles in diverse developmental systems (Patil et al, 2018; Reichel et al, 2019; Zaccara and Jaffrey, 2020; Meyer and Jaffrey, 2017; Lence et al, 2017; Kontur et al, 2020; Geula et al, 2015; Lasman et al, 2020a, 2020b). In planarians—highly regenerative flatworms— m⁶A is essential for the production of progenitor cells in specific lineages, such as the intestine (Dagan et al, 2022), and for suppressing the excessive production of neural-like progenitor cells (Dagan et al, 2022), echoing observations in other organisms where m⁶A modulates key developmental processes (Geula et al, 2015; Schwartz et al, 2014; Kan et al, 2017; Lence et al, 2016; Arribas-Hernández et al, 2018).

Deciphering how m⁶A influences complex developmental programs is highly challenging for several reasons (Zhao et al, 2017). First, m⁶A is pervasive. In planarians, ~7000 different transcripts are modified, displaying variable levels of m⁶A stoichiometries, and expressed across different cell types and states (Dagan et al, 2022). Second, m⁶A functions are mediated by a diverse set of reader proteins, the largest being the YTH-family proteins (YTHDF) (Patil et al, 2018; Meyer and Jaffrey, 2017; Reichel et al, 2019). Notably, planarians possess an expanded family of m⁶A readers, at least five YTHDF proteins, complicating efforts to determine their individual and combined roles (Dagan et al, 2022). Third, conventional approaches for functional analysis, such as inhibiting genes encoding core components of the methyltransferase complex (MTC) or the gene encoding the nuclear m⁶A reader YTHDC-1, result in severe phenotypes including lethality and complete loss of regenerative ability, thereby masking the roles of other m⁶A readers (Dagan et al, 2022; Cui et al, 2023).

The limited understanding of the rules governing m⁶A deposition further complicates the picture. Two major models have been proposed in vertebrate systems: one in which m⁶A is deposited pervasively on compatible sequence motifs but is excluded from regions that are inaccessible to the MTC (for example, in vertebrates regions bound by the exon junction complex; EJC) (Uzonyi et al, 2023; Yang et al, 2022; Luo et al, 2023; He et al, 2023), and another, where deposition is more finely

[1]The School of Neurobiology, Biochemistry & Biophysics, George S. Wise Faculty of Life Sciences, Tel Aviv University, Tel Aviv, Israel. [2]Department of Molecular Genetics, Weizmann Institute of Science, Rehovot, Israel. [3]European Molecular Biology Laboratory, Developmental Biology Unit, Meyerhofstraße 1, 69117 Heidelberg, Germany. [4]Sagol School of Neuroscience, Tel Aviv University, Tel Aviv, Israel. ✉E-mail: owurtzel@tauex.tau.ac.il

regulated based on organismal requirements (Liu et al, 2020a; Batista et al, 2014; Zhang et al, 2017). In planarians, our recent study has suggested that the sequence enriched at m⁶A sites is remarkably simple (GAC motif) (Dagan et al, 2022) compared to the more complex DRACH motif found in other animals (e.g., humans) (Dominissini et al, 2012). Whether an exclusion mechanism similar to that observed in vertebrates (Uzonyi et al, 2023) operates in planarians, and how this might influence the evolutionary dynamics of m⁶A site gain and loss, remains an open question. The availability of single-base resolution profiling methods based on sequencing, such as GLORI, facilitates a finer analysis of m⁶A sequence preference and its conservation (Liu et al, 2023; Shen et al, 2024).

In addition to the challenges of understanding m⁶A installation (Liu et al, 2023), the interpretation of this mark by the m⁶A readers is critical for understanding the regulation of its targets (Patil et al, 2018; Kontur et al, 2020). While vertebrates have three YTHDF paralogs (Meyer and Jaffrey, 2017) and *Drosophila* has only one (Lence et al, 2017), planarian genomes have at least five *ythdf* genes (Dagan et al, 2022). Whether these factors function in a redundant manner, as suggested by several studies in vertebrates (Zaccara and Jaffrey, 2020; Kontur et al, 2020; Lasman et al, 2020b), exert specialized, state or lineage-specific regulatory roles (Liu et al, 2020b), or follow a hybrid model, where some YTHDFs are redundant and others are specialized, as established in *Arabidopsis* (Arribas-Hernández et al, 2018, 2020, 2021b; Flores-Téllez et al, 2023), remains unknown. Work in animal systems provides evidence supporting both the redundancy and specialized function models: some studies have demonstrated that individual YTHDF proteins target distinct sets of m⁶A-modified transcripts (Anders et al, 2018; Han et al, 2019; Hesser et al, 2018; Paris et al, 2019; Shi et al, 2018), while others have found extensive overlap in their functions (Zaccara and Jaffrey, 2020; Kontur et al, 2020; Lasman et al, 2020b). Resolving this ambiguity in planarians is essential for understanding how m⁶A modifications are translated into specific cellular outcomes during new cell production.

To address these challenges, we mapped m⁶A sites in planarians at single-nucleotide resolution, identified 19,328 m⁶A sites across the planarian transcriptome, and characterized the underlying sequence rules governing their deposition. Notably, m⁶A sites appear to be installed independently on each gene, suggesting that genes can gain or lose m⁶A modifications without compromising the functionality of existing sites. Our analysis supports a model in which the evolutionary gain and loss of m⁶A sites are subject to minimal sequence constraints, thereby allowing flexible m⁶A pattern formation across transcripts. Furthermore, by examining the expression patterns and functional contributions of the expanded family of planarian YTHDF proteins, we found that their expression largely overlaps. Moreover, only the simultaneous suppression of multiple YTHDF-encoding genes produced striking phenotypes—animals displaying reduced body size and having a diminished pool of parenchymal progenitors and *cathepsin⁺* cell types. These findings suggest that planarian YTHDFs act, to a large extent, redundantly to promote the production of specific lineages, thereby ensuring proper tissue homeostasis. Collectively, our work advances the understanding of the mechanisms governing m⁶A deposition and recognition, and also provides new insights into the evolutionary dynamics of m⁶A regulation in this regenerative organism and raises new mechanistic questions regarding m⁶A deposition and readout.

# Results

## Single-nucleotide mapping of planarian m⁶A sites using GLORI

m⁶A is abundant across the planarian transcriptome (Dagan et al, 2022), yet previous profiling efforts lacked the resolution required to elucidate the principles governing its deposition. To address this gap, we used GLORI (Shen et al, 2024; Liu et al, 2023), a method that identifies m⁶A sites at a single-nucleotide resolution. RNA was analyzed from control samples, having normal m⁶A levels, and *kiaa1429* (RNAi) animals, in which m⁶A levels are reduced due to inhibited MTC activity (Dagan et al, 2022). GLORI selectively converts adenosines to inosines, but not m⁶A (Liu et al, 2023), enabling quantification of the m⁶A-to-A ratio across the transcriptome. Following GLORI conversion, we prepared and sequenced cDNA libraries, and mapped them to the planarian genome (Rozanski et al, 2019) ("Methods"). For each adenosine in the transcriptome, we calculated an m⁶A score by dividing the number of reads having A mapped to the position by the number of reads having either A or G (the sequencing product of inosine) mapped to the position (Fig. 1A,B; Datasets EV1 and EV2).

We identified 19,328 m⁶A sites across 4718 genes (Fig. 1A,B), with m⁶A scores exceeding 10% in any sample (Fig. EV1A; Dataset EV1; Control and *kiaa1429* (RNAi), 15,309 and 11,274 m6A sites, respectively). In control samples, 5264 sites (27.2%) displayed a median m⁶A score greater than 0.5, while only 1818 sites (9.4%) met this criterion in the *kiaa1429* (RNAi) samples (Dataset EV1; Fig. EV1B). This reduction in the number of detectable m⁶A sites demonstrated the impact of inhibiting this MTC component on m⁶A levels (Fig. EV1B). Further examination of m⁶A sites in genes expressed in all samples revealed median m⁶A scores of 0.88 and 0.45, in control and *kiaa1429* (RNAi) samples, respectively (Fig. EV1C,D; $n = 2284$; Site read coverage >10; "Methods"). Detection of m⁶A sites using GLORI required sufficient gene expression. To evaluate how expression levels influenced m⁶A site detection, we utilized sequencing libraries from the same RNA used for GLORI, which we left untreated. Normalized gene expression values were calculated from the untreated libraries ("Methods") and assigned to expression percentiles (1–100; Fig. EV1E). Using the GLORI libraries, we analyzed the number of m⁶A sites detected within genes assigned to each expression percentile (Figs. 1C and EV1F). The detection of m⁶A sites was strongly correlated with gene expression levels. For instance, over 50% of the genes were not expressed at all, and therefore lacked detectable m⁶A sites (Fig. EV1E,F). Furthermore, only 2.7% of the genes in the 70th percentile had m⁶A sites detectable across biological replicates (Figs. 1C and EV1E; Dataset EV1). Detection increased with expression level, with 90% of the genes in the top expression percentile containing detectable m⁶A sites (Fig. 1C). This could either be a characteristic of the most highly expressed genes or, alternatively, suggest that m⁶A modifications are extremely abundant but remain undetectable in transcripts with insufficient expression levels during m⁶A profiling assays.

m⁶A sites were preferentially found near the 3' end of transcripts (Fig. 1D), a pattern that persisted in both multi-exon and single-exon genes (Figs. 1D and EV1G; Dataset EV1). Moreover, m⁶A sites located towards the 3' end had a higher score compared to other

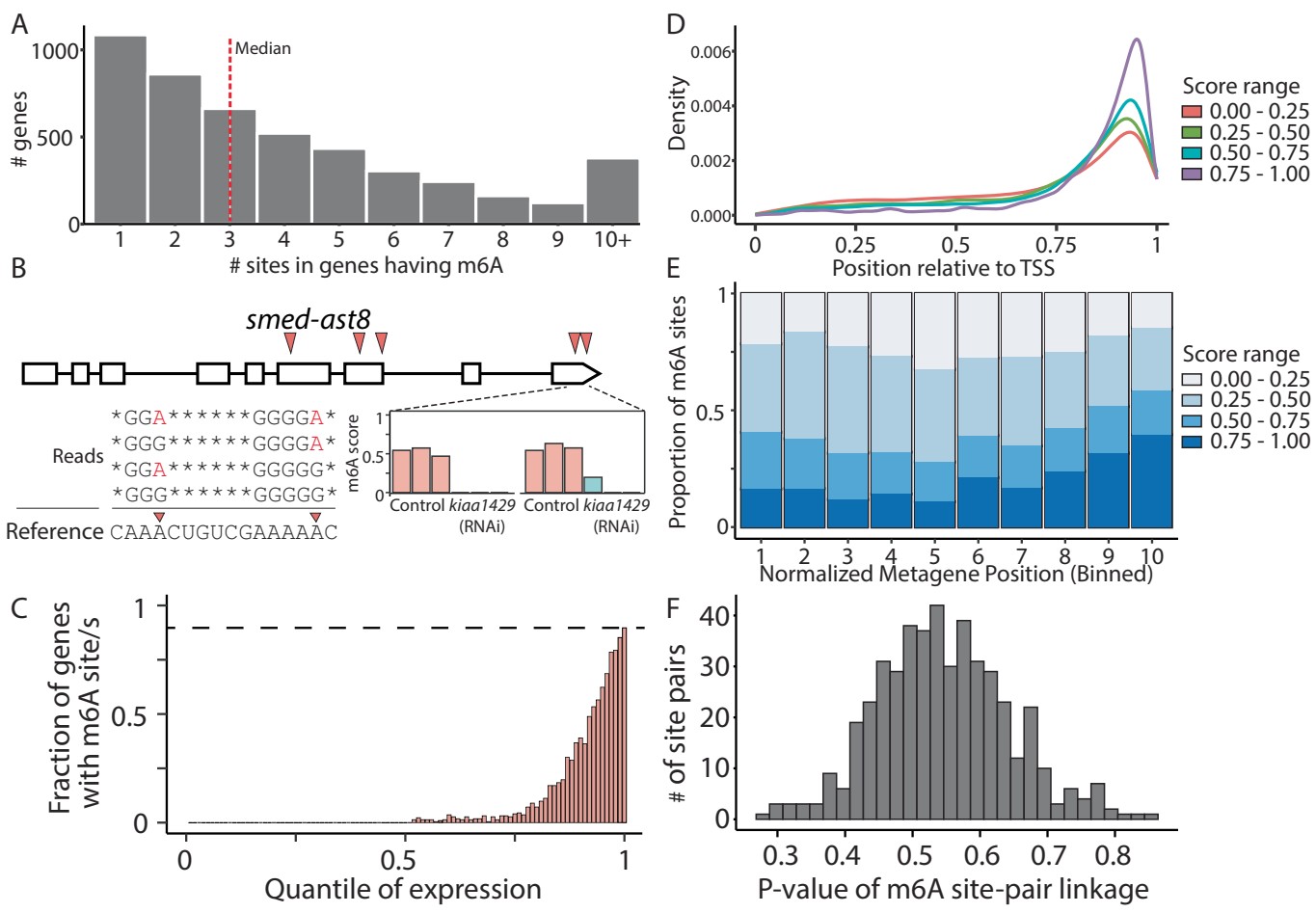

**Figure 1. Characterization of planarian m6A distribution using GLORI.**

(A) The number of detected m⁶A sites per gene is shown ("Methods"). The single nucleotide resolution m⁶A mapping facilitated detection of multiple sites per gene. (B) Mapping of m⁶A sites across the gene *ast8* identified five m⁶A sites (top; red arrowheads), including two sites that were closely adjacent (12 nt). Sequencing reads spanning both the adjacent sites (bottom-left) show how m⁶A sites were detected: Reads containing adenosines (red letters) indicate that a methyl group protected the nucleotide from the GLORI treatment. For each detected site, a score was calculated (bottom-right). Shown are the scores in three biological replicates in control animals (red bars) and in animals depleted of m⁶A (*kiaa1429* (RNAi)), due to suppression of the MTC (Dagan et al, 2022). (C) Shown is the fraction of genes having m⁶A site as a function of the expression percentile ("Methods"). (D) Meta-gene analysis of m⁶A localization showed a strong 3'-end bias of m⁶A sites regardless of the m⁶A score. (E) Distribution of m⁶A scores at different regions across the length of the transcript (bins 1 to 10 indicate relative positions from the 5' to 3' ends, respectively). Sites near the 3' end were modified more frequently compared to other m⁶A sites across the transcript. (F) Distribution of *P* values calculated for assessing linkage in m⁶A installation at nearby sites. The distribution of *P* values supported the interpretation that there was no association in installation of m⁶A in nearby sites, and that m⁶A was installed independently at every site.

m⁶A sites (Fig. 1E; Student's two-tailed *t* test $P = 7.87e−54$; "Methods"). m⁶A sites were even detected in non-polyadenylated, single-exon histone transcripts, indicating that this preference was not necessarily dependent on polyadenylation-associated factors, polyadenylation sequence signals, or the splicing machinery (Fig. EV1G,H; Dataset EV1).

Single-nucleotide resolution detection of m⁶A revealed that individual genes often contained multiple, closely spaced m⁶A sites (Fig. 1A,B). The median distance between the highest scoring m⁶A site in a gene and the nearest secondary site was 18 nucleotides (Fig. EV1I,J; Dataset EV1). This m⁶A site proximity could be a consequence of the MTC activity on several nearby positions on the same molecule (i.e., single-molecule linkage, e.g., via coupled deposition), or the independent activity of the MTC on different molecules (i.e., regional linkage, e.g., guided by *cis* elements). We

examined sequencing reads spanning two adjacent m⁶A sites, and determined whether the modification of one site influenced the likelihood of methylation at the other site (Figs. 1B,F and EV1K). We selected m⁶A site pairs for this analysis that were separated by up to 40 nt, and exhibited methylation levels between 0.4 and 0.7 ($n = 479$). Our results indicated no evidence of linkage between such adjacent sites at the single-molecule resolution (Figs. 1F and EV1K; "Methods"), suggesting that methylation at one site did not alter the probability of methylation at a neighboring site on the same molecule. These results indicated that methylation of adjacent sites was a consequence of independent events of MTC activity, and not an outcome of processive MTC activity at nearby sites. Interestingly, despite lack of linkage in methylation of adjacent sites, we found a moderate correlation (Pearson $r = 0.49$, $P$ value $< 2.2e−16$; "Methods") between the m⁶A scores of adjacent

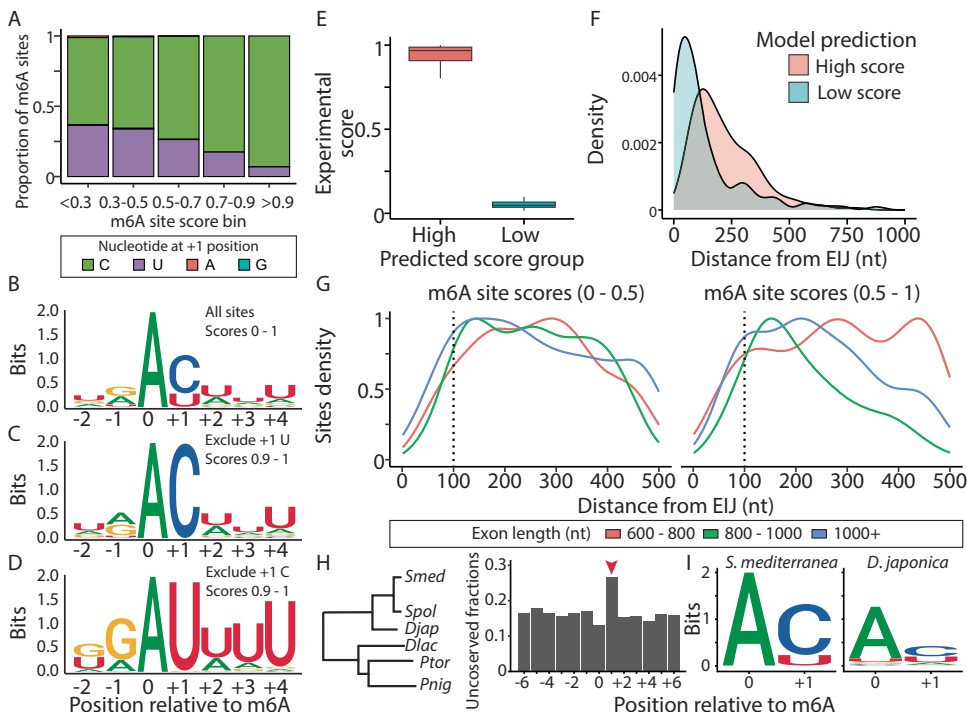

**Figure 2.  Sequence determinants of planarian m⁶A sites.**

(A) Examination of the nucleotide at the +1 position relative to the m⁶A site shows that regardless of the m⁶A score, the nucleotide is either C or U. Moreover, frequently modified sites are characterized by having C at the +1 position. (B–D) Consensus sequences at the m⁶A sites show that. Sites having C at the +1 position are not characterized by additional sequence characteristics (C). By contrast, strong m⁶A sites having U at the +1 position are characterized by additional sequence characteristics, with strong preference for a U at the +4 position (D). (E) Experimentally determined m⁶A scores, which have similar sequence properties, according to a gradient boost model analysis, indicating that sequence independent factors have a major role in determining the likelihood of m⁶A installation at a site. (F) The distance from the EIJ distinguished between similar sequences that differ in their experimentally determined m⁶A score. (G) Assessment of distance of m⁶A site relative to the nearest EIJ indicates that m⁶A sites are less prevalent near the EIJ, regardless of the exon length, and score of the m⁶A site (high and low m⁶A site scores, left and right, respectively). Dotted line indicates 100 nt. (H) Analysis of sequence variation in regions homologous to the detected m⁶A sites in *S. mediterranea* reveals a lack of conservation at the critical position (+1) relative to the m⁶A site (red arrow). A phylogenetic tree (left) depicting the species included in this analysis is shown (left; *Smed*: *S. mediterranea*; *Spol*: *Schmidtea polychroa*; *Djap*: *Dugesia japonica*; *Dlac*: *Dendrocoelum lacteum*; *Ptor*: *Planaria torva*; *Pnig*: *polycelis nigra*) adapted from PlanMine (Rozanski et al, 2019). The higher substitution rate at position +1 is attributed to the increased mutation rate observed in high %GC regions (see Fig. EV4B). (I) Pairwise sequence comparison of m⁶A sites detected in *S. mediterranea* (left) with homologous sequences in *D. japonica* (right) indicates a general lack of sequence conservation of the m⁶A and +1 sites, suggesting divergence in m⁶A site deposition between planarian species.

sites, meaning that if one site had a high m⁶A score, a nearby site was more likely to have a high score as well (Fig. EV1L,M). Altogether, this suggested that local characteristics of the transcript contributed to the level of methylation of nearby sites, in line with recent discoveries (Leger et al, 2021).

## Analysis of sequence preferences associated with m⁶A installation

We next examined the sequences surrounding m⁶A sites and found that even sequences deviating from the canonical installation motif (i.e., DRACH (Dominissini et al, 2012)) can serve as excellent MTC targets, provided they conform to specific rules (Fig. 2A–D). Essentially all m⁶A sites were followed by either a cytosine or uracil, with strong depletion of A and G (Fig. 2A–D). Stratifying m⁶A sites by score, we observed distinct sequence patterns in highly versus lowly methylated sites. Among high-scoring m⁶A sites (score >0.9), 93% (*n* = 879/945) had a cytosine immediately following the m⁶A (Fig. 2C). High-scoring sites lacking a +1 cytosine were characterized by a preceding guanosine and a stretch of uracils following the

m⁶A site, with a +4 uracil present in nearly all sequences (Fig. 2D). This suggested that a +4 U supported efficient m⁶A installation (Fig. 2D). Indeed, low-scoring m⁶A sites with U at the +1 position exhibited a lower frequency of uracil at the +4 position (Fig. EV2A,B; Dataset EV1).

m⁶A sites having either a cytosine or a uracil at the +1 position showed the same preferential localization near the 3' end of transcripts (Fig. EV2C), indicating that their installation in the transcript was governed by similar principles. These findings suggest simple sequence rules for m⁶A methylation: (i) cytosine at (+1) following the adenine promotes high methylation potential; (ii) uracil at (+1) can only support a low methylation frequency when not part of a uracil stretch; (iii) adenine or guanine at (+1) was largely incompatible with methylation. We examined this observation by analyzing a rare event where alleles had sequence variations in the m⁶A motif: In one allele, there was a C following the m⁶A site, and the other allele had a G (Fig. EV2D). Both alleles were similarly expressed in the input (untreated) libraries (Fig. EV2E). However, GLORI libraries revealed that transcripts with the AC allele were predominantly methylated (>90%), while

those with the AG allele were not (Fig. EV2F). This supported the observation that guanine at the +1 position following adenine was essentially incompatible with m⁶A installation.

Our observation that the planarian MTC efficiently modified GAU motifs is reminiscent of findings in *Arabidopsis* (Arribas-Hernández et al, 2021a; Wang et al, 2024). Yet the molecular basis for the extended sequence preference was unclear. We therefore compared METTL3 and METTL14 sequences from representative organisms and focused on two regions implicated in substrate specificity. First, we examined residues linked to a shift in preference toward GGAU over GGAC, which is observed in cancer-associated substitutions: METTL14 R298P and METTL3 R471H (Zhang et al, 2024; Qi et al, 2024). These residues were invariant across organisms and therefore variation at these sites could not explain the altered sequence preference in planarians (Fig. EV3A,B). Second, we compared the METTL3 ZnF1/2 RNA-recognition module (Fig. EV3C). The basic RNA-contact surface was broadly conserved, but a notable difference was found at the position corresponding to human R301 (conserved in *Arabidopsis*; Fig. EV3C), which was substituted by a lysine in planarians. Given that −1G is not required in planarians, especially when +1C is present, this substitution might contribute to broader −1 tolerance. However, no single decisive residue emerged from the alignments.

Our observations suggested that characteristics prevalent in high-scoring m⁶A sites (e.g., cytosine at +1 and uracil at +4) were necessary but not sufficient for high methylation potential, indicating that additional regulatory mechanisms govern m⁶A installation. To explore this, we used gradient boosting regression to model sequence features associated with m⁶A sites, and their contributions to m⁶A scores (Fig. EV4A,B; "Methods"). We then extracted m⁶A sites that were predicted to have the highest m⁶A scores by the model (Fig. 2E; Top 1%; "Methods"), and compared sites belonging to this set, which had a high m⁶A score (>0.8; $n = 134$), with sites having low m⁶A score (<0.1; $n = 120$). We found that sites having high m⁶A score prediction but that had low experimental m⁶A site scores were located near the exon-intron junction (EIJ; Fig. 2F; median distance = 86.5 nt). Sites having both high predicted and experimental m⁶A scores were positioned further away from the EIJ (Fig. 2F; median distance = 194 nt). These results were consistent with recent findings that EIJs are strong predictors of reduced methylation frequency (Uzonyi et al, 2023). A strong depletion of m⁶A sites near EIJs further corroborated this observation (Fig. 2G).

We observed that m⁶A installation follows simple sequence rules (e.g., +1 C), and requires sufficient distance from the EIJ. This suggested that m⁶A sites may be readily gained or lost during evolution by minimal sequence changes. By contrast, if a specific m⁶A site was critical for function, the underlying sequence would be expected to be highly conserved. To evaluate these alternatives, we identified potential homologous sequences of m⁶A sites in five planarian transcriptomes (Rozanski et al, 2019). We compared nucleotide conservation near the m⁶A sites, and found that key positions in the m⁶A motif (e.g., +1) were not appreciably conserved to a greater extent than adjacent nucleotides (Figs. 2H, I and EV4B; Dataset EV2; "Methods"). Interestingly, the +1 position appeared significantly less conserved compared to other nearby positions (Adjusted $P$ value $< 2.12 \times 10^{-16}$; "Methods"), supporting the hypothesis that m⁶A installation at a specific site was not strongly evolutionarily conserved, and hinting at potential

for loss and gain. We tested whether the reduced conservation of the +1 position indicated that there is a negative selection against m⁶A sites, or alternatively a higher substitution rate of C and G nucleotides in this low % GC genome (30–35%) (Grohme et al, 2018). Indeed, the mutation rate of C and G nucleotides near the m⁶A site was significantly higher in every position tested adjacent to the m⁶A site, in pairwise sequence comparison between *S. mediterranea* and *Dugesia japonica* (Adjusted $P$ value $< 1^{-10}$; Fig. EV4B; "Methods"). This analysis indicated that if selective forces act on the sequences of m⁶A sites, they are minor compared to other forces shaping genome sequence identity (e.g., bias toward certain GC content).

Our recent functional analysis of the planarian MTC and *ythdc-1* (predicted to encode a nuclear YTH family member) has revealed that they are required for the production of intestinal progenitors, and for repressing the emergence of cells expressing neural progenitor-associated genes (Dagan et al, 2022). To investigate the cell type specificity of m⁶A methylation, we compared GLORI methylation profiles with the planarian single-cell gene expression atlas (Fincher et al, 2018). Transcripts with high-scoring m⁶A sites were detected in all major cell types (Dataset EV1), suggesting that m⁶A likely has additional roles in cell types not previously examined (Dagan et al, 2022). This finding suggested that elucidation of m⁶A function cannot focus exclusively on analysis of the MTC, as the lethal intestine phenotype appearing following its inhibition (Dagan et al, 2022) could mask m⁶A functions in other cell types. Instead, investigating m⁶A readers, their regulation, and expression in different tissues could offer more targeted insights into the regulatory roles of m⁶A in planarians.

## Planarians have an expanded repertoire of YTHDF proteins

We identified planarian genes that encode potential m⁶A readers by searching for sequences that putatively encode the conserved YTH domain by protein domain analysis (Fig. 3A; "Methods"). This analysis identified five genes encoding planarian YTHDF proteins, in contrast to *Drosophila melanogaster*, which has a single YTHDF-encoding gene, and vertebrates, which encode three (Dominissini et al, 2012; Lence et al, 2016). This finding suggests that there was an expansion of the YTHDF family in planarians compared to other animals (Figs. 3A,B and EV5A,B; Dataset EV3). The putative planarian YTHDFs varied in length (366–665 amino acids; AA) compared to human YTHDFs (559–614 AA), indicating a significant divergence between the planarian and human sequences. We produced a phylogenetic tree of the YTH domains in planarians, humans, and additional representative species to assess their conservation (Figs. 3A,B and EV5A,B; "Methods"). The conservation of key residues associated with m⁶A recognition within the polypeptide suggested that these proteins may interact with similar substrates (Figs. 3A and EV5A). However, the phylogenetic analysis strongly indicated that vertebrate YTHDFs underwent independent duplication events, with no single-copy orthology observed between planarian and vertebrate YTHDFs (Figs. 3B and EV5B; "Methods"). We named the planarian YTHDF-encoding genes *ythdfA–E* to reflect their divergent evolutionary history, which may also contribute to functional differences between planarian and human YTHDFs (Figs. 3A,B and EV5A,B).

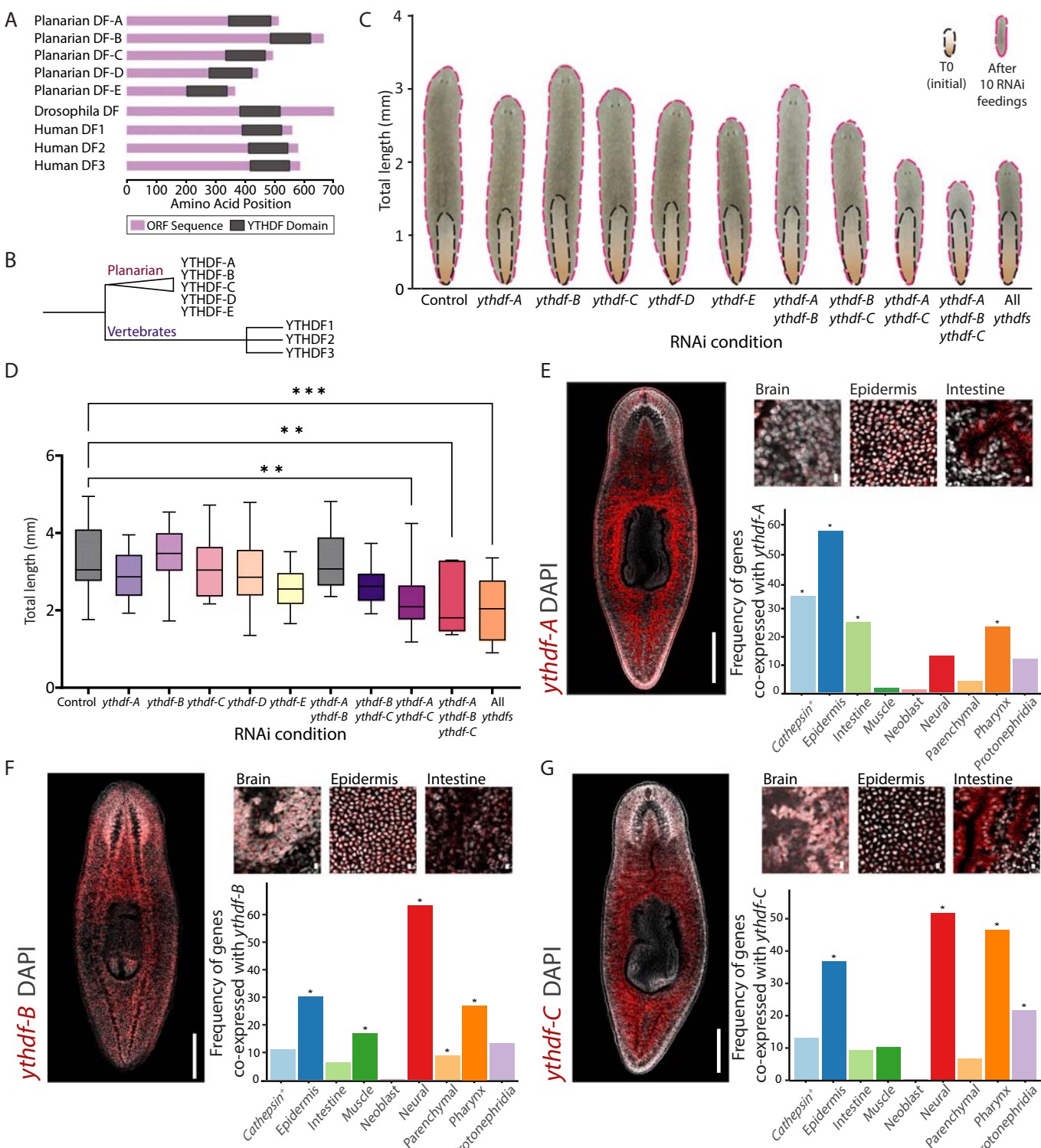

## Redundant roles of *ythdf* genes in regulating planarian body size

The roles of planarian YTHDFs have not been elucidated (Dagan et al, 2022). We analyzed their functions by inhibiting their expression using RNA interference (RNAi) (Fig. 3C,D; "Methods").

Animals were treated with double-stranded RNA (dsRNA) 10 times, and monitored for phenotypes in homeostasis and in regeneration (Figs. 3C,D and EV6B; "Methods"). We did not detect morphological or behavioral phenotypes, in either homeostasis or regeneration (Figs. 3C,D and EV6B; "Methods"). Importantly, in plants, it is well-established that many, but not all, YTHDF paralogs

**Figure 3. Planarians have an expanded family of YTHDF proteins.**

(A) A schematic representation of the YTHDF proteins across different species. The purple bars represent the protein sequence length, and the gray boxes represent the conserved YTH domain. The amino acid positions on the X axis indicate the relative length and placement of the YTH domain within the protein sequence. While the YTH domain is highly conserved across species, the overall protein size exhibits notable variation between vertebrates and invertebrates. (B) A schematic representation of the phylogenetic analysis of YTHDF proteins (see complete analysis in Fig. EV5B). The analysis indicated that planarian YTHDF proteins form a distinct clade from vertebrates, reflecting independent duplication events. Vertebrate YTHDF proteins cluster separately with no evidence of orthology between planarian and vertebrate YTHDFs. (C) Inhibition of multiple *ythdf* genes results in a size reduction phenotype. Animal size was measured at the beginning of the experiment (black) and one week following the last feeding (pink). The average size of the animals is shown before and after the experiment (Animals per group $n > 10$). (D) Animal size measurements following 10 RNAi feedings show a significant reduction in animal size when inhibiting at least two *ythdf* genes. Significance was calculated by using one-way ANOVA followed by Dunnett's test (P values: *ythdf-A* & *ythdf-C* $P = 0.0077$, *ythdf-A* & *ythdf-B* & *ythdf-C* $P = 0.0013$, all *ythdf*s $P = 0.0002$; Animals per group $n > 10$; Boxes represent the IQR, whiskers represent min to max, and central band represents the median). (E–G) Analysis of the expression of *ythdf-A* (D), *ythdf-B* (E), *ythdf-C* (F) by FISH and scRNAseq. Left panels show FISH analysis on the entire organism (scale = 500 µm). Right-top panels show a higher magnification at particular tissues (scale = 10 µm). Right-bottom panels show the frequency of *ythdf* gene co-expression with cell-type-specific genes (Dataset EV4), clustered by lineage origin in the planarian single cell gene expression atlas (Fincher et al, 2018). Asterisk indicates significance relative to the number of genes defining each lineage (Hypergeometric test, P value < 0.05; "Methods"). Source data are available online for this figure.

exhibit redundant activity (Arribas-Hernández et al, 2018, 2020, 2021a, 2021b; Flores-Téllez et al, 2023). Moreover, recent studies of vertebrate YTHDF activity have demonstrated that they function redundantly and have similar biochemical targets (Kontur et al, 2020; Zaccara and Jaffrey, 2020; Lasman et al, 2020b). Despite the divergent evolutionary history of planarian and vertebrate *ythdf*s, redundancy of YTHDFs might have evolved independently in planarians. We therefore tested whether planarian YTHDFs have redundant functions by co-inhibiting their expression.

We co-inhibited all five *ythdf* genes and observed a striking reduction in animal size compared to controls (Fig. 3C,D; one-way ANOVA followed by Dunnett's test $P = 0.003$). In order to pinpoint which *ythdf* genes were driving this phenotype, we examined their expression in a published single-cell RNAseq (scRNAseq) dataset (Fig. EV6A) (King et al, 2024). This analysis revealed that *ythdf-A*, *ythdf-B*, and *ythdf-C* were broadly expressed, whereas *ythdf-D* and *ythdf-E* showed minimal expression (Fig. EV6A), suggesting a limited contribution to the phenotype (Figs. 3C,D and EV6B). To test this hypothesis, we co-inhibited *ythdf-A*, *ythdf-B*, and *ythdf-C* in pairs or together (Figs. 3C,D and EV6B), and compared the worm sizes to the inhibition of all five *ythdf* readers. The phenotype from the triple gene inhibition closely replicated the inhibition of all five genes and was stronger than that of any pair of *ythdf* genes (Figs. 3C,D and EV6B). qPCR confirmed over 80% inhibition in expression for each targeted gene (Fig. EV6C). Interestingly, despite this strong homeostatic phenotype, the animals retained their ability to regenerate (Fig. EV6B), in contrast to the consequence of inhibition of the MTC or of the nuclear m⁶A reader, *ythdc-1* (Dagan et al, 2022).

Suppression of the MTC causes size reduction combined with severe intestine and food ingestion defects (Dagan et al, 2022). Co-inhibition of *ythdf*s indeed resulted in size reduction, but it did not affect food uptake or animal fission. Therefore, such effects might be mediated by different processes regulated by m⁶A and its factors (e.g., *ythdc-1* (Dagan et al, 2022)). To distinguish whether the size reduction after co-*ythdf* inhibition reflected impaired growth upon feeding or accelerated tissue turnover, we microinjected dsRNA targeting the three *ythdf*s into unfed animals over the course of three weeks ("Methods"). Co-*ythdf* (RNAi) animals did not differ in size from controls (Fig. EV6D), despite efficient suppression of the *ythdf* targets (Fig. EV6E). These results indicated that the size reduction was not due to increased tissue turnover but rather

reduced growth in response to feeding, even though food uptake itself remained intact.

To assess what cell types might be affected directly by the three *ythdf*s, which might contribute to the size reduction phenotype, we analyzed their expression by fluorescence in situ hybridization (Fig. 3E–G; FISH; "Methods"). All three *ythdf*s were expressed in multiple tissues throughout the body showing both overlapping and distinct expression patterns (Fig. 3E–G). For example, the three *ythdf*s were similarly expressed in the epidermis (Fig. 3E–G). Additionally, each *ythdf* had a major domain of expression in a specific organ system: *ythdf-A* in the intestine, *ythdf-B* in the brain, and *ythdf-C* in the lining of the intestine and brain (Fig. 3E–G). Re-analysis of scRNAseq data from the planarian cell type-specific gene expression atlas (Fincher et al, 2018) verified that *ythdf*s were expressed across many cell types (Fig. 3E–G). Moreover, the scRNAseq analysis showed that the *ythdf* genes were predominantly expressed in differentiated cells (Fig. 3E–G), in contrast to the MTC components and *ythdc-1*, which are overexpressed in neoblasts (Dagan et al, 2022). This suggested that *ythdf*s function at later stages of cellular differentiation and maintenance. The co-expression of *ythdf*s was consistent with the hypothesis that the *ythdf*s may be functionally redundant.

## YTH-encoding genes are co-expressed in cells but exhibit distinct tissue enrichment

Our FISH and scRNAseq analyses showed that *ythdf-A*, *ythdf-B*, and *ythdf-C* were broadly expressed in multiple tissues (Fig. 3E–G), suggesting that they might be co-expressed in the same cells. We performed multicolor FISH with probe pair combinations to detect *ythdf-A*, *ythdf-B*, and *ythdf-C*, and assess their co-expression (Fig. 4A; "Methods"). First, we examined tissues that showed detectable expression of *ythdf* genes but that did not exhibit strong specificity for expression of any single *ythdf*, such as the epidermis (Fig. 3E–G). We observed broad co-expression with most cells showing expression of at least two *ythdf*s (Fig. 4A). For example, over 60% of the epidermis cells expressed at least two *ythdf*s (Figs. 4A,B and EV7). Examination of tissues that were particularly enriched with expression of one of the *ythdf*s (e.g., *ythdf-A* in the intestine; Fig. 3E) indicated that in addition to the dominant expression of the enriched *ythdf*, other *ythdf*s were also expressed (Fig. 4A). All *ythdf*s exhibited a speckle-like expression pattern,

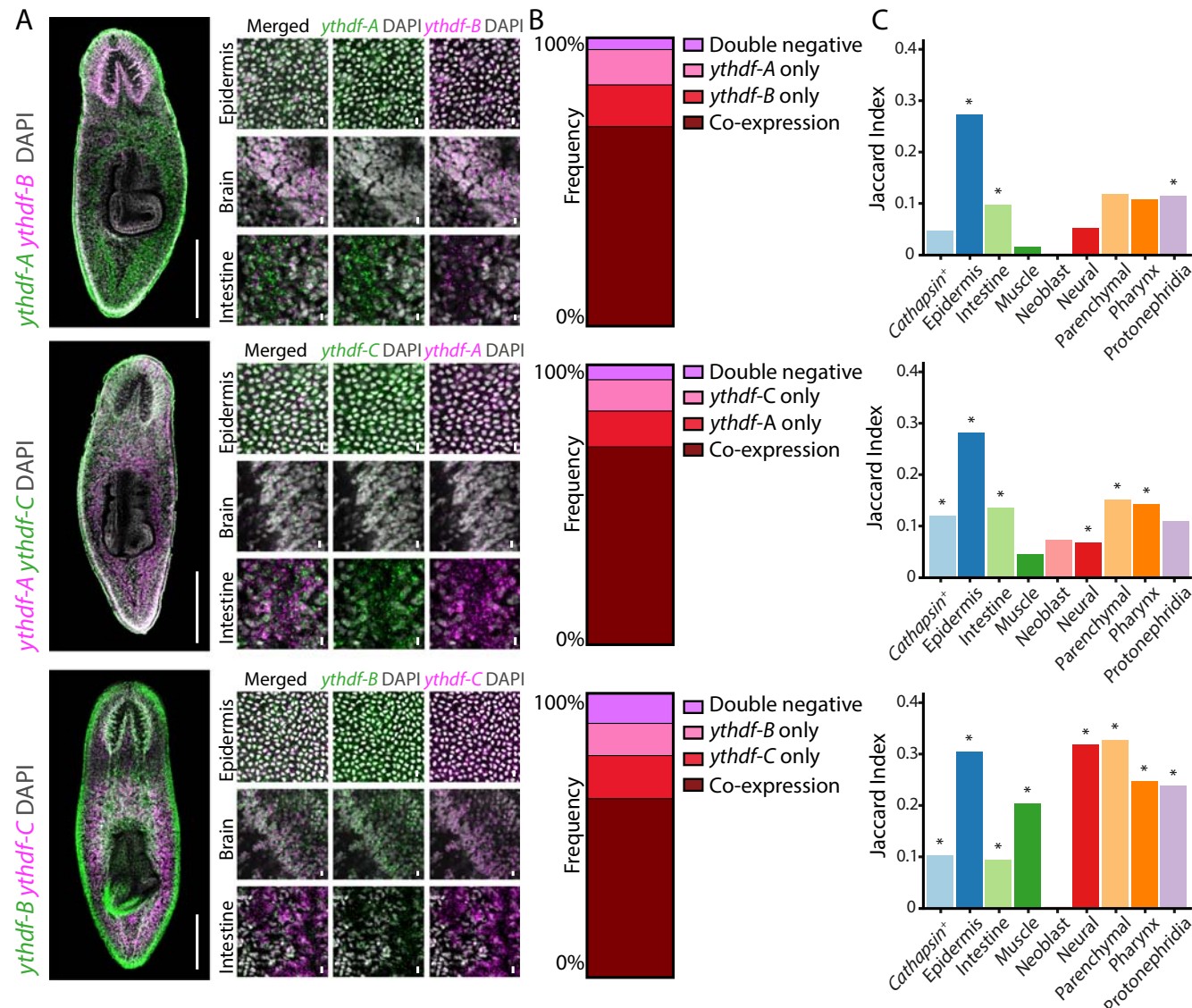

**Figure 4. Co-expression and tissue enrichment of YTH-encoding genes.**

(A) Multicolor FISH of different combinations of *ythdf* genes showing partial co-expression. Expression of each gene is shown either in magenta or green across the gene combination images (scale = 500 μm) (left). Higher magnification images of main clusters showing the differences in expression pattern of the different *ythdf* genes. (scale = 10 μm) (right). (B) Proportional distribution of the expression of the *ythdf* genes in epidermal cells. Multicolor FISH analysis was performed to categorize cells into four groups: double-negative cells (light pink), cells expressing only one *ythdf* (pink and red) and cells co-expressing both *ythdf*s (dark red). Data were collected from 10 distinct epidermal regions from the top of the pharynx to the brain, normalized to the area of each region, and averaged. The proportion of each category was plotted. (C) scRNAseq analysis showing Jaccard index (i.e., overlap of co-expressed genes in tissue/union of genes; "Methods") for each lineage (Fincher et al, 2018) for different pair combinations of *ythdf-A*, *ythdf-B*, and *ythdf-C*. Empirical *P* value of Jaccard Index was determined using 1 M permutations. *$P$ value $< 1 \times 10^{-4}$. Source data are available online for this figure.

with the higher expression (i.e., tissue-enriched *ythdf*) observed as a higher density of speckles (Fig. 4A).

To identify specific lineages expressing multiple *ythdf*s, we analyzed scRNAseq data from the planarian gene expression atlas (Fincher et al, 2018). We examined the co-expression patterns of each *ythdf* with markers associated with specific cell types ("Methods"). Subsequently, we quantified the similarity between the sets of genes co-expressed with different *ythdf* genes by calculating the Jaccard index for pairs of sets (Fig. 4C; "Methods"). To determine the statistical significance of the Jaccard index, we

performed a permutation analysis with $10^6$ iterations to obtain empirical *P* values of the overlap between gene sets ("Methods"). This allowed us to assess the overlap in co-expression profiles between *ythdf*s across cell types systematically. We found broad co-expression of *ythdf-B* and *ythdf-C* with genes representing multiple lineages, including the muscle, neural, parenchymal, pharynx, and protonephridia lineages (Fig. 4C). The overlap between *ythdf-A* and *ythdf-B* or *ythdf-C* was primarily found for the epidermal lineage, with a lower Jaccard index detected for other cell types (Fig. 4C). These findings suggest that YTHDF proteins might function

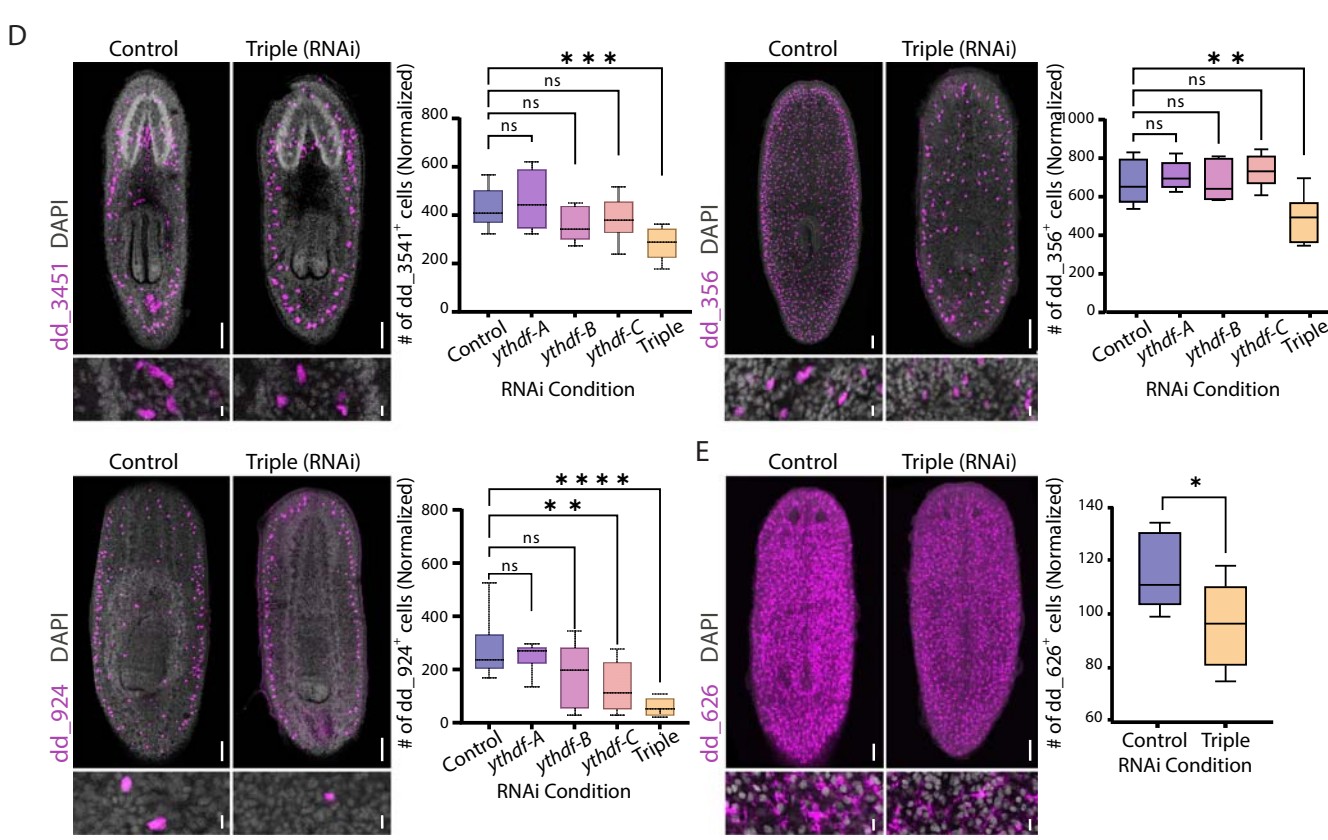

**Figure 5. Inhibition of *ythdf* genes resulted in a decrease in parenchymal cells.**

(A) UpSet plot showing the number of differentially expressed genes across different *ythdf* RNAi conditions ( | log$_2$ (fold change)| >0.5; Adjusted *P* value < 1 × 10$^{-5}$). Each bar represents a unique combination of genes shared among the specified conditions. Bars indicate the number of genes present in the intersections of selected conditions. (B) Heatmap of the top 30 downregulated genes following co-inhibition of *ythdf*s, compared to individual *ythdf* inhibition and control (FDR < 1 × 10$^{-5}$). Displayed are z-scores ranging from −2 to 2. Rows represent genes, and columns represent samples. Blue and red indicate low to high gene expression, respectively. Columns represent biological replicates. The rightmost column denotes whether the gene is highly expressed in the parenchymal lineage (Fincher et al, 2018). (C) UMAP from co-*ythdf* (RNAi) and control samples showing maps separated by treatment. All different cell types are represented in both conditions ("Methods"). (D) FISH analysis following inhibition of *ythdf* genes reveals changes in different parenchymal cell types, as described in the planarian cell type atlas (Fincher et al, 2018). Representative FISH images are shown for animals subjected to *ythdf* co-suppression and controls. Cell counts were normalized to animal size ("Methods"), and compared to control animals (one-way ANOVA; *P* values: dd_3451 triple (RNAi) *P* = 0.0008, dd_356 triple (RNAi) *P* = 0.0028, dd_924 *ythdf-C* (RNAi) *P* = 0.0049, dd_924 triple (RNAi) *P* = 1.59 × 10$^{-5}$; group size *n* > 7; Boxes represent the IQR, whiskers represent min to max, and central band represents the median). Scale bar = 100 μm. (E) FISH analysis detecting *cathepsin*$^+$ cells expressing the marker dd_626 (Fincher et al, 2018) following *ythdf* co-inhibition and control animals. A comparison of normalized cell counts in the region between the pharynx and the brain ("Methods") revealed a significant reduction in *cathepsin*$^+$ cells in co-*ythdf* (RNAi) animals (Student's two-tailed *t* test *P* value = 0.04, group size *n* > 6). Scale bar = 100 μm. Source data are available online for this figure.

redundantly in tissues where their expression overlap, while also exhibiting specialized roles in tissues where they were predominantly expressed. However, both FISH and scRNAseq indicated that overlap in expression of more than a single *ythdf* was prevalent in many tissues.

## Planarian YTHDFs co-regulate gene expression

The simultaneous suppression of the three *ythdf*s resulted in a significant reduction in animal size, whereas suppressing any individual *ythdf* did not produce a similar effect (Fig. 3C,D). Together with the observed co-expression of *ythdf* genes, this finding suggested a potential explanation: molecular redundancy. To further investigate the molecular consequences of *ythdf* inhibition, we measured gene expression by RNA sequencing (RNAseq) following the suppression of *ythdf* genes either individually or together (Fig. 5A,B; Dataset EV5; "Methods"). We observed a highly significant reduction in expression of the suppressed *ythdf* gene (or genes) in each condition, with each targeted gene exhibiting decreased expression of over 79% (Adjusted *P* value < 1 × 10$^{-132}$; Figs. 5A,B and EV8A; Dataset EV5).

Inhibiting any single *ythdf* gene did not alter the expression of other *ythdf*s or MTC-encoding genes, indicating lack of compensatory gene expression within the pathway (Figs. 5B and EV8A; Dataset EV5). Moreover, suppressing a single *ythdf* had a minor effect on gene expression, with the number of differentially expressed genes ranging from 6 to 26 (Fig. 5A,B; Dataset EV5; Adjusted *P* value < 1 × 10$^{-5}$; |log$_2$ fold-change | >0.5; "Methods"). Co-suppression resulted in approximately ~fivefold more genes significantly changing their expression, with a much stronger effect size and significance (Fig. 5A; Dataset EV5). Genes showing altered expression were not enriched for m6A sites (Fig. EV8B–D; "Methods"). Therefore, our results are more consistent with a model in which a differentially expressed gene is regulated, directly or indirectly, by multiple YTHDF proteins, although direct biochemical evidence is limited by the constraints of our model system (see "Discussion").

We initially focused on the identity of genes that were downregulated following co-suppression of the *ythdf*s (Fig. 5B). We annotated the downregulated genes using the planarian cell type atlas (Fincher et al, 2018). Analysis of the 30 most downregulated genes showed that 40% were associated with multiple parenchymal cell types (Fincher et al, 2018) (Fig. 5B;

*n* = 12/30; Adjusted *P* value < 1 × 10$^{-5}$; Dataset EV5; "Methods"), a cell lineage giving rise to multiple secretory cell types (Plass et al, 2018). In addition, we observed a reduction in the expression of genes active in phagocytes (*n* = 28/92; Dataset EV5), suggesting an effect on either intestine or *cathepsin*$^+$ cell lineages, which share similar gene expression profiles (Fincher et al, 2018) (Dataset EV5). Furthermore, we performed Gene Ontology (GO) enrichment analysis ("Methods") on all differentially expressed genes following co-*ythdf* suppression. Several biological processes were overrepresented (Fig. EV8E), including small-molecule metabolic process (GO:0044281) and metabolic process (GO:0008152), with modest enrichment for drug transport (GO:0015893) and drug transmembrane transport (GO:0006855). This enrichment pattern was consistent with perturbed core metabolism and altered transporter activity, which may relate to the observed size reduction.

To refine these observations, we performed scRNAseq on co-*ythdf* (RNAi) and control animals. We sequenced 14,854 cells (Figs. 5C and EV9) from both conditions, and following data processing (Hao et al, 2024) (Figs. 5C and EV9A,B; "Methods"), we annotated cell type identity by comparing gene expression of each cluster with published cell-type-specific markers from the planarian cell atlases (Fincher et al, 2018; Plass et al, 2018) (Fig. EV9C). We identified all major planarian cell types, including cells at different differentiation stages (e.g., neoblasts, post-mitotic progenitors, differentiated cells) (Figs. 5C and EV9C; Dataset EV6). Importantly, all cell populations were detectable in both co-*ythdf* (RNAi) and control animals (Fig. 5C), indicating that the size reduction phenotype was not a consequence of intestine cell depletion, as observed following suppression of the MTC (Dagan et al, 2022).

Using FISH for detecting the expression of downregulated genes, we tested whether the gene expression reduction resulted from a decrease in the number of cells expressing the gene, or from lower expression in a comparable number of cells (Figs. 5D,E and EV8F,G). We found a highly significant reduction in the number of multiple parenchymal cell types following *ythdf* co-inhibition (Figs. 5D and EV8F), which in most cases, was not observed following inhibition of an individual *ythdf* (Figs. 5D and EV8F). FISH quantification of *cathepsin*$^+$ and intestine cells using specific markers showed a reduction in *cathepsin*$^+$, but not in intestine cell numbers (Figs. 5E and EV8G). We note that throughout the experiment, animals appeared to uptake food normally, further indicating that the intestine was not compromised. This strongly

suggested that the reduction in phagocytic gene expression observed in RNAseq (Dataset EV5) likely resulted from depletion of *cathepsin*[+] cells and not an effect on intestinal phagocytes (Fincher et al, 2018; Forsthoefel et al, 2020; Scimone et al, 2018).

We assessed whether the gene expression changes that followed the *ythdf* co-inhibition were also observed after suppression of the planarian MTC (Dagan et al, 2022). We observed only a moderate correlation ($R^2$ range between 0.23–0.4) between gene expression changes emerging following inhibition of the *ythdf*s and MTC components (Fig. EV8H). For example, we analyzed the published gene expression following *kiaa1429* and used the same criteria for determining the identity of the downregulated genes (adjusted $P$ value $< 1 \times 10^{-5}$; $\log_2$ fold-change $< -0.5$). Only 11 genes were similarly downregulated in *kiaa1429* (RNAi) and in the combined *ythdf* suppression (Dataset EV5; Fig. EV8H). The rapid deterioration of the animal following inhibition of the MTC, which involves severe intestine damage (Dagan et al, 2022), likely masked functions mediated by these three *ythdf*s. Notably, a complementary analysis of m[6]A distribution across cell types using our GLORI data (Datasets EV1 and EV2) revealed no distinct enrichment in parenchymal or *cathepsin*[+] cells. This suggested that the depletion of these cell types resulted indirectly from m[6]A regulation inactivation rather than direct targeting of m[6]A-modified transcripts.

We next examined genes that were upregulated following the inhibition of *ythdf*s (Fig. 6A). Suppression of a single *ythdf* resulted in very few significant gene expression changes ranging from 1 to 12 (Dataset EV5; adjusted $P$ value $< 1 \times 10^{-5}$; $\log_2$ fold-change $>0.5$). In comparison, co-suppression of the *ythdf*s resulted in the upregulation of 34 genes, with 41% of the genes annotated as neural-expressed (Dataset EV5) (Fincher et al, 2018). The inhibition of MTC components, including *kiaa1429* suppression, results in the emergence of a population of cells with a distinct neural progenitor-like gene expression profile that is almost undetectable in control animals (Dagan et al, 2022). We previously named these cells *kiaa1429* (RNAi)-specific cells (Dagan et al, 2022). Genes that are uniquely expressed in this cell population were also the most highly induced following co-suppression of the *ythdf*s (Fig. 6A; Dataset EV5). Using FISH for detection of a marker gene (dd_1837) of these *kiaa1429* (RNAi)-specific cells, we observed broad expression across the animal following co-suppression of the three *ythdf*s, beyond the normal domain of this gene's expression around the pharynx (Fig. 6B,C). This highly significant overabundance in dd_1837[+] cells (Fig. 6B; $P = 0.0002$) was especially notable in the head region, in agreement with the observation that these *kiaa1429* (RNAi)-specific cells express genes associated with neural progenitors (Dagan et al, 2022). Interestingly, these highly induced genes are found in multiple adjacent repetitive copies in the planarian genome (Dagan et al, 2022).

To pinpoint the cell types expressing markers of the *kiaa1429* (RNAi)-specific cells after co-*ythdf* (RNAi), we analyzed the scRNAseq dataset, and examined the expression of the markers dd_1837 and dd_422 (Figs. 6D and EV10A,B). A striking induction was limited to clusters 23 and 24 (Fig. 6D and EV10A and B), corresponding to subsets of *mag-1*[+] parenchymal and neural cells, respectively (Fig. EV9C; Dataset EV6). Moreover, induction was also detectable in several additional clusters (Fig. EV10A,B), arguing against overproduction of a normal cell type and instead pointing to an aberrant, or normally transient or rare cell state that

may arise from altered maturation or a change in transcriptional regulation.

Examination of the GLORI data mapped to this region showed that the clustered genes contained multiple m[6]A sites (Fig. 6E; Dataset EV2), and that a significant, yet milder, effect was observed following the suppression of *ythdf-A* and *ythdf-C* (Fig. 6A; Dataset EV5). The presence of multiple m[6]A sites on these duplicated genes suggested that YTHDFs might recognized them, and potentially mediated their suppression in control animals.

## Expression of *ythdf* genes is required for normal progenitor production

The changes in different cell populations following inhibition of *ythdf* genes (Figs. 5D,E and 6B) suggested that animals were unable to produce new cells as they normally would. Failure to maintain or generate tissues in planarians is often a consequence of neoblast depletion or impaired differentiation (Reddien et al, 2005; Lin and Pearson, 2014; Zhu et al, 2015). We therefore tested whether *ythdf* expression is required for normal neoblast cell cycle regulation (Fig. 7A; "Methods"). We performed fluorescence-activated cell sorting (FACS) on Hoechst-labeled planarian cells from control and co-*ythdf* (RNAi) animals to assess shifts in cell cycle states. Specifically, we examined whether the proportion of cells in the X2 gate, comprising G0/G1 neoblasts and recently divided post-mitotic progenitors (Hayashi et al, 2006), was altered in a manner that could account for impaired differentiation (Reddien et al, 2005). FACS analysis revealed no significant changes in the distribution of cell states across gates following triple RNAi (Fig. 7A). Although differences were detectable in specific cell types by FISH (Fig. 5D,E), the overall neoblast cell cycle distribution remained largely unchanged. Similarly, cell cycle analysis on our scRNAseq data indicated that there was no alteration in the proportion of cell cycle states (Fig. EV10C,D).

We next directly assessed mitotic activity by labeling dividing cells with anti-H3P antibody and quantified H3P[+] cells in the animal (Fig. 7B; "Methods"). No significant change in H3P[+] cell number was observed after co-*ythdf* inhibition (Fig. 7B; Student's two-tailed $t$ test, $P = 0.51$), indicating that neoblast mitotic activity was not globally affected. Similarly, immunofluorescence for detecting neoblasts and their recently produced progeny with anti-SMEDWI-1 antibody revealed no change in overall neoblast number following triple RNAi (Fig. 7C; Student's two-tailed $t$ test $P = 0.995$; "Methods"). The lack of change in neoblast abundance and cell-cycle dynamics indicated that the size-reduction phenotype following co-*ythdf* (RNAi) was not due to global neoblast insufficiency. We next considered whether co-*ythdf* suppression resulted in increased cell turnover. We counted apoptotic cells by whole-mount TUNEL in control and co-*ythdf* (RNAi) animals ("Methods"), and detected no difference (Fig. EV11). Therefore, neither general neoblast population dynamics nor cell turnover explained the co-*ythdf* suppression phenotype, pointing instead to subtler, lineage-specific alterations.

To test this hypothesis, we examined if following the co-*ythdf* suppression the emerging dd_1837[+] cell population had the characteristics of recently produced post-mitotic progenitors, namely SMEDWI-1 expression. We detected dd_1837[+] cells using FISH combined with SMEDWI-1 immunolabeling ("Methods") in control and *co-ythdf* (RNAi) animals. We observed an increased

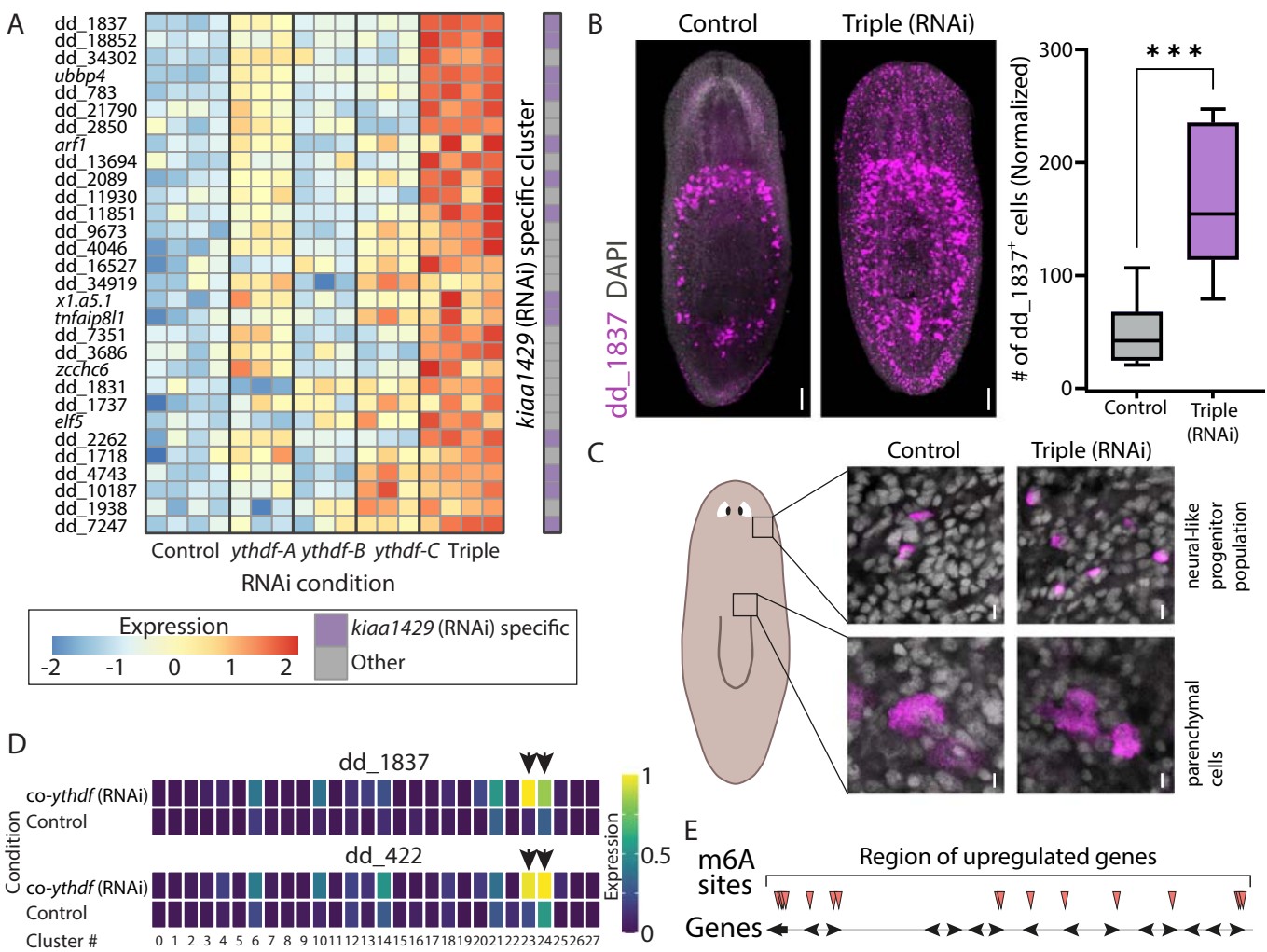

**Figure 6. Inhibition of *ythdf* genes results in excessive production of an abnormal cell population.**

(A) Heatmap of the top 30 upregulated genes following co-inhibition of *ythdf*s, compared to individual *ythdf* suppression and control (FDR < 1 × 10⁻⁵). Displayed are z-scores ranging from −2 to 2. Rows represent genes, and columns represent samples. Blue and red indicate low to high gene expression, respectively. Columns represent biological replicates. The rightmost column denotes whether the gene is highly expressed in the *kiaa1429* (RNAi)-specific cell population (Dagan et al, 2022). (B) FISH analysis detecting the *kiaa1429* (RNAi)-specific cell population marker gene dd_1837 following co-suppression of *ythdf*s and in control animals. dd_1837⁺ cells in the head region were quantified, and counting was normalized to animal size ("Methods"), revealing a significant increase in dd_1837⁺ cells in the RNAi animals compared to controls (Student's two-tailed *t* test ***P value = 2 × 10⁻⁴, group size *n* > 8). Scale bar = 100 μm. (C) High magnification images of dd_1837⁺ cells demonstrating the difference between the neural-like progenitor population (top), which is abundant throughout the planarian body, and the parenchymal population (bottom), which consists of larger cells localized around the pharynx. Scale bar = 10 μm. (D) Comparison of dd_1837 and dd_422 expression in control and co-*ythdf* (RNAi) scRNAseq samples ("Methods"). The most pronounced differences (black arrows) were detected in clusters 23 and 24, corresponding to subsets of the *mag-1⁺* parenchymal and neural populations, respectively. Blue to yellow color denotes low to high mean expression per cluster ("Methods"). (E) Mapping of m6A sites across upregulated genes in the repetitive gene cluster that is overexpressed in *kiaa1429* (RNAi)-specific cells. Red arrowheads indicate methylation sites based on the GLORI analysis (see Datasets EV1 and EV5). Source data are available online for this figure.

number of dd_1837⁺/SMEDWI-1⁺ cells in co-*ythdf* (RNAi) animals compared to controls (Fig. 7D, Student's *t* test, *P* < 1 × 10⁻⁴). Importantly, this increase was not accompanied by a change in the total number of SMEDWI-1⁺ cells (Fig. 7C), indicating that the expansion of dd_1837⁺ progenitor-like cells was not due to altered neoblast abundance. Based on these results, we hypothesized that the increase in the dd_1837⁺ progenitor-like population following co-suppression of *ythdf*s might have represented disrupted differentiation. Specifically, the loss of YTHDF proteins could interfere with the proper regulation of post-mitotic progenitor

maturation, leading to an abnormal expansion of progenitor cells, such as the dd_1837⁺/SMEDWI-1⁺ population, which fail to fully differentiate.

To further investigate the hypothesis that different lineages were differentially affected by the *ythdf* suppression, we examined whether the observed decrease in parenchymal cells (e.g., dd_940⁺; Figs. 5D and EV8F) could be attributed to a reduction in the number of parenchymal progenitors. We combined FISH using a parenchymal marker, dd_940, with SMEDWI-1 IF in control and RNAi animals (Fig. 7E). Quantification of

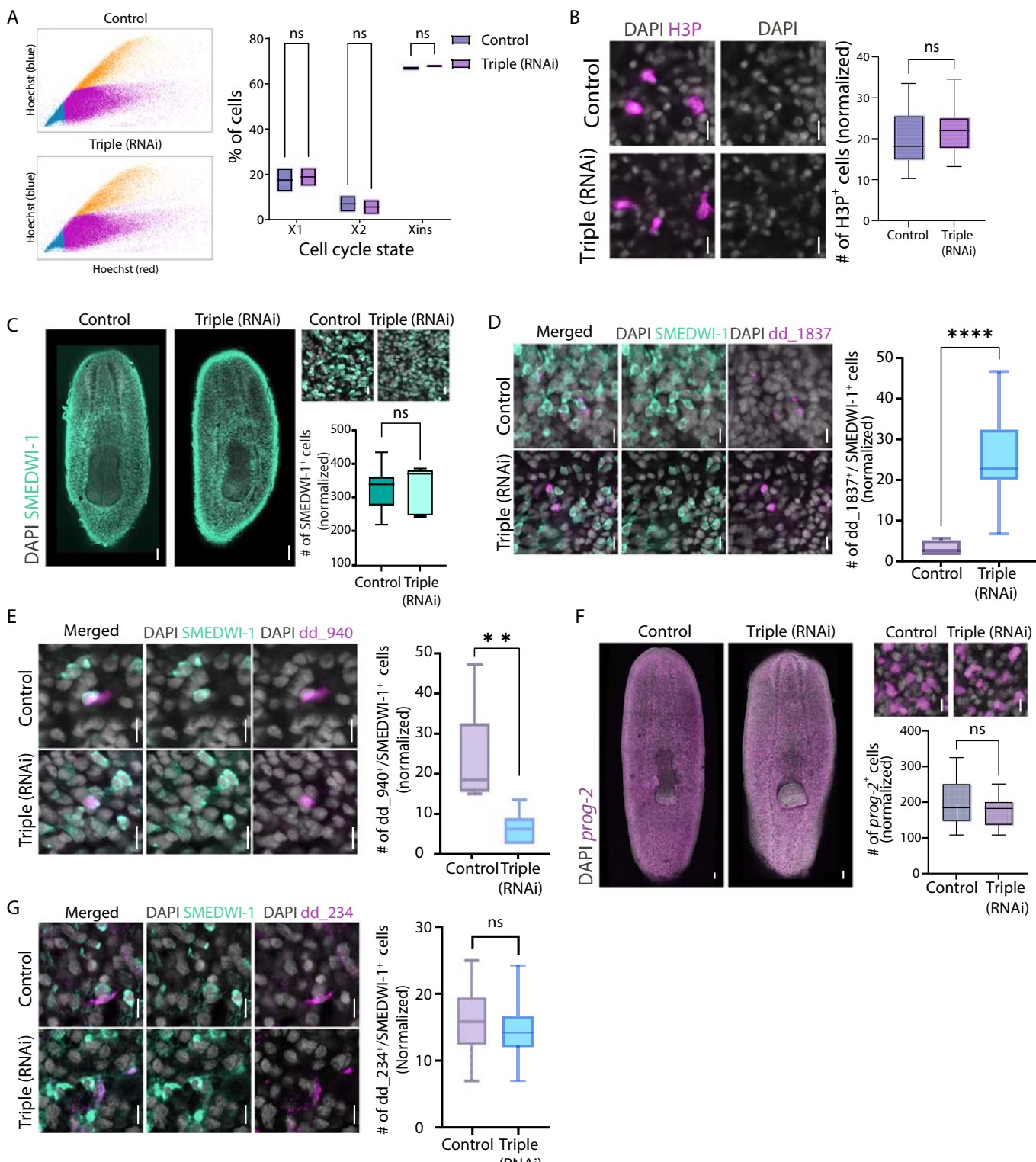

dd_940[+]/SMEDWI-1[+] cells revealed a strong reduction in this parenchymal progenitor population following *ythdf* co-suppression (Fig. 7E; Student's two-tailed *t* test $P = 0.0052$), indicating that YTHDF gene expression regulated this parenchymal progenitor population.

We further examined if *ythdf* inhibition resulted in lineage- and cell-type-specific consequences by measuring the abundance of progenitors of cell types that appeared unperturbed by the *ythdf* suppression (Dataset EV5). First, we quantified the number of epidermal progenitors (*prog-2*[+]) using FISH (Eisenhoffer et al,

**Figure 7.  Expression of YTHDF proteins regulates the size of distinct progenitor populations.**

(A) Representative FACS plots of cells isolated from control and co-*ythdf* (RNAi) animals (left). Quantification of cell proportion within each FACS gate showed no significant differences between control and co-*ythdf* (RNAi) animals (right). FACS experiments were performed in biological duplicates. Boxes represent the interquartile range (IQR), whiskers indicate 1.5 × IQR, and the central line denotes the median. (B) Quantification of H3P$^+$ cells labeling mitotic cells in co-*ythdf* (RNAi) and control animals. H3P$^+$ cells were counted from a single Z-plane per animal and normalized to animal size. No significant difference in the number of mitotic cells was observed between co-*ythdf* (RNAi) and control animals (Student's two-tailed t test P value > 0.05, group size n > 14; Boxes represent the IQR, whiskers represent min to max, and central band represents the median). (C) Quantification of the SMEDWI-1$^+$ population size by IF is shown following co-suppression of *ythdf*s and in controls. SMEDWI-1$^+$ cells were counted in the region between the brain and pharynx, and normalized to animal size, revealing no significant difference in cell number (Student's two-tailed t test P value > 0.05, group size n > 7; Boxes represent the IQR, whiskers represent min to max, and central band represents the median). Scale bar = 100 μm. (D) Detection of dd_1837 and SMEDWI-1 by FISH and IF, respectively, is shown following co-*ythdf* suppression and in controls. Cell counts of dd_1837$^+$/SMEDWI-1$^+$ cells in the head region were normalized to animal size. A significant increase in dd_1837$^+$/SMEDWI-1$^+$ cells was observed in the co-*ythdf* (RNAi) condition (Student's two-tailed t test; P value = $7.93 \times 10^{-5}$; group size n > 9. Boxes represent the IQR, whiskers represent min and max, and central band represents the median). Scale bar = 10 μm. (E) Detection of the parenchymal cell marker dd_940 and SMEDWI-1 by FISH and IF, respectively, is shown following co-*ythdf* suppression and in controls. Cell counts of dd_940$^+$/SMEDWI-1$^+$ in the region between the brain and pharynx were normalized to animal size and compared to control. A significant decrease in dd_940$^+$/SMEDWI-1$^+$ cells was observed in the RNAi animals (Student's two-tailed t test, P value = 0.0052; group size n > 7. Boxes represent the IQR, whiskers represent min and max, and central band represents the median). Scale bar = 10 μm. (F) Detection of epidermal progenitors using the cell marker *prog-2* by FISH, in control and co-*ythdf* (RNAi) animals. *prog-2*$^+$ cells were counted between the brain and pharynx compared to controls following animal size normalization. No significant difference was observed (Student's two-tailed t test, P value > 0.05; group size n > 11). Scale bar = 10 μm. (G) Detection of the *cathepsin*$^+$ progenitor cells using the gene dd_234 and SMEDWI-1 by FISH and IF, respectively, in control and co-*ythdf* (RNAi) animals. Detected dd_234$^+$/SMEDWI-1$^+$ cells between the brain and pharynx were counted. Counts were normalized to animal size. No significant difference was observed in the number of dd_234$^+$/SMEDWI-1$^+$ cells between RNAi animals and controls. (Student's two-tailed t test, P value > 0.05; group size n > 17). Scale bar = 10 μm. Source data are available online for this figure.

2008; Tu et al, 2015; Wurtzel et al, 2017). Our analysis revealed no significant difference in the number of epidermal progenitors between control and co-*ythdf* (RNAi) animals (Fig. 7F). We next examined a progenitor population representing a subset of the *cathepsin*$^+$ cells (Fincher et al, 2018) (dd_234$^+$), which appeared to be unaffected by the RNAi (Dataset EV5). Indeed, FISH labeling with dd_234 and SMEDWI-1 immunolabeling revealed no significant difference in dd_234$^+$/SMEDWI-1$^+$ cells following co-*ythdf* (RNAi) (Fig. 7G).

Taken together, these results demonstrated that YTHDF proteins were not required for overall stem cell maintenance or proliferation, but were essential for the proper production of specific progenitor populations. They further show, for the first time in planarians, that the production of specific progenitor populations is regulated by the combined activity of YTHDFs, likely co-expressed in the same cell, revealing a hidden layer of gene expression regulation. This model suggests that planarian YTHDFs act downstream of general neoblast proliferation and potentially upstream of terminal differentiation, having important roles in facilitating cellular maturation.

## Discussion

m$^6$A is an essential mRNA modification in diverse biological systems (Meyer and Jaffrey, 2017). In planarians, studying the MTC has revealed its essential role in producing intestine cells (Dagan et al, 2022). Analysis of a gene encoding the putative planarian m$^6$A nuclear reader, *ythdc-1*, has revealed nearly identical functions (Dagan et al, 2022). Yet, the presence of multiple other m$^6$A pathway readers (Dagan et al, 2022), together with the widespread abundance of m$^6$A on planarian mRNA, suggests that the pathway has additional roles in other cell types and contexts.

Previous analyses of m$^6$A in planarians, similar to many other systems, lacked single-nucleotide resolution, making it challenging to pinpoint true m$^6$A sites, especially given the short installation motifs (Dagan et al, 2022; Cui et al, 2023). Our study provides the first single-nucleotide resolution map of m$^6$A in planarians.

Notably, per-site m6A:A stoichiometry was substantially higher in planarians than in mammals (Liu et al, 2023) or plants (Wang et al, 2024), indicating a more 'switch-like' modification regime than reported in other profiled taxa. Analysis of the detected m$^6$A sites revealed that m$^6$A deposition in planarians is governed by relatively simple sequence determinants, with strict requirements that distinguish compatible from incompatible sequences. We found that adenosines followed by cytosine at the +1 position are robust candidates for methylation. By contrast, when uracil was found at the +1 position, additional sequence elements (e.g., +4 U) were required to facilitate m$^6$A installation. Certain sequences, particularly purines at the +1 position, were completely refractory to methylation. This pattern suggests that, in planarians, evolution acts chiefly by gaining or losing compatible m$^6$A sites, not by adjusting per-site methylation levels. The fact that many bona fide m$^6$A sites diverge from the canonical DRACH motif (Dominissini et al, 2012) might reflect an adaptation of the m$^6$A pathway to the low (30%) GC content of planarian genomes (Vila-Farré et al, 2023). These results echo plant studies reporting m6A enrichment at GAU motifs (Wang et al, 2024; Arribas-Hernández et al, 2021a). Given the evolutionary distance between planarians and plants, broader taxon sampling is needed to determine whether this similarity represents convergent evolution or deep conservation. Either way, the shared motif highlights the MTC flexible substrate recognition across distant lineages, a property, to our knowledge, not yet detected in animals beyond planarians.

Our single-nucleotide resolution analysis of m$^6$A indicated that methylation of adjacent sites was frequently mediated as independent events. Despite the lack of linkage in m$^6$A installation, the moderate correlation between m$^6$A scores of nearby sites indicated that additional local characteristics of the transcript (e.g., general accessibility to the MTC) contributed to methylation level (Uzonyi et al, 2023). This model of m$^6$A deposition suggested that each site was regulated as an autonomous unit. Therefore, the functional impact of m$^6$A may be distributed over the transcript rather than being driven by individual sites. Moreover, the potential lack of direct association between methylation of adjacent sites suggested that genes can gain or lose m$^6$A sites without impacting the

potential functionality of remaining sites. The gain and loss of sites can occur rapidly through single-nucleotide mutations (e.g., +1 C to +1 G). This scenario supports an evolutionary model in which selective pressures for retaining m⁶A sites in a gene could act primarily at the level of transcript function rather than by strict nucleotide conservation.

In addition to installation, our work highlights the role of m⁶A in regulating cellular processes. In vertebrates, the three YTHDF paralogs appear to have highly redundant functions in regulating m⁶A-modified transcripts, predominantly facilitating mRNA turnover (Kontur et al, 2020; Lasman et al, 2020b; Zaccara and Jaffrey, 2020). In planarians, there are five YTHDF homologs, which we found to have an evolutionary history distinct from their vertebrate counterparts. These planarian *ythdf*s are weakly but broadly expressed (Fincher et al, 2018; Plass et al, 2018; Dagan et al, 2022), and their overlapping expression patterns hint at functional redundancy. Although techniques like CLIP-seq would be ideal for assessing in vivo binding affinities (Zaccara and Jaffrey, 2024), such studies are currently precluded by the lack of antibodies for planarian YTHDFs; moreover, in vitro pulldown assays are unlikely to fully recapitulate the native binding dynamics. Indeed, suppression of an individual *ythdf* does not result in a phenotype. By contrast, suppression of multiple *ythdf*s caused a size reduction, and disrupted neoblast differentiation toward multiple parenchymal and *cathepsin*⁺ cell types, processes that are dynamic and require tight regulation of transcriptional programs (Scimone et al, 2018; Raz et al, 2021; Frankovits et al, 2025). The effects observed after co-*ythdf* suppression were lineage-restricted, not a global disruption of neoblast differentiation. They likely reflect two factors: first, reader-dosage redundancy across progenitor lineages with YTHDFs operating at different levels in distinct lineages, and second, the lineage-specific architecture of m⁶A targets with transcripts methylated differently across cell types. Consistent with this model, epidermal and dd_234⁺ *cathepsin*⁺ progenitors were unaffected by co-*ythdf* suppression, whereas other lineages (e.g., dd_940⁺ parenchymal) showed pronounced m⁶A dependence. Taken together, these findings indicate no global progenitor defect, but instead context-dependent sensitivity to YTHDF activity and m⁶A regulation.

Our observations are compatible with the possibility that planarian YTHDFs may be molecularly redundant, but because of the distinct evolutionary histories (e.g., gene duplications) of planarian and vertebrate YTHDFs, such redundancy is likely the consequence of independent processes. Importantly, YTHDF functions show both redundancy and specialization across eukaryotes. In *Arabidopsis*, for instance, some YTHDF paralogs act redundantly, while others perform specialized roles (Arribas-Hernández et al, 2021a, 2021b; Flores-Téllez et al, 2023). We therefore suggest that the balance between redundancy and specialization is shaped by species-specific duplication histories and expression programs, rather than being hardwired to particular YTHDF identities.

Our findings contribute to a broader understanding of epitranscriptomic regulation in this regenerative organism. In planarians, the straightforward sequence requirements for m⁶A installation coupled with the redundant functionality of YTHDF m⁶A readers suggest a flexible mechanism to fine-tune gene expression rapidly. This flexibility is likely crucial for processes such as stem cell differentiation and tissue regeneration, where rapid shifts in transcript abundance are required. Comparative

studies will likely be instrumental in revealing how variations in m⁶A regulatory mechanisms contribute to the diverse strategies for regulation of gene expression across species.

# Methods

### Reagents and tools table

| Reagent/resource | Reference or source | Identifier or catalog number |
| --- | --- | --- |
| **Experimental models** | | |
| *Schmidtea mediterranea*, asexual isolate | Sánchez Alvarado Lab, Stowers Institute | |
| **Recombinant DNA** | | |
| pGEM-T Easy vector | Promega | A1360 |
| **Antibodies** | | |
| Anti-DIG-POD | Roche | 11207733910 |
| Anti-FITC-POD | Roche | 11426346910 |
| Anti-SMEDWI-1 (mouse monoclonal) | Gift from Dr. Jochen Rink | |
| Goat anti-mouse HRP-conjugated | Abcam | ab6721 |
| Anti-phospho-Histone H3 (H3P) | Sigma | 04-817 |
| Goat anti-rabbit HRP | Abcam | ab6721 |
| **Oligonucleotides and other sequence-based reagents** | | |
| PCR primers | This study | Appendix Table S1 |
| qPCR primers | This study | Appendix Table S2 |
| GLORI indices | This study | Appendix Table S3 |
| **Chemicals, enzymes and other reagents** | | |
| Roche Western Blocking Reagent | Roche | 11921673001 |
| Heat-inactivated horse serum | Biological Industries | 04-124-1 A |
| TRI Reagent | Sigma | 9424 |
| RevertAid First Strand cDNA Synthesis Kit | Thermo Scientific | K1621 |
| RevertAid H Minus First Strand cDNA Synthesis Kit | Thermo Scientific | K1631 |
| TranscriptAid T7 High Yield Transcription Kit | Thermo Fisher | K0441 |
| SuperScript III Reverse Transcriptase | Thermo Fisher | 18080051 |
| ExoSAP-IT | Affymetrix | 75001 |
| T4 RNA Ligase 1 | NEB | M0437M |
| KAPA HiFi PCR Kit | KAPA Biosystems | KK2601 |
| AMPure XP beads | Beckman Coulter/ Agencourt | A63881 |
| NEBNext Poly(A) mRNA Magnetic Isolation Module | NEB | NEB-E3370 |
| NEBNext Ultra II Directional RNA Library Prep Kit | NEB | NEB-E7760S |

| Reagent/resource | Reference or source | Identifier or catalog number |
|---|---|---|
| RNA Fragmentation Reagents | Invitrogen | AM8740 |
| Dynabeads MyOne Silane | Invitrogen | 37002D |
| FastAP | Thermo Scientific | EF0652/3/4 |
| ApopTag Red In Situ Apoptosis Detection Kit | Merck | S7165 |
| GEM-X Universal 3′ Gene Expression v4 4-plex kit | 10x Genomics | 1000779 |
| **Software** | | |
| bcl-convert | Illumina | v4.2.7 |
| Cell Ranger | 10x Genomics | v9.0.1 |
| Seurat (R package) | Hao et al, 2024 | v5.3.0 |
| MAFFT | Katoh et al, 2019 | v7 |
| PhyML | Guindon and Gascuel, 2003 | PhyML |
| iTOL | Letunic and Bork, 2024 | Online service |
| Bowtie2 | Langmead and Salzberg, 2012 | v2.4.1 |
| featureCounts | Liao et al, 2014 | v2.0.0 |
| DESeq2 | Love et al, 2014 | v1.42.1 |
| g:Profiler | Kolberg et al, 2023 | Online service |
| Cutadapt | Martin, 2011 | v4.6 |
| HISAT-3 N | Zhang et al, 2021 | 2.2.1-3n-0.0.3 |
| Samtools | Li et al, 2009 | v1.19 |
| Picard tools | Broad Institute | v2.21.4 |
| txtools (R package) | Garcia-Campos and Schwartz, 2024 | 1.0.4 |
| Rsamtools (R package) | Bioconductor | 2.22.0 |
| Loupe Browser | 10x Genomics | v9 |
| Huygens Professional | Scientific Volume Imaging | v24.04 |
| OLYMPUS cellSens Dimension | Olympus | v3.2 |
| Thermo Fisher Cloud Platform | Thermo Fisher | Online service |
| Adobe Photoshop / Illustrator | Adobe | — |
| ImageJ | NIH | — |
| **Other** | | |
| Olympus IXplore SpinSR microscope | Olympus | — |
| Hamamatsu ORCA-Flash4.0 V3 camera | Hamamatsu | — |
| Zeiss LSM800 confocal microscope | Zeiss | — |
| Leica S9i stereomicroscope | Leica | — |
| BD FACSymphony S6 sorter | BD Biosciences | — |
| Chromium Controller | 10x Genomics | — |
| Illumina NextSeq 2000 / NextSeq 550 | Illumina | — |
| Illumina NovaSeq 6000 | Illumina | — |
| Drummond Nanoject III | Drummond Scientific | 3-000-207 |

## Sample fixation

Animals (*S. mediterranea* asexual isolate) were killed with 5% N-acetyl-cysteine in PBS for 5 min, followed by fixation in 4% formaldehyde diluted in PBSTx (PBS and 0.1% Triton X-100) for 20 min. Animals were then briefly washed in PBSTx, incubated in a 50:50 PBSTx:methanol solution for 10 min and stored in methanol at −20 °C until further analysis.

## Fluorescence in situ hybridization (FISH)

FISH was performed as previously described (King and Newmark, 2013) with minor modifications. Briefly, fixed animals were bleached with hydrogen peroxide and formamide for 2 h on a light table, then treated with proteinase K (2 µg/ml) in 1× PBSTx for 10 min followed by fixation in 4% formaldehyde for 10 min. After overnight hybridizations, samples were washed twice in pre-hyb solution, 1:1 pre-hyb-2× SSC, 2× SSC, 0.2× SSC, PBSTx. Blocking was performed in 0.5% Roche Western Blocking Reagent and 5% inactivated horse serum in 1× PBSTx. Animals were incubated in an antibody overnight at 4 °C (anti-DIG-POD, 1:1500; Roche, CAT11207733910). Post-antibody washes and tyramide signal amplification were carried out as previously described (King and Newmark, 2013). Finally, specimens were counterstained with DAPI overnight at 4 °C (Sigma, 1 µg/ml in PBSTx).

## Fluorescence in situ hybridization (FISH) combined with immunofluorescence

Fixed animals were rehydrated and bleached with hydrogen peroxide and formamide for 2 h. They were then treated with proteinase K (2 µg/ml) in 1× PBSTx for 10 min, followed by fixation in 4% formaldehyde for 10 min. After overnight hybridization, samples were sequentially washed in pre-hybridization solution, 1:1 pre-hybridization: 2× SSC, 2× SSC, 0.2× SSC, and PBSTx. For immunostaining, animals were blocked in PBSTB (PBS with 0.1% Triton X-100 and 0.25% BSA) and incubated overnight at 4 °C with anti-SMEDWI-1 antibody (1:4000; kindly provided by Dr. Jochen Rink). After incubation, animals were rinsed in PBSTB and washed seven times over 4 h. Then, samples were labeled overnight with a goat anti-mouse HRP-conjugated antibody (1:300; Abcam, CAT ab6721). Following six washes over 3 h in PBSTB, antibody development was performed using the tyramide signal amplification (TSA) system with FITC-tyramide (1:1500), as previously described (King and Newmark, 2013). Signal inactivation was achieved by treating specimens with 1% sodium azide for 1 h. The FISH protocol continued with blocking in 0.5% Roche Western Blocking Reagent (CAT #11921673001) and 5% inactivated horse serum (Biological Industries CAT #04-124-1 A) in 1× PBSTx. Animals were incubated overnight at 4 °C with an anti-DIG-POD antibody (1:1500). Post-antibody washes and tyramide development were carried out as previously described (King and Newmark, 2013). Finally, animals were counterstained with DAPI (1 µg/ml in PBSTx, Sigma; 1:5000) overnight at 4 °C.

## RNA purification

Animals were collected into 700 µl TRI Reagent (Sigma; CAT #9424) and homogenized using 0.5 mm zirconium beads in a bead-beating homogenizer (Allsheng; Bioprep-24) for two cycles of 45 s

each and 15 s hold in between, at 3500 RPM followed by incubation at room temperature for 5 min. Next, 140 µl of chloroform was added to each tube, followed by vigorous shaking for 15 s and incubation at room temperature for 3 min. Then, samples were centrifuged at 4 °C, 12,000× g for 25 min for phase separation. The upper phase was transferred into a new tube and 500 µl of isopropanol was added. Tubes were inverted five times and incubated for 10 min at room temperature. RNA was precipitated by centrifugation at 12,000× g for 45 min at 4 °C. The resulting RNA pellet was washed twice with 75% ethanol, followed by centrifugation at 7500× g for 5 min at 4 °C. After air-drying for 10 min, RNA was resuspended in 30 µl of nuclease-free water. RNA concentration was measured by Qubit (Invitrogen; Q33226) according to the manufacturer's protocol.

## Molecular cloning

Planarian cDNA was synthesized from total RNA using RevertAid First Strand cDNA Synthesis Kit (Thermo Scientific™, CAT K1621). Target gene amplification was performed using gene-specific primers (Appendix Table S1) and the resulting PCR products were cloned into pGEM-t vector using the manufacturer's protocol (Promega; CAT A1360). Plasmids were delivered into *E. coli* TOP10 (Thermo Fisher Scientific) by the heat-shock method. Briefly, 100 µl of bacteria was mixed with 5 µl of each of the cloned vectors, incubated on ice for 30 min, and subjected to heat shock at 42 °C for 45 s. Transformed bacteria were supplemented with 350 µl of SOC medium, and incubated at 37 °C for 1 h for recovery. Following recovery, bacteria were plated on agarose plates containing 1:2000 ampicillin, 1:200 isopropyl-β-D-thiogalactoside (IPTG), and 1:625 5-bromo-4-chloro-3--indolyl-β-D-galactopyranoside (X-gal). Plates were incubated overnight at 37 °C. Sequences of purified plasmids were validated by Sanger sequencing.

## Synthesis of dsRNA for RNAi experiments

Double-stranded RNA (dsRNA) was synthesized as previously described (Rouhana et al, 2013). Briefly, in vitro transcription (IVT) templates were prepared by PCR amplification of cloned target sequences using primers with 5′ flanking T7 promoter sequences. dsRNA was synthesized using the TranscriptAid T7 High Yield Transcription Kit (ThermoFisher; CAT K0441). Transcription reactions were incubated overnight at 37 °C, followed by treatment with RNase-free DNase for 20 min. RNA was purified by ethanol precipitation and resuspended in 25 µl of double-distilled water (ddH$_2$O). The integrity of dsRNA was assessed on a 1% agarose gel, and its concentration was quantified using a Qubit 4 fluorometer (Thermo Scientific), ensuring a final concentration above 5 µg/µl. In all RNAi experiments, *S. mediterranea* asexual animals were used. Animals were starved for at least 7 days before dsRNA feeding. Animals were fed a mixture of dsRNA and beef liver in a 1:2 ratio twice a week. During the experiment, the animals were visually inspected to validate food uptake.

## Whole-mount in situ hybridization for detection of *ythdf* expression

Whole-mount in situ hybridization was conducted as described previously (King and Newmark, 2013) with a few modifications: In brief, worms were euthanized in 5% NAC in PBS, fixed in 4% PFA in 50% PBS containing 0.15% Triton X-100 and dehydrated in 100% MeOH. After rehydration in PBS containing 0.3% Triton X-100 (PBSTx), worms were bleached in a bleaching solution (1.2% H$_2$O$_2$, 5% formamide, 0.5×SSC in H$_2$O) on a light table for ~1 h and treated with 2 µg/mL Proteinase K (NEB) in PBSTx for 10 min followed by post-fixation in 4% PFA for 10 min. The samples were placed in pre-hyb (50% Formamide, 5× SSC, 1× Denhardt's solution, 100 µg/µl heparin, 1% Tween 20, 1 mg/ml torula yeast RNA, 50 mM DTT) at 58 °C for 2 h and then in hyb (50% formamide, 5× SSC, 1× Denhardt's solution, 100 µg/µl heparin, 1% Tween 20, 0.25 mg/ml torula yeast RNA, 50 mM DTT, 0.05 g/ml dextran sulfate) with the RNA probes at 58 °C overnight. The next day, samples were washed with wash hyb (50% Formamide, 0.5% Tween 20, 5×SSC, 1× Denhardt's solution); 1:1 wash hyb: 2×SSC (0.1% Tween 20); 2×SSC (0.1% Tween 20); 0.2×SSC (0.1% Tween 20) at 58 °C for 1 h (each solution). Samples were then incubated in blocking solution (5% sterile horse serum, 0.5% Roche western blocking reagent in PBSTx) for 1 h at room temperature. Following blocking, samples were incubated with antibody (anti-DIG-POD (Roche), anti-FITC-POD (Roche)) in blocking solution (1:2000 from the antibody stocks) at 4 °C overnight. Fluorophore development was done through tyramide amplification by incubating samples in TSA buffer (2 M NaCl, 0.1 M boric acid in H$_2$O, pH 8.5) containing 0.006% H$_2$O$_2$, 20 µg/ml 4-Iodophenylboronic acid and tyramide (Rhodamine or FAM) for 30 min at room temperature. For two-color in situ, peroxidase activity of the first antibody was quenched by incubating in 200 mM sodium azide in PBSTx for at least 1 h at room temperature. The color reaction of the second antibody was then developed exactly like the first one. After development, samples were placed in Scale S4 (10% glycerol, 15% DMSO, 40% sorbitol, 4 M Urea, 2.5% DABCO, 0.1% Triton X-100 in H$_2$O) at 4 °C overnight and mounted on slides the next day.

## Imaging of *ythdf*s FISH

Imaging the fixed samples was performed using an Olympus IXplore SpinSR microscope equipped with a Hamamatsu Orca Flash4.0 V3 camera. Images were acquired using OLYMPUS cellSens Dimension 3.2 software. For overview images, stitching was conducted using Huygens Professional 24.04. Three samples were imaged per condition, capturing both dorsal and ventral overviews using an Olympus UPLXAPO 10x air objective (NA = 0.4). Then, high resolution images of the head and tail regions were obtained using an Olympus UPLXAPO 20x air objective (NA = 0.8).

## Cell counting in microscopy images

Fluorescence and confocal images were acquired using a confocal microscope (Zeiss LSM800), and live images were taken using a stereomicroscope (Leica S9i). Cell counting was performed on images of animals captured from either the dorsal or ventral side, depending on the gene expression pattern. Sections for counting were imaged z-stacks with 4 µm optical sections, spanning the intestinal lumen to the epidermis. Cell quantification using gene expression markers was conducted on at least ten animals per condition, unless noted in the figure legend. Counting included the region anterior to the pharynx, with the number of cells normalized

to the quantified area, unless indicated otherwise. Parenchymal cell counts were performed on a z-projection of the entire animal, while markers of other cell types were quantified section by section within a defined region indicated in the figure legend. Images were processed using Adobe Photoshop, and masks of planarian outlines were defined using the Lasso and Object Selection tools. The background color outside the defined mask was standardized in Adobe Photoshop or Adobe Illustrator. Cell counting and analysis were performed using ImageJ.

## RNA quantification, cDNA synthesis, and RT-PCR analysis

RNA concentration for each sample was measured and normalized using a Qubit 4 fluorometer (Invitrogen, Q33226). cDNA synthesis was performed on 1 μg of RNA from each sample using the RevertAid H Minus First Strand cDNA Synthesis Kit (Thermo Scientific, CAT K1631) according to the manufacturer's protocol, using oligo(dT) primers. Target gene expression (Appendix Table S2) was measured using the QuantStudio 3 Real-Time PCR system (Applied Biosystems), with two technical replicates per sample and at least two biological replicates per condition. The relative gene expression fold-change was calculated using the ΔΔCt method, with *gapdh* expression used as the endogenous control. Forward and reverse primers for each target gene were designed, and primer efficiency was tested prior to the experiment using the standard curve method, employing five cDNA concentrations. Analysis was conducted via the Thermo Fisher Cloud Platform for data processing and visualization, and GraphPad Prism.

## Phylogenetic analysis planarian YTHDFs

Localization of YTH domains was determined across the planarian transcriptome (dd_v6) (Rozanski et al, 2019) using InterPro analysis (PF04146; PTHR12357:SF65) (Blum et al, 2025) on predicted protein sequences of the planarian transcriptome. The predicted protein sequences of YTH domains were extracted from *ythdc/df* genes from various species representing a diversity of animals: (*Homo sapiens, Mus musculus, Drosophila melanogaster, Danio rerio, Schmidtea mediterranea, Polycelis nigra, Dugesia japonica, Macrostomum lignano, Nematostella vectensisi, Capitella teleta, Hydra vulgaris,* and *Echinococcus multilocularis*). For sequences published in UniProt (Dataset EV3), the YTH domain was extracted manually from the "Family & Domain" section. The predicted polypeptide YTH domain sequences were aligned using MAFFT (Katoh et al, 2019) with an iterative refinement method L-INS-i. Then, phylogenetic analysis was performed by maximum likelihood-based inference using PhyML (Guindon and Gascuel, 2003) with parameters: [non-parametric bootstrap analysis with 100 replicates, number of relative substitution rate categories 8, and LG as the substitution model]. The resultant phylogenetic tree was visualized with iTOL (Letunic and Bork, 2024).

## RNA library preparation and sequencing

RNA sequencing libraries for co-*ythdf* (RNAi) and their control samples were prepared using 1 μg of total RNA. mRNA was isolated using NEBNext® Poly(A) mRNA Magnetic Isolation Module (NEB-E3370) and library preparation was performed using the NEBNext

Ultra II Directional RNA Library Prep Kit for Illumina (NEB-E7760S) according to the manufacturer's protocol. Sequencing was performed on an Illumina NextSeq 550 at the Life Sciences interdepartmental sequencing unit of Tel Aviv University. Libraries were sequenced paired-end with a read length of $2 \times 38$ bp.

## PolyA RNA sequencing analysis

Paired-end RNAseq files were mapped to the planarian transcriptome (ddV6) using Bowtie2 (version 2.4.1) with default parameters (Langmead and Salzberg, 2012). A gene expression matrix was produced using the featureCounts module from Subread-2.0 package with parameters [-s 2 -p] (Liao et al, 2014). Transcriptome assembly (dd_v6) was obtained from PlanMine (Rozanski et al, 2019) and counts of isotigs corresponding to the contig were summed (e.g., _0_1, _0_2, _0_3, etc.) to a single contig in the gene expression matrix (_0). Lowly expressed genes having a sum of less than 60 reads across samples were excluded from further analysis. DESeq2 (v1.42.1) was used to determine differentially expressed genes by pairwise gene expression analysis of each condition with control samples (Love et al, 2014). The differential expression analysis table was annotated using data extracted from the planarian cell-type-specific gene expression atlas (Fincher et al, 2018).

## Expression-matched bootstrap to test differential expression enrichment among m6A targets

Differential expression results were annotated with m6A status from GLORI (≥1 mapped site per gene). Genes were filtered by their DESeq2 determined expression (baseMean ≥200; Dataset EV5). Missing adjusted *P* values were set to 1 and differential expression was defined for this analysis as having adjusted *P* value < 0.05. To control for the bias that m6A detectability increases with expression, a stratified bootstrap was performed on $\log_{10}(\text{baseMean})$: the range was divided into ten quantile bins, and within each bin equal numbers of genes were sampled with replacement from gene sets with and without m6A. The per-bin sample size was set so that the total per group equaled $0.8 \times \min$ (genes with m6A, genes without m6A). The analysis was run for 1000 iterations. For each iteration, the proportion of differentially expressed genes was computed for each group and their difference recorded. An empirical two-sided *P* value was obtained across the bootstrap distribution, and 95% confidence intervals were taken as the 2.5th–97.5th percentiles. Expression-matching quality was summarized as the absolute difference in mean $\log_{10}(\text{baseMean})$ between sampled groups.

## Gene ontology analysis

Gene Ontology (GO) enrichment was performed on genes differentially expressed after co-suppression of the *ythdf* genes (adjusted *P* value $< 1 \times 10^{-4}$). GO annotations for the planarian transcriptome assembly were downloaded from PlanMine (Rozanski et al, 2019). Enrichment was computed with the g:Profiler server (Kolberg et al, 2023) with the following parameters [all results: false; ordered: false; no iea: false; domain scope: annotated; measure underrepresentation: false; significance threshold method: g_SCS; user threshold: 0.05; no evidences: false].

## Statistical analysis

Statistical analyses were conducted using GraphPad Prism v10 and R v4.3. Data are presented as described in the figure legends. Comparisons between two groups were performed using two-tailed unpaired Student's *t*-tests unless stated otherwise in the figure legend or "Methods". Comparisons among multiple groups were performed using one-way ANOVA followed by Dunnett's correction for multiple comparisons, unless stated otherwise. The number of independent biological replicates (*n*) is indicated in the figure legends. Statistical significance was defined as $P < 0.05$ unless noted otherwise. *P* values reported as $<2.2e{-}16$ indicate that the true *P* value is smaller than R's reporting threshold using double-precision floating-point arithmetic.

## Multiple sequence alignment of METTL and METTL14

Multiple sequence alignments were generated separately for METTL3 and METTL14. Protein sequences from planarian species were retrieved from PlanMine (Rozanski et al, 2019), and sequences from other metazoans and *Arabidopsis* were obtained from UNIPROT (UniProt Consortium, 2021). Incomplete sequences were manually excluded. Alignments were computed with MAFFT (Katoh et al, 2019) with parameters [--reorder --auto], which used iterative refinement strategies (L-INS-i/E-INS-i). Resulting MSAs were inspected with the MAFFT MSA Viewer (Katoh et al, 2019). Sequence IDs used for alignment shown in panel A [dd_Smed_v6_6641_1, dd_Djap_v4_42697_1_2, dd_Pnig_v3_13187_1_1, dd_Pten_v3_41598_1_1, gr_Mlig_v2_1509_14222_1, Q9HCE5, H2QQ29, A4IFD8, Q3UIK4, D4A701, Q5ZK35, A0A4W3GX50, Q6NU56, A0A8C5LPT8, Q6NZ22, A0A6P3VWG8, A0AAQ5X5J5, A0A1A8IUC4, A0A8C4YZM2, Q9VLP7, A0AAJ7SKX8, Q94AI4, XP_020877343]. Sequence IDs used for alignment shown in panel B [dd_Smed_v6_6450_0_2, dd_Djap_v4_56295_7_2, dd_Pten_v3_42496_1_1, dd_Pnig_v3_32465_1_1, A0A0R4J041, F7FFC6, A0A1U7R3Z3, Q86U44, G2HF97, G3QT25, A0A8C0RD13, F1S8J8, A6QQV4, G5BFU9, A0A9L0RMN8, A0A6I8NA09, A0A8V0XKP3, F1R777, A0A8C6NHA0, H2LSG5, OAP00159.1, CAE6131794.1, KAG7557359.1]; Sequences IDs used for alignment shown in panel C [dd_Smed_v6_6450_0_2, dd_Djap_v4_56295_7_2, dd_Pten_v3_42496_1_1, dd_Pnig_v3_32465_1_1, dd_Ptor_v3_22751_1_1, A0A0R4J041_MOUSE, F7FFC6, A0A1U7R3Z3, Q86U44, G2HF97, G3QT25, A0A8C0RD13, F1S8J8, A6QQV4, G5BFU9, H0UYU0, A0A9L0RMN8, A0A6I8NA09, A0A8V0XKP3, F1R777, A0A8C6NHA0 H2LSG5, OAP00159, CAE6131794, KAG7557359]. Sequence IDs starting with "dd" represent predicted protein sequences from planarian transcriptome assemblies.

## scRNAseq analysis of co-expression with *ythdf* genes

Analysis was performed on data previously published as part of the planarian cell atlas (Fincher et al, 2018) (GSE111764, Principal Clustering Digital Expression Matrix). A co-expression gene list was produced using FindMarkers, where ident.1 is defined as cells with non-zero expression level for the YTHDF gene in question, and min.pct set to 0. Each gene list was filtered for genes with p-adjusted value lower than 0.05, and average-log2 fold change larger than 0.25 to analyze significant co-expressed genes for each YTHDF. Defining lineage association for each gene was performed

using scRNAseq data available through Planarian Digiworm (Fincher et al, 2018). Each gene was associated with a specific lineage based on the marker genes of each main/sub cluster in the annotated dataset. This was performed using merge functions in R, creating tables containing the contig ID as well as main and sub clustering annotations. For bar chart analysis of these co-expressed genes indicating the frequency of how many genes are associated with each lineage, the number of genes associated with each cluster was divided by the number of co-expressed genes in each condition. This includes genes that are not associated with any main/sub cluster, although the unknown genes were not plotted in these graphs. Statistical significance was measured using a hypergeometric test for each bar in the bar charts. The number of genes in each lineage/cluster defines the gene population (N), the number of co-expressed genes overall is the sample size (n), and the number of genes associated with each lineage/cluster from the sample size is the number of successes (k). Each time the number of successes was tested to determine whether it is greater than the random variable (x), using a *P* value threshold of 0.05.

Co-expressed genes were assessed for overlap between *ythdf* homologs by finding common genes associated with each cluster. Of this union, the bar charts show frequency of overlap between *ythdf*s in terms of the co-expressed genes, by taking the number of genes for each cluster that are common for the two/three *ythdf*s divided by the union. This was performed for each lineage, followed by statistical evaluation that included performing a permutation analysis. For each *ythdf* co-expression list, we determined the number of genes associated with a specific lineage. We then randomly selected genes ($10^6$ iterations) from the pool of all genes linked to that lineage, ensuring that the size of each random set matched the number of genes in the original list. We repeated this process for pairs or the three of *ythdf* (A-C) co-expression lists, comparing the number of shared genes between the original lists to the overlap observed in the $10^6$ random iterations. A *P* value was then calculated to determine whether the overlap in the original lists was significantly greater than expected by chance, indicating a non-random association between the gene lists.

## GLORI treatment to RNA

For the GLORI protocol, 100–150 ng of double poly(A)-selected RNA per sample was used. RNA was fragmented to an approximate size of 300 nt using the Invitrogen RNA Fragmentation Reagents kit (Invitrogen, AM8740). Cleanup with Dynabeads MyOne Silane (37002D) was performed after each step, except those where the sample was eluted in water. DNase treatment and dephosphorylation were conducted by incubating each sample with T4 polynucleotide kinase (NEB), TURBO DNase, and FastAP (Thermo Scientific; EF0652/3/4) for 30 min at 37 °C in 5× FNKBuffer (1:1:2 ratio of T4 PNK buffer:FastAP buffer:$H_2O$). Subsequently, 3′ RNA barcode adapter ligation was performed using 100 pmol (Appendix Table S3) and 36 U of T4 RNA ligase (NEB) for 1.5 h at room temperature (23 °C) for each sample. After the 3′ adapter ligation, all samples were pooled, with 20% of the pool retained as a control-RNA sample. The remaining 80% of the pooled RNA underwent GLORI treatment as previously described (Shen et al, 2024). This involved deamination of the RNA with glyoxal and sodium nitrite followed by RNA deprotection and purification as outlined in the published protocol (Shen et al, 2024). cDNA synthesis for both control and treated pools

was carried out using the rTd reverse transcription primer (AGACGTGTGCTCTTCCG) and SuperScript III Reverse Transcriptase (SuperScript III, 18080051). After synthesis, residual rTd primer was removed by adding 3 μL of ExoSAP-IT (Affymetrix, 75001) and incubating at 37 °C for 12 min. RNA hydrolysis was then performed in 1 M NaOH at 70 °C for 12 min. For 5′ adapter ligation, 50 pmol of 5iLL-22 DNA adapter (/5Phos/ANNNNNNAGATCGGAAGA GCGTCGTGTAG/3ddC/) was ligated with 45 U of T4 RNA Ligase 1 (NEB, M0437M) for 4 h at room temperature. PCR amplification of cDNA was conducted using the KAPA HiFi PCR Kit (KAPA Biosystems KK2601). The resulting amplified cDNA libraries were then purified using AMPure XP beads (Agencourt, A63881). Sequencing was performed on Illumina NovaSeq 6000.

## Processing of GLORI libraries

GLORI library sequences were trimmed using cutadapt with parameters [--rrbs --paired -q 20 --stringency 1 -e 0.3 --length 15] (Martin, 2011). Then, GLORI libraries were mapped to the planarian genome. The *S. mediterranea* genome version schMedS1 downloaded from PlanMine (Rozanski et al, 2019), and mapping was performed by reduced-representation alignment with HISAT-3N (Zhang et al, 2021) with parameters [--sp 1,0 --max-intronlen 1000] either in paired-end or single-end mode. Mapping files were converted to BAM with samtools v1.19 (Li et al, 2009) and duplicates were marked by using PICARD tools MarkDuplicates function v2.21.4 [VALIDATION_-STRINGENCY = LENIENT REMOVE_DUPLICATES=true CRE-ATE_INDEX=true]. BAM files were processed using a custom R pipeline with the txtools package (Garcia-Campos and Schwartz, 2024). BAM files were loaded in 1 M read chunks using Rsamtools. Alignments were filtered by applying a scan flag that excluded duplicates (isDuplicate = FALSE), reads failing quality control (isNotPassingQualityControls = FALSE), and secondary alignments (isSecondaryAlignment = FALSE). Gene annotation was loaded from a BED file, and its corresponding genome sequence was used (Rozanski et al, 2019). Transcriptomic reads were extracted and filtered to remove read mapping exceeding 700 nt and requiring a minimum of 10 reads per transcript. A data table was then generated using the txtools covNuc mode (Garcia-Campos and Schwartz, 2024), which recorded both coverage and nucleotide frequencies. For each adenine position, counts of A and G nucleotides were obtained and used to compute a GLORI score as the fraction of A of the sum of A and G; positions with fewer than 10 combined A + G counts were discarded. An overall adenine frequency was estimated, and Fisher's exact tests were performed on positions with a GLORI score greater than 0.1 and a combined count of at least 10, with significance defined as $P < 0.05$.

## Metagene analysis of m⁶A position

A metagene plot was generated to visualize m⁶A site distribution along normalized gene lengths. Input data were filtered based on number of exons in gene, GLORI score, and replication support. If provided, coverage thresholds were applied. Gene lengths were normalized to a predefined scale (1000 nt). Single-exon genes were optionally retained or excluded. Sequence-based filtering was performed if genome sequences and motif constraints were specified by using regular expressions matching included or excluded sequences. Histogram or density plots were generated based on user preference.

## Sequence motif analysis of m⁶A sites

Sequences were extracted from the transcriptome FASTA file based on coordinates provided in the m⁶A sites table (Dataset EV1). Filtering was performed based on strand orientation, GLORI score thresholds, and minimum replicate support. Sequences were converted from DNA to RNA. Additional filtering was applied using wildcard patterns to retain or exclude specific sequences, as labeled in figures. Sequence logos displaying information content were generated using ggseqlogo (Wagih, 2017).

## Co-occurrence linkage analysis between adjacent m⁶A sites

Pairs of adjacent m⁶A sites were selected based on three criteria: (1) sites were within 40 nucleotides of each other, (2) each site had a median score between 0.4 and 0.7 in control libraries, and (3) one of the sites in the pair was identified as a dominant site. For each of the identified pairs, reads overlapping both sites were extracted from control BAM files using pysam library v0.22.1. Linkage between paired m⁶A sites was assessed to determine whether methylation at one site was associated with methylation at the other. A contingency table was constructed for every possible nucleotide combination across the paired sites, and a chi-square test of independence was applied to evaluate linkage between methylation statuses. Results of the linkage analysis were visualized in R.

## Gradient boosting regression analysis of m⁶A site sequences

A predictive model was trained to correlate sequence features with m⁶A site scores. The analysis included filtering the dataset to retain sequences with a median control coverage of at least 40 reads. Sequences around the m⁶A sites (6 nt upstream and downstream) were one-hot encoded using OneHotEncoder from scilearn v1.5.2. A gradient boosting regression model was utilized, and the dataset was divided into training and testing subsets to evaluate the model performance using mean squared error and $R^2$ score. The trained model was subsequently used to predict scores for the full dataset, and feature importance values were calculated to assess the contribution of individual features. Predicted scores were combined with metadata (e.g, distance of m⁶A site from EIJ, read depth coverage) for further analysis. Sequences having the top 1% percentile predicted m⁶A score were categorized based on experimental median control score into groups: "Coherent" (median control score >0.8) and "Low score" (median control score <0.1). Density plots were constructed in R to visualize the distribution of distances from the EIJ in the different categories.

## Analysis of m⁶A site conservation in five planarian species

Local sequence conservation of m⁶A sites was analyzed using custom Python and R scripts. First, a tab-delimited file of m⁶A sites was parsed to extract 13-nucleotide local motifs and their associated median m⁶A scores. The motifs were converted from RNA (U) to DNA (T) and then back to RNA notation (T to U) prior to analysis. Homologous sequences were identified by

matching site IDs from the m⁶A file to those in a homolog mapping CSV, and the corresponding FASTA files were selected based on a prefix index. For each m⁶A site, a sliding-window alignment was performed against the full sequence of each homolog. The alignment with the best Hamming score (i.e., the number of matching nucleotides) was retained. Alignments were discarded if the best score was ambiguous or if the Hamming distance exceeded a predefined threshold. Ambiguous mapping or matches with Hamming distance exceeding 3 were removed. Sequence logos were then generated using the ggseqlogo package (Wagih, 2017). For statistical evaluation of positional conservation, a two-sample proportion test was performed using the R function prop.test. For each position in the motif, the number of conserved nucleotides (as indicated by the match vector) was summed across all retained alignments, and the proportion of matches at that position was compared to the combined proportion of matches at all other positions. *P* values were calculated and subsequently adjusted using a Bonferroni correction to account for multiple testing.

## Fluorescence-activated cell sorting (FACS)

Planarians were macerated into small fragments using a blade. Tissue fragments were collected in calcium-free, magnesium-free buffer supplemented with 0.1% BSA (CMFB) and dissociated by gentle pipetting for 10 min. To minimize cell clumping, the suspension was passed through a 40 μm cell strainer. Cells were pelleted by centrifugation at $315\times g$ for 5 min at 4 °C, and resuspended in CMFB containing Hoechst 33342 (20 μl/ml) for 45 min at room temperature in the dark. Prior to sorting on a BD FACSymphony S6, cells were stained with propidium iodide (5 μg/ml) to assess viability. FACS gating was performed as previously described for planarian cell populations (Hayashi et al, 2006).

## Preparation of scRNAseq libraries

Animals were fed ten times with either non-targeting dsRNA (Control) or a mix of equal amounts of *ythdf-A*, *ythdf-B*, and *ythdf-C* dsRNAs. Following the RNAi feedings, animals were starved for one week. Cells from control or co-*ythdf* (RNAi) animals were isolated by FACS using Hoechst labeling as described above. Purified cells were counted and pelleted by centrifugation ($400\times g$, 5 min). The supernatant was discarded, and cells were resuspended in 1× PBS supplemented with 0.5% BSA to a final concentration of 1000 cells/μl. Cell counts were determined using a hemocytometer, and viability was assessed by Trypan blue (0.4%) exclusion.

Single-cell libraries were prepared at the Genomic Research Unit, Tel Aviv University, using the 10x Genomics Chromium Controller with the GEM-X Universal 3′ Gene Expression v4 4-plex kit with two biological replicates per condition (i.e., Control and co-*ythdf* (RNAi)) with barcoding performed as follows: [co-*ythdf* (RNAi) replicate #1: OB1; co-*ythdf* (RNAi) replicate #2: OB2; Control replicate #1: OB3; Control replicate #2: OB4]. Cell suspensions were diluted in nuclease-free water according to the manufacturer's instructions to 10,000 cells per sample. cDNA synthesis, barcoding, and library construction were performed following the manufacturer's protocol. The resulting libraries were sequenced on an Illumina NextSeq 2000 at Tel Aviv University

using a P2 sequencing kit. The libraries were sequenced in paired-end configuration as follows: [R1: 28; R2: 90; I1: 10; I2: 10].

## scRNAseq data processing and analysis

Raw sequencing files were converted to bcl using the bcl-convert-4.2.7 tool and then processed using the 10x Genomics cellranger-9.0.1 package *multi* command with the following parameters [--localmem 640 --localcores 90], with mapping of the data to the dd_v6 transcriptome sequence. Processing of the scRNAseq count matrices from the x10 Genomics pipeline was performed with the R Seurat package v5.3.0 (Hao et al, 2024). For each sample, a Seurat object was created [min.cells = 3; min.features = 200]. Per-cell quality metrics were computed, including the percentage of features matching a mitochondrial/unannotated contigs ("^mtRNA|unidentified") and a ribosomal-associated contigs ("dd-7-0|dd-4-1"). The four samples were merged into a single object with sample-specific cell IDs. Low-quality cells were removed using fixed thresholds [nFeature_RNA > 200, nFeature_RNA < 7000, and nCount_RNA > 500]. Counts were normalized by LogNormalize [scale.factor = 10,000], and 2000 highly variable features were selected with the "vst" method. All genes were scaled while regressing out ribosomal-associated contigs [vars.to.regress = "percent.rb"]. Principal component analysis (PCA) was performed on the variable features, and the first 20 PCs were used for downstream steps. A shared nearest-neighbor graph was constructed using the FindNeighbors function [dims = 1:20], clusters were identified using the FindClusters [resolution = 0.5], and UMAP and t-SNE embeddings were computed using the same 20 PCs. Cluster markers were identified with FindAllMarkers with params [only.pos = TRUE; min.pct = 0.25; logfc.threshold = 0.25). For cell-cycle analysis, S- and G2/M-phase gene lists were obtained from a previous analysis (Fincher et al, 2018). CellCycleScoring was then applied to assign per-cell phase. Differences in phase distributions between treatments within each cluster were evaluated by chi-squared tests. The merged Seurat object was exported to a 10x Loupe Browser file using the loupeR function create_loupe_from_seurat. Seurat automatically determined the presence of 27 clusters (0 to 26). The exported loupe object was loaded in the Loupe cell viewer (x10 Genomics; v9), and cluster identity was annotated manually by comparing the expression of top-cluster markers with published cell-type-specific gene expression markers from the planarian cell atlas and the planarian gene expression tools (Fincher et al, 2018; Hoffman and Wurtzel, 2023). Cluster labels, which include additional numeric identity in parenthesis (e.g., intestine (4)) refer to the sub-cluster annotation in Fincher et al planarian cell atlas. The following markers were used when assaying cluster identity: [0: dd_332_0, dd_3549_0, dd_363_0, dd_61_0; 1: dd_4804_0, dd_2548_0, dd_10988_0, dd_6732_0, dd_14111_0; 2: dd_12332_0, dd_14806_0, dd_11487_0, dd_12381_0, dd_7544_0; 3: dd_15344_0, dd_10394_0, dd_14937_0, dd_7664_0, dd_11349_0; 4: dd_10859_0, dd_3902_0, dd_2972_0, dd_10142_0, dd_10527_0; 5: dd_2408_0, dd_13413_0, dd_2092_0, dd_5615_0, dd_1706_0; 6: dd_238_1, dd_801_0, dd_3704_0, dd_750_0, dd_7752_0; 7: dd_158_0, dd_5644_0, dd_3517_0, dd_2286_0, dd_2614_0; 8: dd_508_0, dd_1410_0, dd_6915_0, dd_2869_0, dd_9050_0; 9: dd_10059_0, dd_4454_1, dd_5530_0, dd_2716_0, dd_7541_0; 10: dd_12112_0, dd_3063_0, dd_14331_0, dd_3093_0, dd_11968_0; 11: dd_5391_0, dd_5958_0, dd_480_0, dd_3226_0, dd_2147_0; 12: dd_10872_0, dd_7251_0, dd_10044_0, dd_16064_0;

13: dd_1240_0, dd_1171_0, dd_267_0, dd_194_0, dd_75_0; 14: dd_19825_0, dd_13662_0, dd_19428_0, dd_17551_0, dd_8425_0; 15: dd_11520_0, dd_4575_0, dd_7742_0, dd_15254_0, dd_23871_0; 16: dd_3101_0, dd_1127_0, dd_9650_0, dd_2923_0, dd_888_0; 17: dd_6553_0, dd_1477_1, dd_4905_0, dd_13564_0, dd_3640_0; 18: dd_351_0, dd_950_0, dd_364_0, dd_357_0, dd_1688_0; 19: dd_13986_0, dd_1152_2, dd_10552_0, dd_12126_1; 20: dd_1248_0, dd_9633_0, dd_210_1, dd_210_0, dd_3069_0; 21: dd_11710_0, dd_13583_0, dd_13376_0, dd_4314_0, dd_940_0, dd_924_0; 22: dd_7616_0, dd_4166_0, dd_766_0, dd_1700_0, dd_372_0; 23: dd_8929_0, dd_26589_0, dd_15304_0, dd_43726_0, dd_451_0; 24: dd_17986_0, dd_16581_0, dd_12988_0, dd_21108_0, dd_12014_0; 25: dd_9375_0, dd_1103_0, dd_20538_0, dd_2457_0, dd_52946_0; 26: dd_10820_0, dd_6737_0, dd_1161_1, dd_7816_0, dd_626_0]. Cluster 28 (id: 27) was identified and separated manually as protonephridia using the following markers: [27: dd_4841_0, dd_5784_0, dd_1213_0, dd_2210].

Treatment-comparison heatmaps were generated between treatments (e.g., Control and co-*ythdf* (RNAi)). log-normalized expression values were used to compute mean expression for every cluster-treatment combination. Cluster-by-treatment grid was generated, and missing combinations were imputed as 0 to maintain alignment across conditions. To enable consistent visual comparison across genes with different dynamic ranges, expression was normalized per-gene basis to a 0 to 1 scale.

## H3P labeling

H3P labeling was performed as previously described (Wenemoser and Reddien, 2010; LoCascio et al, 2017) with minor modifications. Animals were fixed in Carnoy's solution and bleached overnight in 6% $H_2O_2$ in methanol on a light table. Samples were rinsed twice in methanol, washed twice in PBSTx (0.1% Triton X-100), and treated with proteinase K (2 µg/ml, NEB) in PBSTx for 10 min. Post-fixation was performed in 4% formaldehyde for 10 min, followed by three 10 min washes in PBSTx. Samples were blocked in 10% horse serum for 2 h at room temperature and incubated overnight at 4 °C with anti-phospho-Histone H3 antibody (Sigma 04-817) diluted 1:100 in blocking solution. The following day, samples were washed seven times in PBSTx (0.1%) for 20 min each, blocked again in 10% horse serum for 2 h, and incubated overnight at 4 °C with goat anti-rabbit HRP (Abcam; ab6721) diluted 1:100 in blocking solution. Samples were then washed seven times in PBSTx for 20 min each, incubated in PBSTi for 30 min, and transferred to rhodamine tyramide (1:1000 in PBSTi, 0.8 ml per well) for 30 min. Hydrogen peroxide (5 µl of 0.322% stock per 0.8 ml reaction, final concentration 0.002%) was added directly, and samples were incubated with shaking for 45 min. After seven additional 20-min washes in PBSTx, samples were stained with DAPI (1:2500 in PBSTx) and incubated overnight at 4 °C in the dark.

## Injections of dsRNA to planarians

Animals were injected with either *unc22* dsRNA (non-targeting control) or an equal mix of dsRNA targeting *ythdf-A*, *ythdf-B*, and *ythdf-C* using a Nanoject III (CAT 3-000-207, Drummond Scientific). Planarians were placed dorsal side up on wet filter paper and injected posterior to the pharyngeal cavity. Following the initial puncture, three consecutive injections of 33 nL each were delivered at a rate of 66 nL/s. Injections were performed six times at 3-day intervals. Animal size was measured four days after the final injection, and fixation was performed 5 days after the last injection.

## TUNEL labeling

TUNEL labeling of control and co-*ythdf* (RNAi) samples was performed after six dsRNA injections administered over three weeks, as large (fed) animals are otherwise incompletely labeled. TUNEL was performed as previously described (Pellettieri et al, 2010) with minor modifications. Animals were fixed in 5% NAC and bleached overnight in 6% hydrogen peroxide in PBSTx. Samples were then treated with ApopTag Red In Situ Apoptosis Detection Kit (S7165, Sigma-Aldrich). Groups of five animals per tube were incubated with the TdT enzyme mix for 4 h at 37 °C, followed by four washes in PBSTx. Samples were then blocked for 2 h in 0.5% Roche Western Blocking Reagent and 5% heat-inactivated horse serum in TNTx, followed by overnight incubation at 4 °C with anti-DIG antibody (1:1500). The next day, samples were washed seven times in PBSTx, and developed with rhodamine tyramide (1:1000). Samples were counterstained with DAPI (1:2500 in PBSTx) overnight at 4 °C, and mounted in Vectashield.

## Data availability

All sequencing data produced here, including GLORI libraries and RNAseq gene expression data, was uploaded to the Sequence Read Archive (SRA) (Leinonen et al, 2011) with BioProject accession PRJNA1226107.

The source data of this paper are collected in the following database record: biostudies:S-SCDT-10_1038-S44318-025-00662-3.

## Peer review information

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

## Acknowledgements

We thank Hila Kobo and Tamar Katzir for assistance with Illumina sequencing at the Tel Aviv University core facilities. We thank Rami Khosravi for assistance with scRNAseq library preparation at the Tel Aviv University single cell core unit. We thank Kristina Karin Mirkes for support in confocal microscopy imaging. We thank Prof. Joel Hirsch and Dr. Aldema Sas-Chen for

critical input on the manuscript. We thank the Wurtzel lab for critical input. OW is supported by the Israel Science Foundation (grant 2039/18) and the European Research Council (no. 853640). OW is a Zuckerman Faculty Scholar. AP is supported by the EIPOD-LinC Postdoc Fellowship from the European Molecular Biology Laboratory. HTKV is supported by the European Molecular Biology Laboratory.

## Author contributions

**Yarden Yesharim**: Conceptualization; Data curation; Formal analysis; Investigation; Visualization; Methodology; Writing—original draft; Writing—review and editing. **Ophir Shwarzbard**: Data curation; Software; Formal analysis; Investigation; Writing—original draft. **Jenny Barboy-Smoliarenko**: Formal analysis; Investigation; Visualization; Methodology. **Prakash Varkey Cherian**: Investigation. **Ran Shachar**: Software; Formal analysis; Investigation; Methodology. **Amrutha Palavalli**: Investigation; Visualization; Methodology. **Hanh Thi-Kim Vu**: Resources; Supervision; Investigation; Methodology; Writing—review and editing. **Schraga Schwartz**: Resources; Software; Formal analysis; Investigation; Writing—original draft; Writing—review and editing. **Omri Wurtzel**: Conceptualization; Resources; Data curation; Software; Formal analysis; Supervision; Funding acquisition; Validation; Investigation; Visualization; Methodology; Writing—original draft; Project administration; Writing—review and editing.

Source data underlying figure panels in this paper may have individual authorship assigned. Where available, figure panel/source data authorship is listed in the following database record: biostudies:S-SCDT-10_1038-S44318-025-00662-3.

## Disclosure and competing interests statement

The authors declare no competing interests.

# Expanded View Figures

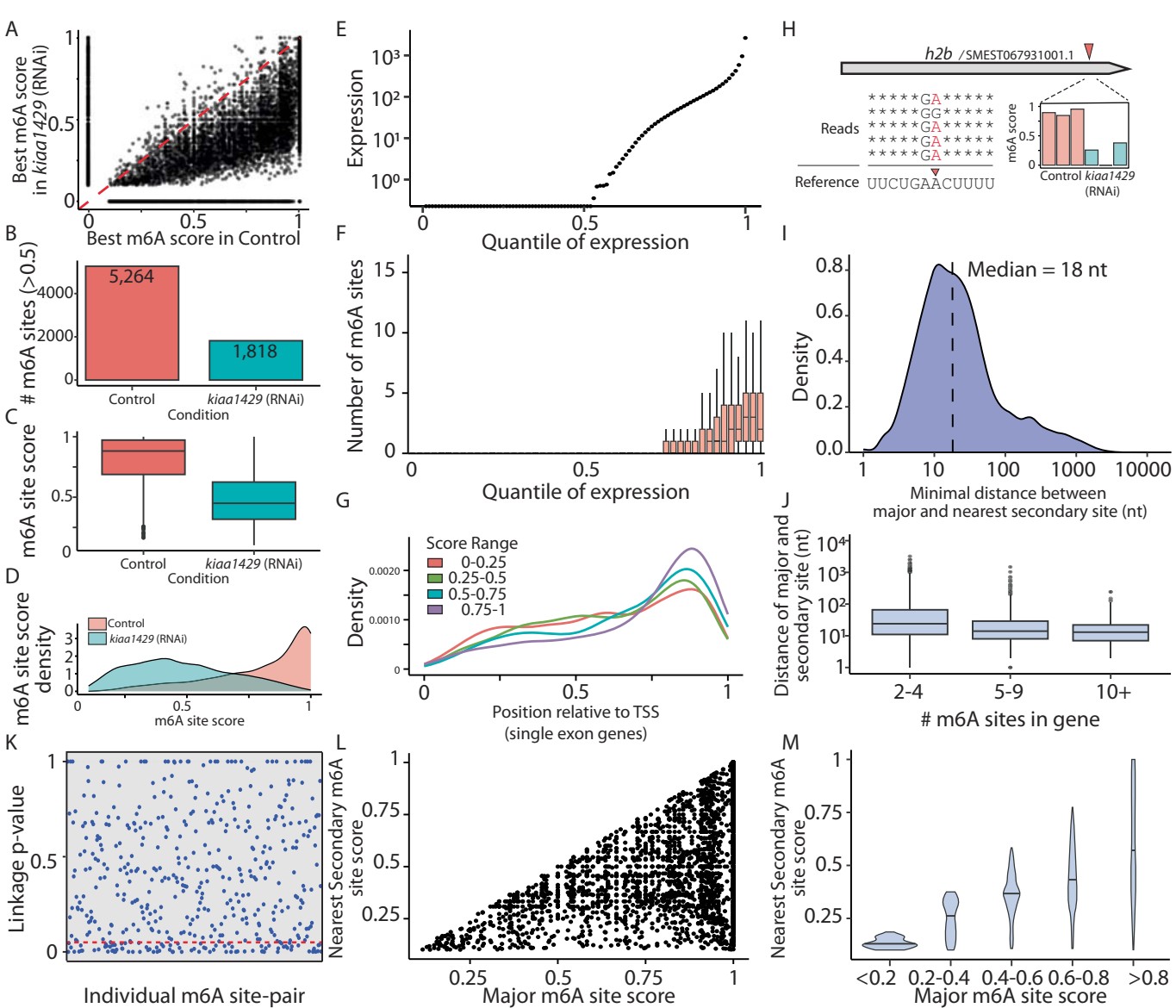

**Figure EV1. Properties of planarian m⁶A sites.**

(A) The scatter plot displays m⁶A site scores (black dots) in control samples, where m⁶A levels are normal, and in *kiaa1429* (RNAi) samples, where m⁶A levels are reduced. The red dashed line represents a slope of one. For each site, the score shown is the highest observed across any of the three replicates for each condition. (B) Number of m⁶A sites with median score greater than the 0.5 in control and *kiaa1429* (RNAi) libraries demonstrates the reduction in overall methylation in *kiaa1429* (RNAi). (C, D) Shown is the distribution of median scores of m⁶A sites in control and *kiaa1429* (RNAi) libraries as boxplot (C) and density (D) plots, for sites having coverage of at least 10 sequencing reads ($n = 2284$) in each of the three replicates used for the two conditions in this analysis. Boxplots show the median (center line), the IQR (box), and whiskers extending up to 1.5× the IQR range, with points beyond show outliers. (E) Shown is the median normalized expression (Love et al, 2014) of each gene in input libraries as a function of the quantile of expression. For example, the median expression of ~50% of the genes is 0. (F) The median number of m⁶A sites is shown (median score in control libraries >0.1) as a function of quantile of gene expression, which was calculated based on the expression level of the gene in the input GLORI (i.e., untreated) libraries. (G) Metagene analysis of m⁶A positions ($n = 2609$) of single gene exons shows a strong enrichment towards the 3′ end, indicating that 3′ end preference is not unlikely to be governed only by splicing specific factors. (H) Canonical histone transcripts are also targets for methylation. Shown is a canonical *h2b* (dd_2520) transcript with an m⁶A site (red arrow) detected near the 3′ end. Read sequences that cover the m⁶A site are shown (bottom left) with the methylated A highlighted in red. The m⁶A score of the site in different replicates indicates that the m⁶A installation is dependent on the MTC. (I) Shown is the minimal distance between the site having the highest m⁶A score (i.e., major) with a secondary site (i.e., m⁶A site having a lower score than the major site), in genes that have multiple m⁶A sites. (J) Shown is a comparison of distance between the major and secondary site for genes with multiple m⁶A sites. Boxplots show the median (center line), the IQR (box), and whiskers extending up to 1.5x the IQR range, with points beyond show outliers. Sizes of each category: 2–4 sites ($n = 3132$), 5–9 sites ($n = 1266$), and 10+ sites ($n = 379$). (K) The P value of linkage of m⁶A installation is shown for m⁶A site-pairs (Methods). For each site pair (blue dot) a P value was calculated (y axis). (L, M) Correlation of m⁶A score between the major m⁶A site and the nearest secondary site is shown.

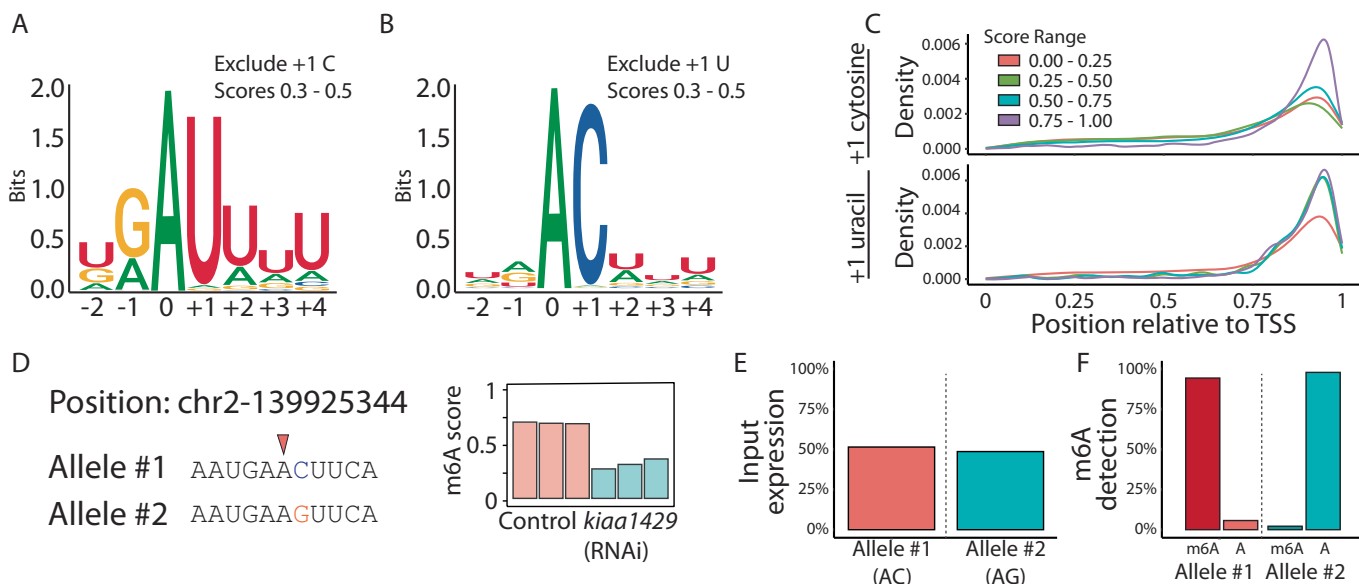

**Figure EV2. Sequence characteristics of m⁶A sites.**

(A) Shown is a sequence motif of m⁶A sites having low scores (0.3–0.5) and lacking a +1 C. The presence of a stretch of Us is observed as well as preference for a −1 G. (B) Shown is a sequence motif of m⁶A sites having low scores (0.3–0.5) and lacking a +1 U. There is no strong sequence preference for particular nucleotides other than a +1 C. (C) Metagene analysis of sites having either a +1 C (top) or +1 U (bottom) shows a similar distribution across the gene, indicating that the sites are installed similarly, despite the incompatibility of the +1 U with the known m⁶A installation consensus DRACH. (D–F) A rare example of an m⁶A site located at a sequence position that differs between alleles is shown. (D) The site (left, red arrow) is methylated in the allele with the AC sequence, whereas the other allele, which contains a +1 G, is refractory to m⁶A installation and remains unmethylated. The m⁶A site score (right) indicates methylation in the allele containing a +1 C. (E) The expression levels of both alleles were nearly identical, as determined by counting reads covering the sequence difference and distinguishing between them based on the nucleotide identity at the +1 position relative to the m⁶A site. (F) The fraction of methylated and unmethylated A is shown for both alleles, demonstrating that a +1 G is incompatible with m⁶A deposition.

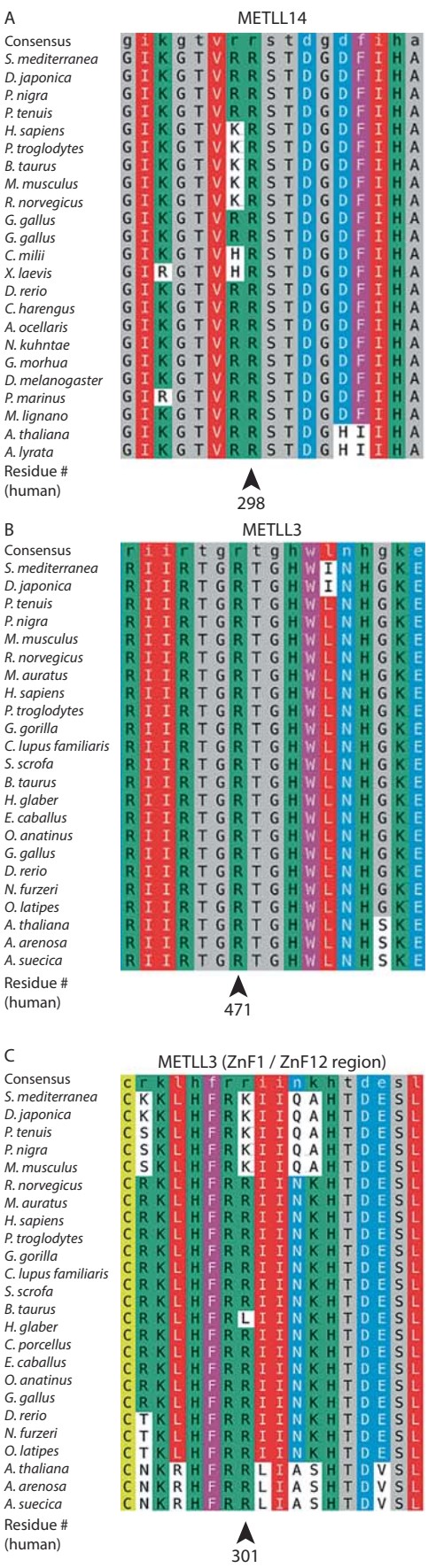

**Figure EV3. Multiple alignment of key RNA recognition regions in the MTA.**

(A–C) Alignments ("Methods") include multiple planarian species, diverse metazoans, and representative *Arabidopsis* orthologs; residue numbering follows the human proteins. (A, B) The residues annotated, human cancer-related, change the RNA sequence binding preference of the human MTC from GGAC to GGAU (Zhang et al, 2024; Qi et al, 2024). (C) METTL3 ZnF1/2 region. The arrow marks the position homologous to human R301, a residue on the RNA-interaction surface of the MTC required for efficient MTC activity (Huang et al, 2019). In planarians, this position is substituted by lysine, preserving charge. We speculate that this change might have contributed to altered binding of the RNA residue.

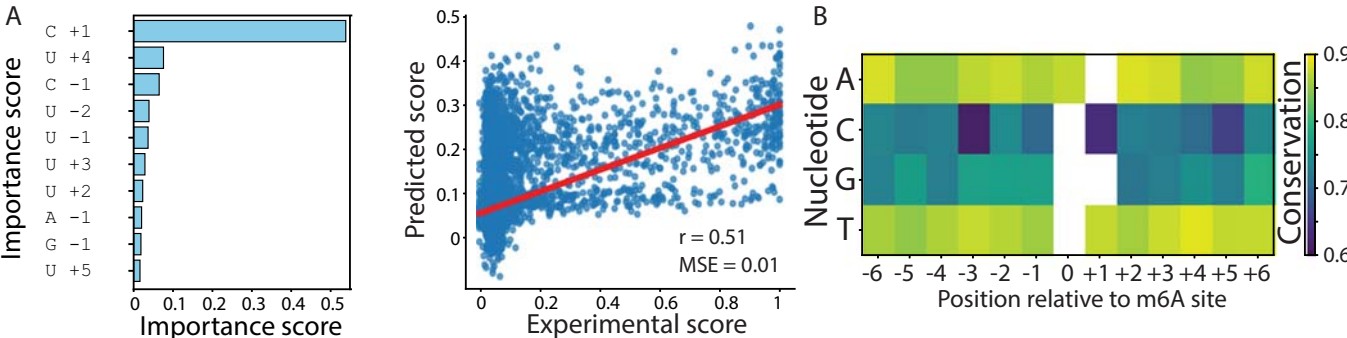

**Figure EV4. Sequence characteristics of m⁶A sites.**

(A) The importance scores of sequence features near the m⁶A site are shown. The strongest determinant of m⁶A installation is the presence of a +1 C, while a + 4U is also associated with high methylation potential (left). A model incorporating only local sequence features performed better at identifying sequences that were poor methylation targets than at predicting methylation-compatible sequences (right), suggesting that factors beyond local sequence identity influence methylation levels. (B) Sequence conservation was calculated for individual m⁶A sites between *Schmidtea mediterranea* and *Dugesia japonica*. The comparison revealed similar conservation patterns across sites when considering the nucleotide identity near the m⁶A site. Overall, C and G were less conserved than A and T.

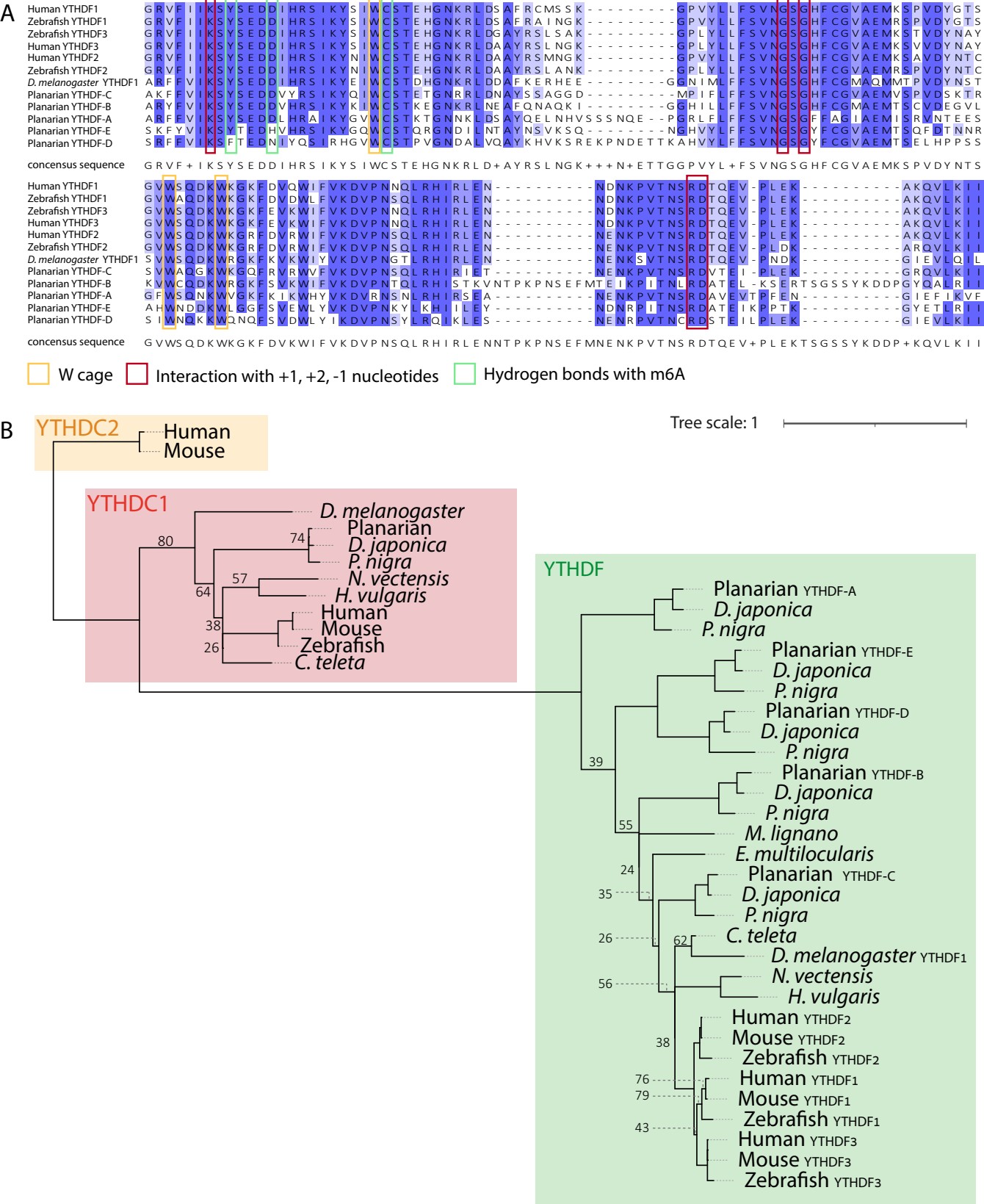

◀  **Figure EV5.  phylogenetic analysis of planarian YTHDF proteins.**

(**A**) Multiple sequence alignment of the conserved YTH-domain from YTHDF proteins in planarians, zebrafish, and *Drosophila*. Darker colors represent higher levels of amino acid conservation. (**B**) Phylogenetic tree illustrating the evolutionary relationships among various YTH-domain proteins across different species (Dataset EV3). Numbers indicate bootstrap values (1–100); bootstrap values > 85 were omitted for visual clarity. Planarian: *Schmidtea mediterranea*; *Spol*: *Schmidtea polychroa*; *D. japonica*: *Dugesia japonica*; *Pnig*: *polycelis nigra*; *C. teleta*: *Capitella teleta*; *M. lignano*: *Macrostomum lignano*; *Echinococcus multilocularis*; *H. vulgaris*: *Hydra vulgaris*; *N. vectensis*: *Nematostella vectensis*; *D. melanogaster*: *Drosophila melanogaster*.

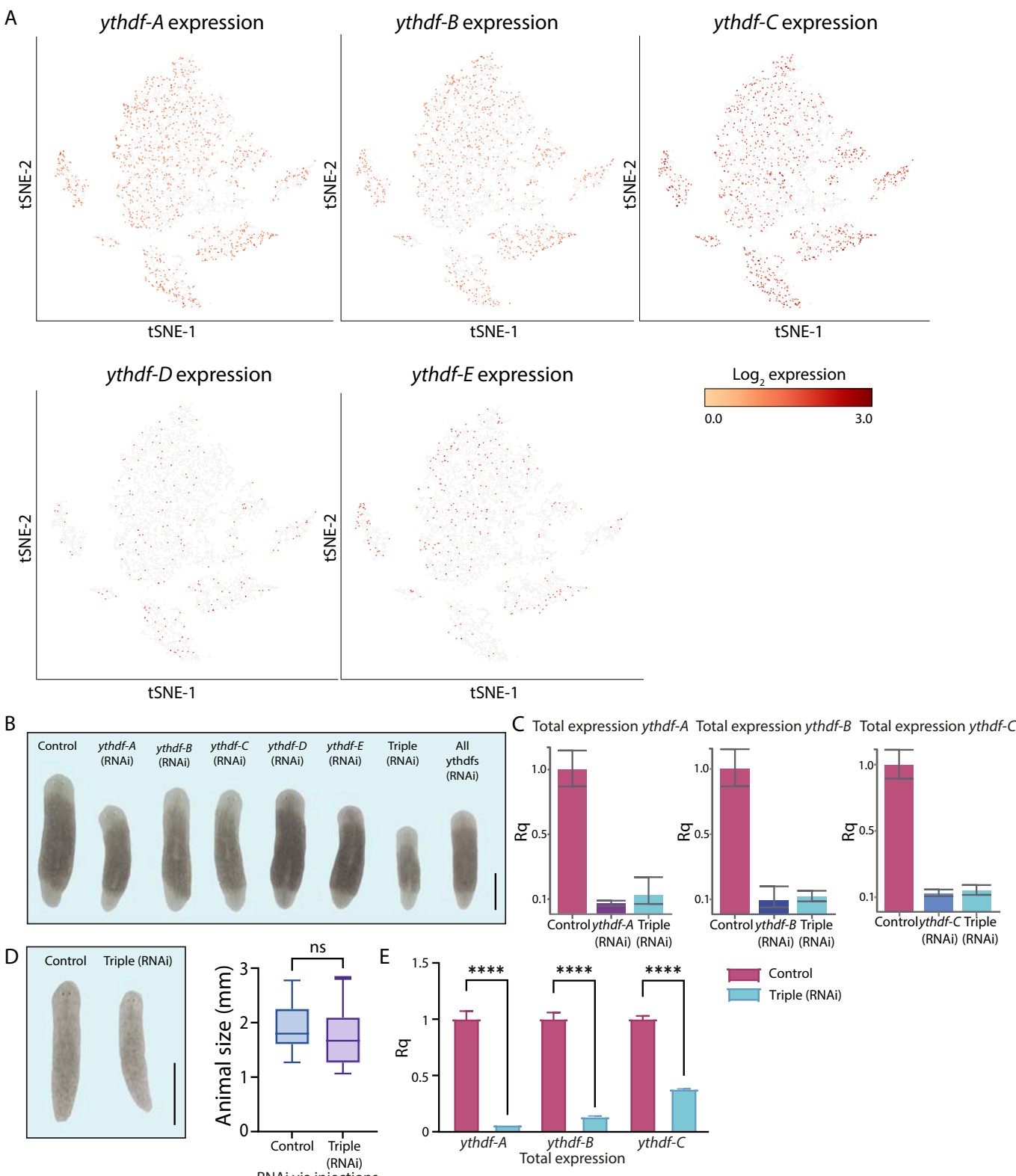

◄ **Figure EV6.** *ythdf* **expression is not required for planarian regeneration.**

(A) *ythdf*-encoding genes exhibit broad expression across planarian cells based on scRNAseq data. Shown here is the expression of the five *ythdf* genes, extracted from a published dataset of G0 planarian cells (King et al, 2024). (B) *ythdf* (RNAi) animals successfully regenerated heads and tails following amputation (15/15). Scale bar = 1 mm. (C) qPCR analysis showing significant downregulation of *ythdf-A*, *ythdf-B*, and *ythdf-C* gene expression following RNAi in both single and triple inhibition conditions. Error bars indicate the 95% confidence interval. Data include two technical replicates and three biological replicates. *P* value for each condition compared to its control was as high as 0.0078. (D) Comparison of animal sizes following three weeks of co-*ythdf* dsRNA injections compared to control animals. Shown are representative images (left) and quantification. There was no significant difference in the animal sizes (Student's two-tailed *t* test; *P* value = 0.511; group size *n* = 18; Boxes represent the IQR, whiskers represent min to max, and central band represents the median) ("Methods"). Scale = 1 mm. (E) Shown is qPCR validation that the co-*ythdf* (RNAi) were efficiently silenced by the microinjections. **** Student's *t* test *P* value < 0.0001. Data includes three biological replicates, each measured by two technical replicates, error bars represent SD and the center is the mean.

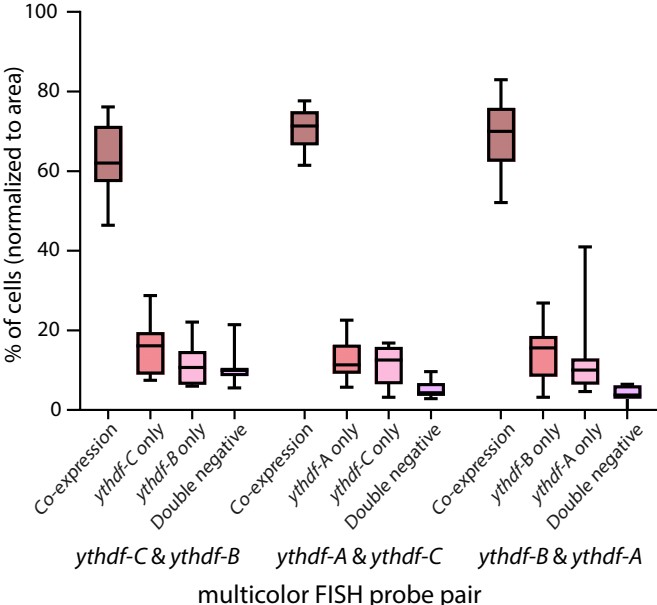

**Figure EV7. Quantification of *ythdf* expression in the planarian epidermis.**

Boxplots representing the percentage of epidermal cells expressing *ythdf* genes. Boxes represent the IQR, whiskers represent min to max, and central band represents the median. Multicolor fluorescence in situ hybridization (FISH) analysis was used to categorize cells into four groups: double-negative cells, single-positive cells (expressing only one of the two analyzed *ythdf*), and double-positive cells (co-expressing the two analyzed *ythdf* genes). Data were collected from 10 distinct epidermal regions from the top of the pharynx to the brain and normalized to the area of each region ("Methods").

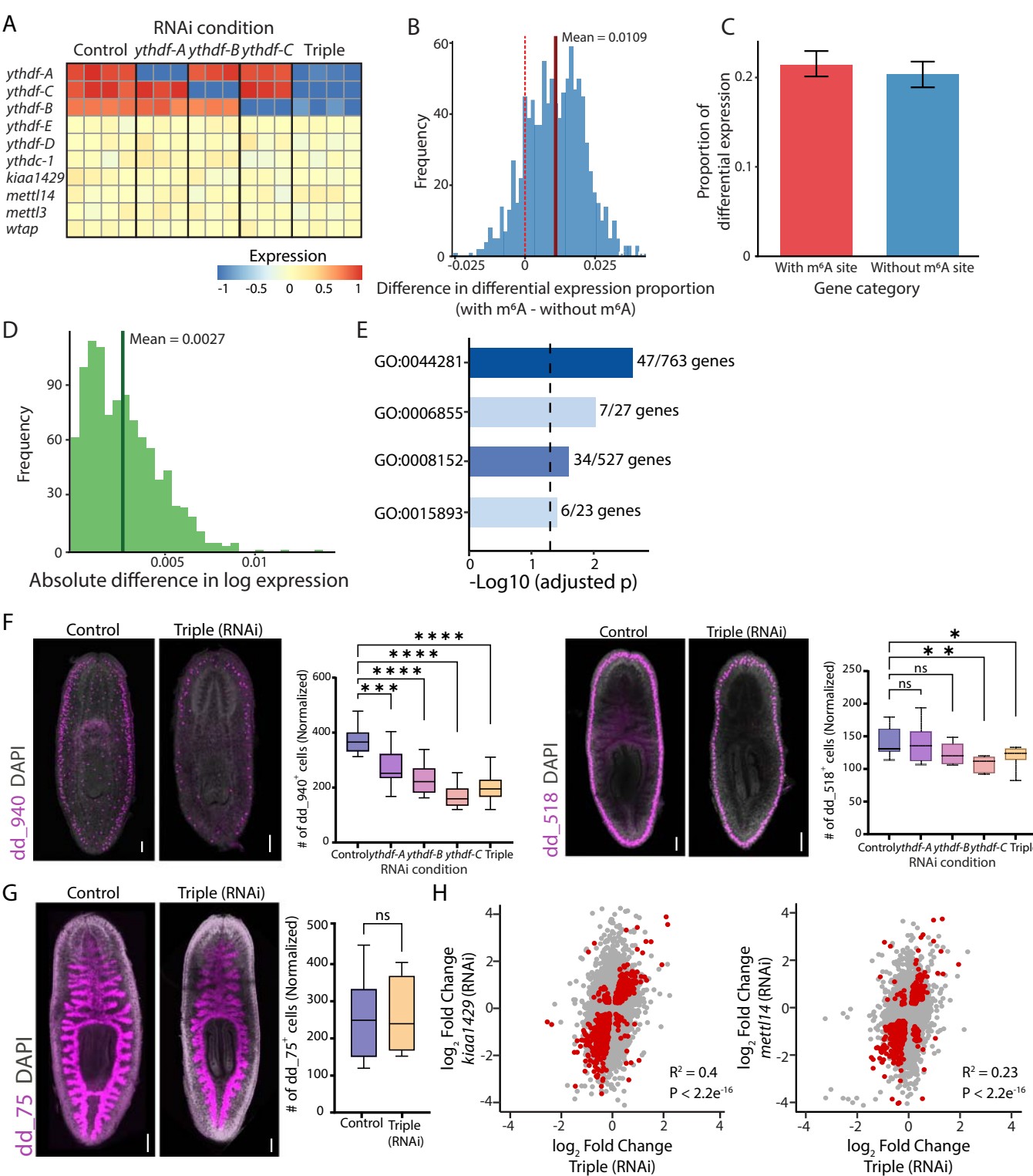

**Figure EV8.  Inhibition of *ythdf* genes resulted in a reduction in parenchymal cells.**

(A) Heatmap of YTH-domain protein encoding genes and MTC encoding gene expression following inhibition of *ythdf* genes. Inhibition of *ythdf* genes did not affect the expression of other components of the m$^6$A pathway. Displayed are z-scores ranging from −1 to 1 (FDR < 1 × 10$^{-5}$). Each column represents a biological replicate.
(B) Distribution of the difference in the proportion of differentially expressed genes between expression-matched sets with versus without mapped m$^6$A sites across 1000 bootstrap iterations. The mean difference was calculated (red thick line). The red dashed line represents no difference. Empirical two-sided *P* value = 0.302, indicating no enrichment in the number of differentially expressed genes among m6A-containing genes after co-*ythdf* suppression. (C) Mean fraction of differentially expressed genes (adjusted *P* < 0.05) for genes having m$^6$A versus genes without m$^6$A sites across the same bootstraps; error bars show 95% confidence intervals. (D) Distribution of the absolute difference in baseline log expression between the matched gene sets (mean = 0.0027), confirming close expression matching. Lower value indicates a greater similarity in gene expression between the groups. (E) Shown are significantly enriched biological processes (gene ontology categories) in differentially expressed genes following the co-suppression of *ythdf*s ("Methods"). (F) FISH analysis showing changes in different parenchymal cell types (Fincher et al, 2018) following inhibition of *ythdf* genes using the markers dd_940 (top right) and dd_518 (bottom left). Cell counts were normalized to animal size ("Methods") in *ythdf-A* (RNAi), *ythdf-B* (RNAi), *ythdf-C* (RNAi), and triple (RNAi) animals and compared to control animals. Statistical significance was assessed using one-way ANOVA followed by Dunnett's correction; *P* values: dd_940 *ythdf-A* (RNAi) *P* = 0.0007, *ythdf-B* (RNAi) *P* = 5.3 × 10$^{-6}$, *ythdf-C* (RNAi) *P* = 3.0 × 10$^{-9}$, triple (RNAi) *P* = 1.85 × 10$^{-8}$; dd_518 *ythdf-C* (RNAi) *P* = 0.0039, triple (RNAi) *P* = 0.0451; group size *n* > 8, Boxes represent the IQR, whiskers represent min to max, and central band represents the median. Scale bar = 100 μm. (G) FISH analysis detecting intestinal cells expressing dd_75, in triple (RNAi) and control animals. Cell count of dd_75$^+$ cells in the intestinal region spanning the pharynx was normalized to animal size, revealing no reduction in intestine cell number following *ythdf*s inhibition (Student's two-tailed *t* test, *P* > 0.05, group size *n* > 8). Scale bar = 100 μm. (H) Correlation of gene expression changes between *kiaa1429* (RNAi) and triple (RNAi) compared to their controls (left), and *mettl14* (RNAi) and Triple (RNAi) compared to their controls (right). Each colored dot represents a gene, red and black represents significant (FDR < 0.05 for both conditions) and nonsignificant change in gene expression compared to controls, respectively.

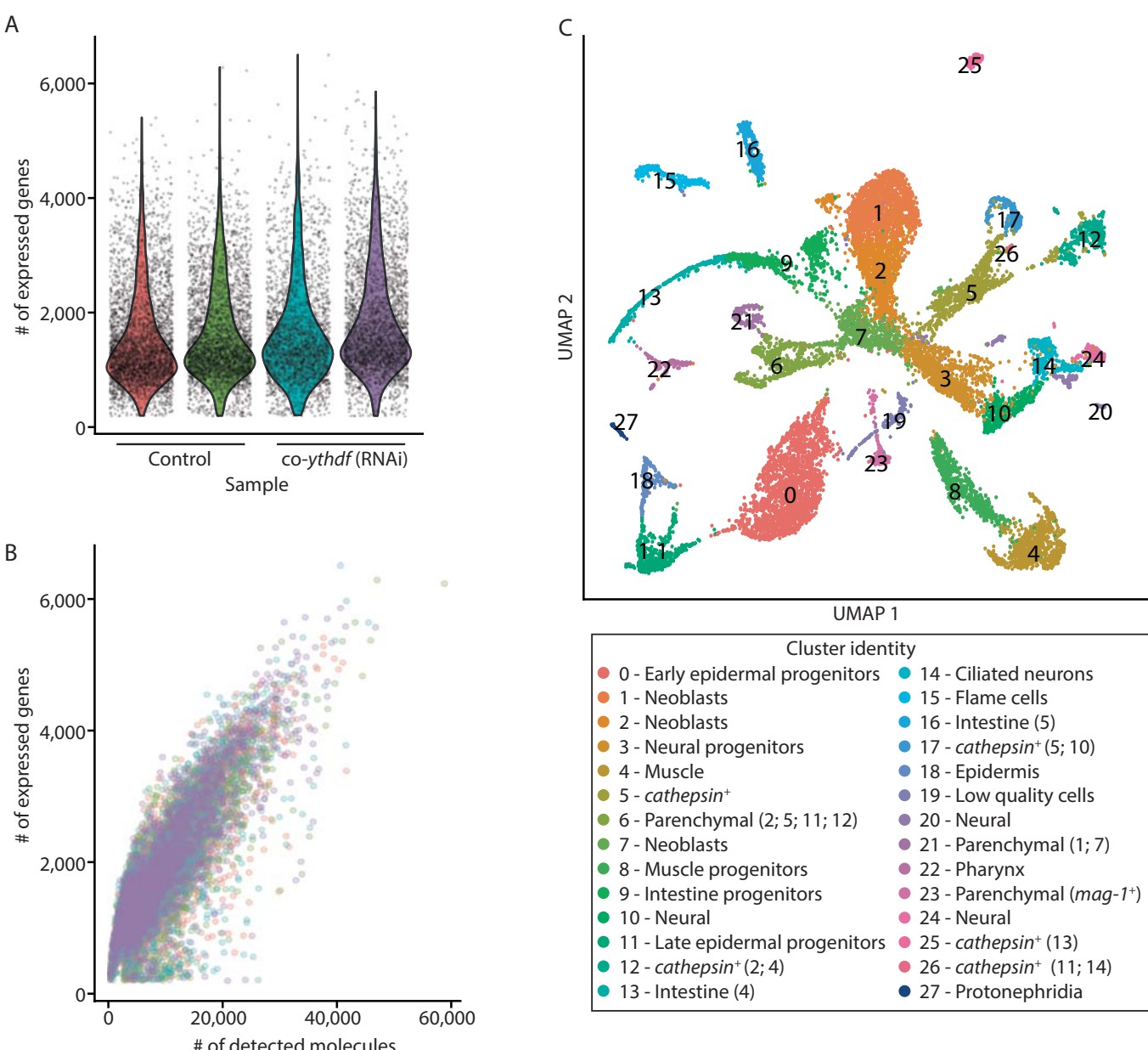

**Figure EV9. scRNAseq quality and clustering after co-*ythdf* suppression.**

(A) Unique transcripts detected per cell shown in two biological replicates of control and co-*ythdf* (RNAi) samples, showing comparable library complexity. Dots represent cells. (B) Genes detected as a function of detected molecules (UMIs) per cell, indicating similar saturation across conditions Dots represent cells. (C) UMAP of cells from the four sequenced samples, with clusters (0–27) annotated based on expression of gene markers ("Methods"). Numbers in parenthesis indicate specific sub-cluster identity based on the Planarian Cell Atlas annotation (Fincher et al, 2018).

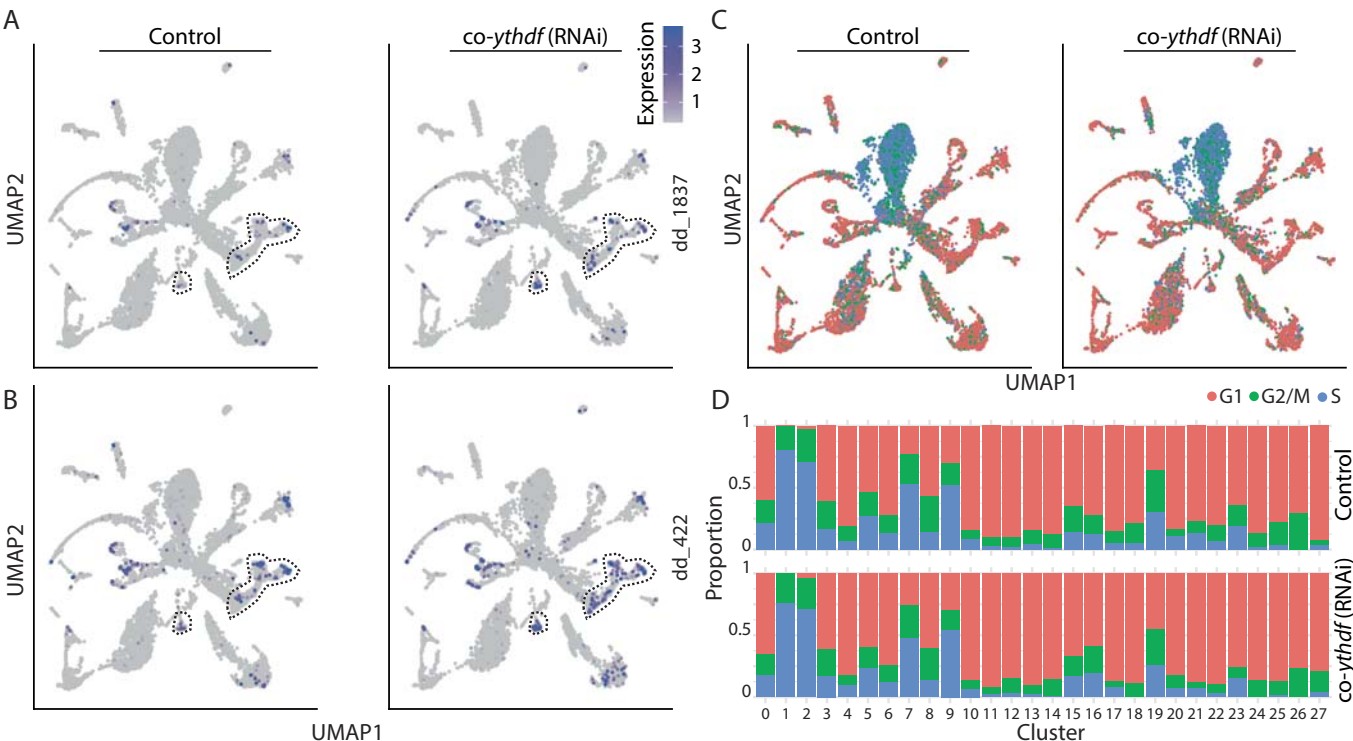

**Figure EV10. scRNAseq gene expression and cell cycle analysis.**

(**A, B**) UMAP plots showing the expression of dd_1837 (**A**) and dd_422 (**B**) in Control and co-*ythdf* (RNAi) conditions. Dashed outlines mark clusters with strongest induction. Gray to purple, low to high log-normalized expression, respectively. (**C**) UMAP plots colored by inferred cell-cycle phase (G1, red; G2/M, green; and S, blue) for Control and co-*ythdf* (RNAi) conditions ("Methods"). (**D**) Proportion of cells in G1, G2/M, and S phase cells per cluster in Control (top) and co-*ythdf* (RNAi) (bottom), showing broadly similar cell-cycle composition across conditions.

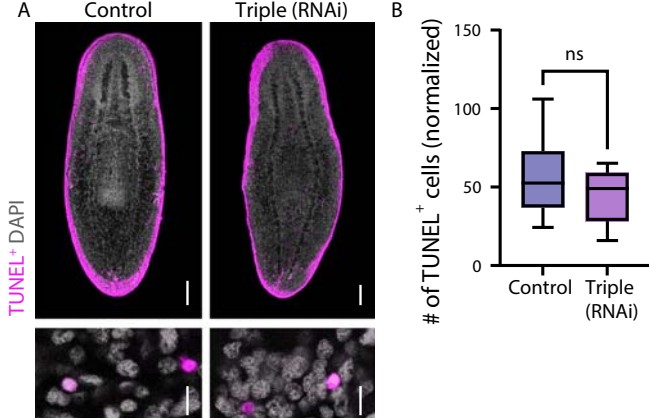

**Figure EV11.  Analysis of cell death by whole-mount TUNEL.**

(A) TUNEL analysis of apoptotic cells in control and co-*ythdf* (RNAi) animals following six dsRNA injections ("Methods"). scale bar = 100 µm. (B) Counts of TUNEL⁺ cells across the entire body were normalized to animal size, revealing no significant difference in the number of apoptotic cells following co-*ythdf* inhibition (right, Student's two-tailed *t* test *P* > 0.05; group size *n* > 10).

