## [Peer Review File · The EMBO Journal]

Single-nucleotide m⁶A mapping uncovers redundant YTHDF function in planarian progenitor fate selection

Yarden Yesharim, Ophir Shwarzbard, Jenny Barboy-Smoliarenko, Prakash Cherian, Ran Shachar, Amrutha Palavalli, Hanh Thi-Kim Vu, Schraga Schwartz, and Omri Wurtzel

Corresponding author: Omri Wurtzel (owurtzel@tauex.tau.ac.il)

Review Timeline:

Submission Date:	3rd Mar 25
Editorial Decision:	5th May 25
Revision Received:	2nd Sep 25
Editorial Decision:	23rd Oct 25
Revision Received:	13th Nov 25
Accepted:	28th Nov 25

Editor: Daniel Klimmeck

Transaction Report:

Dear Omri,

Thank you again for the submission of your manuscript (EMBOJ-2025-120652) to The EMBO Journal, and providing us with a preliminary point-by-point response to the concerns raised by the referees. Please accept my apologies for getting back to you with unusual delay due to protracted referee input as well as detailed discussion in the editorial team. Your study was assessed by three reviewers with expertise in stem cells, planarian tissue biology and epitranscriptomics, whose comments are enclosed below.

As you will see from their comments, the referees acknowledge the potential interest and value of your findings. However, they also express important issues regarding the completeness of your study of the results, which need to be addressed thoroughly to make them supportive of publication in the EMBO Journal. In more detail, referee #3 expresses major concerns regarding precedence over related literature and depth of insights provided into GAU methylation patterns and the details of cross-compensation of m6A-YTHDF function in the planarian context (ref#3, standfirst, pts.1-4). Reviewer #1 also states that depth of characterisation of the planarian tissue phenotypes is underdeveloped currently (ref#1, pts. 1-3). Further, the reviewers raise a number of issues related to the presentation of the findings, additional controls and improved methods annotation required, statistics applied and overall discussion of related literature, that would need to be conclusively addressed to achieve the level of robustness and clarity needed for The EMBO Journal.

Given the overall interest stated and broader angle of your results, we are able to invite you to revise your manuscript experimentally to address the referees' comments, along the lines sketched in your outline. I need to stress though that we do require strong support from the referees on a revised version of the study in order to move on to publication of the work.

Please feel free to contact me if you have any questions or need further input on the referee comments.

When submitting your revised manuscript, please carefully review the instructions below.

Please feel free to approach me any time should you have additional questions related to this.

Thank you for the opportunity to consider your work for publication.

I look forward to your revision.

Best regards,

Daniel

Daniel Klimmeck, PhD
Senior Editor
The EMBO Journal

Instruction for the preparation of your revised manuscript:

- 1) a .docx formatted version of the manuscript text (including legends for main figures, EV figures and tables). Please make sure that the changes are highlighted to be clearly visible.
- 2) individual production quality figure files as .eps, .tif, .jpg (one file per figure).
- 3) a .docx formatted letter INCLUDING the reviewers' reports and your detailed point-by-point response to their comments. As part of the EMBO Press transparent editorial process, the point-by-point response is part of the Review Process File (RPF), which will be published alongside your paper.

4) a complete author checklist, which you can download from our author guidelines ([https://wol-prod-cdn.literatumonline.com/pb-assets/embo-site/Author Checklist%20-%20EMBO%20J-1561436015657.xlsx](https://wol-prod-cdn.literatumonline.com/pb-assets/embo-site/Author%20Checklist%20-%20EMBO%20J-1561436015657.xlsx)). Please insert information in the checklist that is also reflected in the manuscript. The completed author checklist will also be part of the RPF.

6) It is mandatory to include a 'Data Availability' section after the Materials and Methods. Before submitting your revision, primary datasets produced in this study need to be deposited in an appropriate public database, and the accession numbers and database listed under 'Data Availability'. Please remember to provide a reviewer password if the datasets are not yet public (see <https://www.embopress.org/page/journal/14602075/authorguide#datadeposition>).

7) Our journal encourages inclusion of *data citations in the reference list* to directly cite datasets that were re-used and obtained from public databases. Data citations in the article text are distinct from normal bibliographical citations and should directly link to the database records from which the data can be accessed. In the main text, data citations are formatted as follows: "Data ref: Smith et al, 2001" or "Data ref: NCBI Sequence Read Archive PRJNA342805, 2017". In the Reference list, data citations must be labeled with "[DATASET]". A data reference must provide the database name, accession number/identifiers and a resolvable link to the landing page from which the data can be accessed at the end of the reference. Further instructions are available at .

8) At EMBO Press we ask authors to provide source data for the main and EV figures. Our source data coordinator will contact you to discuss which figure panels we would need source data for and will also provide you with helpful tips on how to upload and organize the files.

Numerical data can be provided as individual .xls or .csv files (including a tab describing the data). For 'blots' or microscopy, uncropped images should be submitted (using a zip archive or a single pdf per main figure if multiple images need to be supplied for one panel). Additional information on source data and instruction on how to label the files are available at .

9) We replaced Supplementary Information with Expanded View (EV) Figures and Tables that are collapsible/expandable online (see examples in <https://www.embopress.org/doi/10.15252/emboj.201695874>). A maximum of 5 EV Figures can be typeset. EV Figures should be cited as 'Figure EV1, Figure EV2' etc. in the text and their respective legends should be included in the main text after the legends of regular figures.

11) For data quantification: please specify the name of the statistical test used to generate error bars and P values, the number (n) of independent experiments (specify technical or biological replicates) underlying each data point and the test used to calculate p-values in each figure legend. The figure legends should contain a basic description of n, P and the test applied. Graphs must include a description of the bars and the error bars (s.d., s.e.m.).

Please remember: Digital image enhancement is acceptable practice, as long as it accurately represents the original data and conforms to community standards. If a figure has been subjected to significant electronic manipulation, this must be noted in the

figure legend or in the 'Materials and Methods' section. The editors reserve the right to request original versions of figures and the original images that were used to assemble the figure.

We realize that it is difficult to revise to a specific deadline. In the interest of protecting the conceptual advance provided by the work, we recommend a revision within 3 months (3rd Aug 2025). Please discuss the revision progress ahead of this time with the editor if you require more time to complete the revisions.

Referee #1:

In the manuscript 'Single-nucleotide m6A mapping uncovers redundant YTHDF function in planarian progenitor fate selection,' Yesharim and colleagues present single nucleotide resolution mapping of m6A sites that expands our understanding of the simple sequence rules that guide m6A deposition on planarian RNAs. They also perform single and group knockdown of YTHDF family m6A readers to determine how m6A readers contribute to planarian biology. Surprisingly, individual YTHDF family members have overlapping, but unique expression in planaria, and only cause observable phenotypes when depleted as a group, which includes a smaller size animal size and altered cellular composition. Bulk sequencing of the triple knockdown worms revealed changes in tissue proportion and progenitor numbers, including a decrease in some secretory and phagocytic cell types and expansion of a previously identified abnormal neural-like progenitor state (Kiaa1429 specific cell population). The data is very interesting, and the single nucleotide resolution methods provide new mechanistic insight. However, the current form of the paper does not seem to significantly advance our understanding of m6a function in planarian tissue homeostasis and/or physiology (regeneration, tissue turnover, etc.). Most of the phenotypes they report seem very similar to the phenotype previously observed for kiaa1429 depletion, if less severe. However, I think the authors could address this concern with a more detailed characterization of the triple knockdown phenotype. I've detailed my thoughts on this below:

Major Criticisms:

- The size reduction phenotype with the YTHDF triple knockdown is very interesting, but the mechanisms driving this phenotype are not deeply explored in the manuscript. Do the animals fission more to produce smaller animals? Is stem cell proliferation reduced? Is apoptosis increased? Since the animals are still capable of regeneration, it seems like an important insight to understand how m6A modifications contribute to the regulation of tissue turnover and stem cell fate determination
- The observation that several adult tissues are depleted and an abnormal progenitor population expands in the triple knockdown is convincing. However, I don't think the authors investigate stem cell differentiation rates in sufficient depth or breadth to make definitive mechanistic insights. Is decreased differentiation across all lineages? Are rates of stem cell symmetric vs. asymmetric divisions altered to increase stem cell numbers while decreased progenitors? Or, are the lineage progenitors specified at similar rates, but dying or being cleared at higher rates to alter final animal composition. These questions should be possible to answer using methods available in planaria. Determining which might be true will provide insight into the function of the m6A modification in planarian cell type specification.
- The dramatic accumulation of the kiaa1429-specific cells is very interesting, and suggests that this cell state accumulates when m6a modifications are not properly regulated. But, what does the cell state do? What genes are expressed in this cell type? Do any of them imply a cellular function that might explain the decreased size and altered cellular composition of the worms? The distinct spatial localization of the cell type suggests it may not be a random abnormal progenitor. Connecting the function or existence of this cell type to the other phenotypes observed would deepen the insight of the paper and the impact of its' findings.

Minor Criticisms:

- It would be more informative if the authors used box plots with data points to allow interpretations of animal number and replicates
- The authors state that cellular quantitation of microscopy images were performed manually, so were the analysts blinded to the conditions they were quantitating?

Referee #2:

This study resolves m6A sites in planarians at single-nucleotide positions across the transcriptome and comprehensively determines roles for the ythdc class of m6A reader proteins. The authors find ythdc factors have overlapping but tissue-specific expression, evoking a possible code of m6A regulation important in cell-type specific gene regulation. Surprisingly, knockdown of a combination of three different ythdf factors led to a novel phenotype of reduced animal size without loss of tissue homeostatic integrity. This is particularly interesting because most perturbations of planarian growth described to date cause a wholesale failure of stem-cell-dependent tissue maintenance. A cross-species analysis of m6A site conservation surprisingly

indicates widespread rapid evolutionary changes to m6A sites. Analysis of genes misregulated after ythdc RNAi identified several classes of cell-type specific markers that m6A detection is important for specifying the abundance of several classes of tissue-specific progenitors, particularly in the parenchymal compartment. Therefore, a combination of ythdc factors exerts control over cell type specific gene regulation relevant for distinct tissue renewal programs.

The work is clear and compelling, and it makes a novel contribution to understanding both gene regulation and genomics in planarians, as well as the general properties of m6A. As such the work would be very interesting to a broad audience. I believe the work is already comprehensively conducted and does not require additional experiments or analysis to make the key points. I have some minor suggestions or ideas for future work below.

Minor comments:

P9 clarify what is meant by "nuclear ythdc-1", do authors mean that they are testing only the nuclear component of YTHDC-1 activity or that they predict YTHDC to be mainly a nuclear protein, or some other interpretation?

The co-expression analysis is a creative application of the single-cell data and interesting to see that individual cell types are enriched by different combinations of the ythdc homologs. I wonder, could mRNAs expressed uniquely in different cell types provide a way to test whether distinct ythdc complexes might have a preference for different target m6A sites? Or would the authors predict that ythdc factors all have the same specificity? If there were differential specificity I wonder if this could help to explain the diverse tissue-specific effects observed after inhibition of these genes.

The size control phenotype is quite interesting. Could this be due to an inability to grow upon feeding or instead be a result of faster turnover? It may be beyond the scope of this report, but would be interesting to know whether the triple ythdc-A;-B;-C RNAi delivered without food would cause an acceleration of degrowth. Could any of the target mRNAs misregulated after triple-gene ythdc RNAi potentially help explain the failed growth phenotype?

The size reduction phenotype is very clear. I wonder if a jittered scatterplot overlaying the box plots could even more strongly show the weight of evidence (in terms of n) that Figure 3D depicts?

Relatedly and as a point perhaps for future work to understand the developmental mechanisms of ythdc factors, the failed growth phenotype is indeed rather rare in the planarian literature I believe. Miller and Newmark reported a similar phenotype after inhibition of an insulin-like peptide or an insulin receptor. Perhaps that gene is targeted by m6A and the ythdc-a/-b/-c factors or could signal upstream of downstream m6A deposition/detection? (doi: 10.1387/ijdb.113443cm). To clarify, I think the current manuscript is fully complete and do not think this needs to be explored in this paper but could be an interesting idea for future research.

The work identifies and characterizes several novel progenitor populations regulated by m6A. Are any previously identified planarian progenitors (eg eyes, pharynx, epidermis, etc) also identified through the RNAseq analysis? It is interesting to consider how/why m6A may be critical for some lineages but not others.

Referee #3:

The study by Yesharim et al. uses flat worms to study the regulatory system relying on N6-methyladenosine (m6A) in mRNA and m6A-binding proteins in the YTHDF phylogenetic clade. The eukaryotic systems with the most advanced knowledge on molecular mechanisms and biological functions of this very important regulatory system are currently vertebrates (with important contributions mainly from human cell culture, mouse and zebrafish) and the plant *Arabidopsis thaliana*. Investigations on m6A-YTHDF in flat worms are, therefore, a very welcome addition to the objective of understanding eukaryotic m6A function, not least because considerable controversy on the matter exists in the mammalian literature in particular, but even the plant literature is not completely spared for this problem either.

The current study rests on the previous finding that the m6A methyl transferase in flatworms is essential because knockdown of its subunits causes intestinal malfunction and inability to take up food, a phenotype explained largely by defective function of the nuclear m6A-binding protein YTHDC (Dagan et al., 2022). This study follows up by addressing two key questions:

- (1) Which transcripts and which sequence contexts are methylated in flat worms, as addressed by single-nucleotide resolution transcriptome-wide m6A mapping?
- (2) What are the biological functions planarian YTHDF proteins, and do they have specialised or redundant functions?

To address the first question, the authors successfully implement the single-nucleotide resolution GLORI-seq technique. These results are high-quality and provide a very useful basis also for future studies. The most interesting finding is that m6A in

planarians does not only occur in the canonical DRACH context, but also in GAU-type motifs, often with a U at the +4 position. This suggests that the methyl transferase itself has been shaped a bit differently by evolution in planarians compared to mammals. However, very similar results, in particular methylation of (G)GAU motifs and enrichment of pyrimidines just downstream of methylated As (often ACUCU) have been reported for the plants *Arabidopsis thaliana* and *Oryza sativa* (Arribas-Hernández et al., 2021a; Wang et al., 2024), so it is not a genuinely new observation that the methyltransferase can evolve to methylate this type of motif in addition to the canonical DRACH motif. What would be expected for a high-impact publication now would be identification of the methyltransferase properties that allow GAU methylation in addition to DRACH methylation. At the very least, an analysis of alignments of methyl transferase subunit sequences with including planarian/plant and mammalian sequences would be expected to see if it is possible to pinpoint candidate residues implicated in this important expansion of target sequences.

To address the second question, a variety of approaches are used to study the five planarian YTHDF proteins. The most important observation is that only combined knockdown of the three most highly expressed YTHDFs produce phenotypes and that these YTHDFs are largely co-expressed, suggesting (but not proving, as the authors correctly and carefully note) redundant molecular function. The study also includes valuable insight into what the cellular defects of the combined YTHDF knockdowns are, including clear demonstration that the intestinal defects seen in methyltransferase and YTHDC knockdowns are not detected in the YTHDF knockdowns.

The study is well executed and adds important new insights into the m6A-YTHDF system in planarians, but it does not reach the level where concrete, genuinely new insights into eukaryotic m6A-YTHDF function are obtained. Points of criticism to address in a revised version of the manuscript are listed below:

1. In its present form, the manuscript does not frame the study correctly within the current understanding of eukaryotic m6A function. Nearly all of the plant literature is entirely overlooked which for this study in particular is quite serious for the following reasons:

- (i) The GAU context of m6A methylation has already been described in plants (Arribas-Hernández et al., 2021a; Wang et al., 2024).
- (ii) The matter of YTHDF redundancy/specialization is better described in *Arabidopsis* than in any other eukaryotic system.
 - (A) It is the first system where clear genetic redundancy between divergent YTHDFs was described (Arribas-Hernández et al., 2018; Arribas-Hernandez et al., 2020).
 - (B) Evidence for co-expression, target overlap, and in vivo competition for target binding between genetically redundant YTHDFs has been obtained (Arribas-Hernández et al., 2021b).
 - (C) Clear evidence for redundant molecular function of 8/11 YTHDFs, but specialized functions of 3/11 YTHDFs has been obtained (Flores-Téllez et al., 2023), with some understanding of molecular elements needed for redundant functions (Flores-Téllez et al., 2023; Tankmar et al., 2023).Hence, thorough mention of that knowledge is mandatory when the question of YTHDF specialization is brought up, and comparisons to that frame of reference is appropriate in the discussion.

2. A phylogenetic analysis of YTHDFs, including the five planarian YTHDFs, is of course mandatory for this study. But the analysis shown in Figure 3B is too simplistic and superfluous. It is relevant to know how many clades these YTHDF proteins represent, how different they really are from more studied YTHDFs in other organisms (at least vertebrate and fly YTHDFs), whether they are planarian peculiarities, or found more widely in metazoans? For inspiration on how such analyses can be done, Figure 1 in (Flores-Téllez et al., 2023) can be consulted.

3. The mRNA-seq data in YTHDF knockdowns are not sufficiently exploited. We should know

- (i) whether differentially expressed genes tend to be m6A targets or not to have an idea of whether expression differences could be a direct consequence of YTHDF knockdown or not
- (ii) not only where differentially expressed genes are expressed, but also whether they encode proteins acting in particular pathways or with shared molecular function - a gene ontology-type analysis, as simple as it is, is still informative.

4. It is a major weakness of the paper that it does not contain information on transcripts bound by the YTHDFs (at least YTHDF A/B/C). The authors state in the discussion that antibodies are not available for these proteins, but this is hardly a good reason. The proteins, or parts thereof, can be produced heterologously and antibodies can be raised.

In conclusion, I find this study to have much scientific merit, and find it publishable (if not necessarily in *The EMBO Journal*) provided that

- (1) the necessary manuscript edits be done to properly present the findings in the context of current knowledge on eukaryotic m6A-YTHDF function
- (2) additional analyses be added to fully exploit the obtained data.

The work adds interesting knowledge on the m6A-YTHDF system in planarians, which is important, because knowledge of how this system has evolved in different organisms is key to understand what its origins are and what it really does. However, the study does not reveal any genuinely new molecular principles not already reported in other systems.

REFERENCES

- Arribas-Hernandez, L., Simonini, S., Hansen, M.H., Paredes, E.B., Bressendorff, S., Dong, Y., Ostergaard, L., and Brodersen, P. (2020). Recurrent requirement for the m(6)A-ECT2/ECT3/ECT4 axis in the control of cell proliferation during plant organogenesis. *Development* 147, dev189134.
- Arribas-Hernández, L., Bressendorff, S., Hansen, M.H., Poulsen, C., Erdmann, S., and Brodersen, P. (2018). An m6A-YTH Module Controls Developmental Timing and Morphogenesis in Arabidopsis. *The Plant cell* 30, 952-967.
- Arribas-Hernández, L., Rennie, S., Köster, T., Porcelli, C., Lewinski, M., Staiger, D., Andersson, R., and Brodersen, P. (2021a). Principles of mRNA targeting via the Arabidopsis m6A-binding protein ECT2. *eLife* 10, e72375.
- Arribas-Hernández, L., Rennie, S., Schon, M., Porcelli, C., Enugutti, B., Andersson, R., Nodine, M.D., and Brodersen, P. (2021b). The YTHDF proteins ECT2 and ECT3 bind largely overlapping target sets and influence target mRNA abundance, not alternative polyadenylation. *eLife* 10, e72377.
- Dagan, Y., Yesharim, Y., Bonneau, A.R., Frankovits, T., Schwartz, S., Reddien, P.W., and Wurtzel, O. (2022). m6A is required for resolving progenitor identity during planarian stem cell differentiation. *The EMBO journal* 41, e109895.
- Flores-Téllez, D., Tankmar, M.D., von Bülow, S., Chen, J., Lindorff-Larsen, K., Brodersen, P., and Arribas-Hernández, L. (2023). Insights into the conservation and diversification of the molecular functions of YTHDF proteins. *PLoS genetics* 19, e1010980.
- Tankmar, M.D., Reichel, M., Arribas-Hernández, L., and Brodersen, P. (2023). A YTHDF-PABP interaction is required for m6A-mediated organogenesis in plants. *EMBO reports* 24, e57741.
- Wang, G., Li, H., Ye, C., He, K., Liu, S., Jiang, B., Ge, R., Gao, B., Wei, J., Zhao, Y., Li, A., Zhang, D., Zhang, J., and He, C. (2024). Quantitative profiling of m6A at single base resolution across the life cycle of rice and Arabidopsis. *Nature communications* 15, 4881.

Summary of changes in manuscript item designation

Original item designation	Revised item designation
Figure S1	Figure EV1
Figure S2A-F	Figure EV3
Figure S2G-H	Figure EV4
Figure S3	Figure EV5
Figure S4	Figure EV6
Figure S5	Figure EV7
Figure S6	Figure EV8
Table S1	Dataset EV1
Table S2	Dataset EV2
Table S3	Dataset EV3
Table S4	Dataset EV4
Table S5	Dataset EV5
Table S6	Dataset EV7
N/A (new EV figure)	Figure EV3
N/A (new EV figure panels)	Figure EV6D-E
N/A (new EV figure panels)	Figure EV8B-E
N/A (new EV figure)	Figure EV9
N/A (new EV figure)	Figure EV10
N/A (new EV figure)	Figure EV11
N/A (new EV dataset)	Dataset EV6
N/A	Appendix Table S1
N/A	Appendix Table S2
N/A	Appendix Table S3

Summary of reviewer comments and relevant new data

Reviewer	Core criticism	Summary of experiment / analysis	Key changes
#1 - 1	Mechanism underlying size-reduction phenotype after co-ythdf inhibition is unclear.	Analysis of changes to cell state and progenitor production for finding the processes limiting body size:  1. FACS cell-cycle profiling to detect shifts in stem-cell vs. progenitor fractions. 2. Mitosis counts (H3P) to learn whether reduced proliferation may contribute to the phenotype. 3. TUNEL to quantify apoptosis as potential mechanism underlying size-reduction 4. scRNAseq following co-ythdf (RNAi) analysis to observe cell-type specific effects of co-ythdf (RNAi). 	New data: Fig 7A-B; EV11, 5C; 6D; EV6D-E; EV8B-E, EV10C-D. New scRNAseq Dataset (Dataset EV6).
#1 - 2	Stem-cell differentiation dynamics across lineages not analyzed in depth.	Lineage-resolved profiling  1. FISH/IF for key markers (e.g, epidermal progenitors; dd_234⁺ cells) to quantify progenitor counts. 2. scRNAseq analysis to map cell type gene-expression and cell cycle shifts following co-ythdf suppression. 	New data: Fig 7A,B,E,G; EV11; 5C; EV6B-E; EV10;. New scRNAseq Dataset EV6.
#1 - 3	Function/identity of accumulated kiaa1429-specific (dd_1837⁺) cells remains speculative.	Characterize aberrant population confocal imaging to separate normal dd_1837 expression from the dispersed neural-like state; use scRNAseq data to further investigate this transcriptional state.	New data: Fig 7A; Fig 5C-D; EV10; Dataset EV6
#2 - minor	Test if reader-specific target preference could be inferred.	Performed integration of cell-type-specificity data with differential gene expression analysis and m6A presence.	Analysis is described, but did not result in conclusive evidence due to limited statistical power

#2	Could failed growth stem from feeding vs. turnover differences?	Short dsRNA-injection trial (3 weeks, no feeding) to see if growth defect persists independently of nutrition.	New data: Added Fig EV6D-E, updated text accordingly.
#3 - 1	Plant literature & broader eukaryotic context for m6A/YTHDF missing.	Rewrite introduction & discussion to integrate Arabidopsis data on GGAU motifs and YTHDF redundancy / specialization.	Significant changes in introduction, results, and discussion. New data: Fig EV3
#3 - 2	Need insight into GAU-motif installation mechanism .	Perform sequence analysis of METTL3/14 regions implicated in sequence recognition. Aligned sequence from diverse organisms including mammals and plants.	New data: Fig EV3
#3 - 3	Bulk RNAseq under-used; direct vs. indirect effects unclear.	Perform GO analysis and correlation analysis between the presence of m ⁶ A and differential expression following co-ythdf (RNAi).	New data: Fig EV8B-E

Referee #1:

In the manuscript 'Single-nucleotide m6A mapping uncovers redundant YTHDF function in planarian progenitor fate selection,' Yesharim and colleagues present single nucleotide resolution mapping of m6A sites that expands our understanding of the simple sequence rules that guide m6A deposition on planarian RNAs.

We thank the reviewer for the comment. The analysis we performed is the first of its kind in planarians, and the first time GLORI-seq is applied to a non-mammalian system. We therefore consider this single base, single molecule resolution data to be important for anyone interested in the comparative analysis and evolution of m6A installation.

They also perform single and group knockdown of YTHDF family m6A readers to determine how m6A readers contribute to planarian biology. Surprisingly, individual YTHDF family members have overlapping, but unique expression in planaria, and only cause observable phenotypes when depleted as a group, which includes a smaller size animal size and altered cellular composition. Bulk sequencing of the triple knockdown worms revealed changes in tissue proportion and progenitor numbers, including a decrease in some secretory and phagocytic cell types and expansion of a previously identified abnormal neural-like progenitor state (Kiaa1429 specific cell population).

We thank the reviewer for their thoughtful description of the results. Suppression of writer complex genes leads to a complete loss of m6A activity. This approach has been particularly useful for uncovering the acute, essential functions of m6A in planarian biology (see Dagan et al., 2022). However, it has been less effective in revealing lineage-specific contributions of m6A that do not result in rapid animal deterioration. By systematically analyzing m6A readers, we were able to disentangle roles of m6A in regulating specific planarian cell lineages.

The data is very interesting, and the single nucleotide resolution methods provide new mechanistic insight. However, the current form of the paper does not seem to significantly advance our understanding of m6a function in planarian tissue homeostasis and/or physiology (regeneration, tissue turnover, etc.). Most of the phenotypes they report seem very similar to the phenotype previously observed for kiaa1429 depletion, if less severe.

We thank the reviewer for this important comment. Suppression of *kiaa1429* disrupts all aspects of m6A activity, resulting in a severe phenotype. Surprisingly, it required co-suppression of three *ythdf* genes to uncover a subset of *kiaa1429* depletion phenotype. This revealed a distinct, non-acute, role of the pathway that could not be detected in the rapidly lethal *kiaa1429* suppression model.

This allowed us to uncover lineage-specific contributions of m⁶A to homeostasis, providing new insights into its physiological roles.

To directly address the reviewer's concerns, we performed a set of experiments (described below) that examined the potential causes of the phenotype, and provided additional insights into the mechanism underlying the phenotype.

I've detailed my thoughts on this below:

Major Criticisms:

1) The size reduction phenotype with the YTHDF triple knockdown is very interesting, but the mechanisms driving this phenotype are not deeply explored in the manuscript. Do the animals fission more to produce smaller animals? Is stem cell proliferation reduced? Is apoptosis increased? Since the animals are still capable of regeneration, it seems like an important insight to understand how m6A modifications contribute to the regulation of tissue turnover and stem cell fate determination

We agree with the reviewer and have added new experiments and analyses to pinpoint the cellular basis of the size-reduction phenotype. Collectively, these data argue against global changes in proliferation or cell death and instead support a model in which YTHDF co-suppression perturbs specific progenitor lineages and, as a consequence, limits the animals' ability to utilize nutrients efficiently for growth. The new data and their interpretation in response to the reviewer's comments are detailed below:

(i) Fission and food uptake do not explain the phenotype. We monitored food uptake and fission behavior across cohorts and observed no differences between control and co-*ythdf* (RNAi) animals.

- **We now state this explicitly in the Results (lines 337-338):**

“Co-inhibition of *ythdfs* indeed resulted in size reduction, but it did not affect food uptake or animal fission”.

(ii) Stem-cell proliferation is not globally reduced. We quantified mitoses by phospho-Histone H3 (H3P) immunolabeling, and detected no significant difference between controls and co-*ythdf* (RNAi) (new Fig 7B). Consistently, SMEDWI-1 immunofluorescence did not reveal a reduction in overall neoblast abundance (Fig 7C). In parallel, flow cytometry showed preserved cell-cycle state distributions (G0, G1, S/G2/M) after co-*ythdf* suppression (new Fig 7A). Together, these data argue against a global proliferation defect.

- **New flow cytometry and H3P data (Fig 7A-B):**

- New text describing these results is found in lines 584 - 604.

(iii) **Apoptosis is not elevated.** Whole-mount TUNEL assays showed no increase in apoptotic cells in co-*ythdf* (RNAi) animals compared with controls (new Fig EV11), indicating that a global increase in cell death / turnover is not the driver of reduced size.

- New Figure EV11 showing TUNEL analysis:

- Text describing these results (lines 598 - 604):

“The lack of change in neoblast abundance and cell-cycle dynamics indicated that the size-reduction phenotype following co-*ythdf* (RNAi) was not due to global neoblast insufficiency. We next considered whether co-*ythdf* suppression resulted in increased cell turnover. We counted apoptotic cells by whole-mount TUNEL in control and co-*ythdf* (RNAi) animals (Materials and Methods), and detected no difference (Fig EV11). Therefore, neither general neoblast population dynamics nor cell turnover explained the co-*ythdf* suppression phenotype, pointing instead to subtler, lineage-specific alterations.”

(iv) **Lineage-specific, not global, differentiation effects.** We included a new scRNAseq dataset that we produced from co-*ythdf* (RNAi) and control animals. The scRNAseq analysis and new FISH analysis revealed selective depletion of progenitors belonging to affected lineages, with no change

to that other examined lineages (e.g., epidermal progenitors). These findings fit a model in which m⁶A readers tune differentiation dynamics in specific progenitor compartments rather than acting as global regulators of neoblast output.

- **New figure panels 7F-G examining unperturbed populations of lineage-specific progenitors**

- **New panels EV10C-D examining cell cycle states across cell types**

- **New panel Fig 5C, scRNAseq data showing recovery of all cell types in *co-ythdf* (RNAi) and control samples.**

- **Results and Discussion text was added. In particular in lines 626 - 637:**

“We further tested the hypothesis that *ythdf* inhibition resulted in lineage- and cell-type-specific consequences by measuring the abundance of progenitors of cell types that appeared unperturbed by the *ythdf* suppression (Dataset EV5). First, we quantified the number of epidermal progenitors (*prog-2*⁺) using FISH (Eisenhoffer et al, 2008; Tu et al, 2015; Wurtzel et al, 2017). Our analysis revealed no significant difference in the number of epidermal progenitors between control and co-*ythdf* (RNAi) animals (Fig 7F). We next examined a progenitor population representing a subset of the *cathepsin*⁺ cells (Fincher et al, 2018) (*dd_234*⁺), which appeared to be unaffected by the RNAi (Dataset EV5). Indeed, FISH labeling with *dd_234* and SMEDWI-1 immunolabeling revealed no significant difference in *dd_234*⁺/*SMEDWI-1*⁺ cells following co-*ythdf* (RNAi) (Fig 7G).

Taken together, these results demonstrate that YTHDF proteins were not required for overall stem cell maintenance or proliferation, but were essential for the proper production of specific progenitor populations”.

(v) The phenotype depends on growth context and nutrient utilization. To distinguish active size reduction from impaired growth capacity, we repeated the triple *ythdf* RNAi by dsRNA injection without feeding during the experiment. Under these non-feeding conditions, co-*ythdf* (RNAi) animals were not smaller than controls (new figure, Fig EV6D–E). Conversely, under standard feeding, co-*ythdf* (RNAi) animals ate normally but grew markedly less. Together, these observations indicated that the size reduction phenotype emerged primarily when animals were in growth mode and is not due to a catabolic collapse.

- **New Figure EV6D-E showing lack of size reduction upon delivery of dsRNA without food by microinjection.**

- **Relevant text describing the results (lines 339 - 345):**

“To distinguish whether the size reduction after co-*ythdf* inhibition reflected impaired growth upon feeding or accelerated tissue turnover, we microinjected dsRNA targeting the three *ythdfs* into unfed animals over the course of three weeks (Materials and Methods). Co-*ythdf* (RNAi)

animals did not differ in size from controls (Fig EV6D), despite efficient suppression of the *ythdf* targets (Fig EV6E). These results indicated that the size reduction was not due to increased tissue turnover but rather reduced growth in response to feeding, even though food uptake itself remained intact”.

(vi) Transcriptome analyses point to metabolism. Differential expression and GO enrichment highlight small-molecule metabolic processes among the most significantly impacted biological processes (new figure Fig EV8E). This is compatible with the hypothesis that the combined suppression of *ythdf* activity compromises metabolic programs required to convert ingested nutrients into biomass.

- **New Figure EV8E**

- **Text now reads (lines 458 - 464):**

“Furthermore, we performed gene ontology (GO) enrichment analysis (Materials and Methods) on all differentially expressed genes following co-*ythdf* suppression. Several biological processes were overrepresented (Fig EV8E), including small-molecule metabolic process (GO:0044281) and metabolic process (GO:0008152), with modest enrichment for drug transport (GO:0015893) and drug transmembrane transport (GO:0006855). This enrichment pattern was consistent with perturbed core metabolism and altered transporter activity, which may relate to the observed size reduction”.

Working model and implications. Our integrated data indicated that the size reduction after YTHDF co-suppression is not attributable to increased fission, reduced global neoblast proliferation, or elevated apoptosis. Instead, YTHDF readers appear to control lineage-specific differentiation programs that are collectively required for efficient nutrient utilization and organismal growth. In this view, m⁶A-YTHDF regulation couples the maturation (and/or maintenance) of specific cell types to systemic metabolic capacity. The preserved regenerative ability is consistent with intact core neoblast functions, while the growth defect reveals a context-dependent vulnerability that becomes evident during sustained tissue growth (e.g., with feedings). We have added this model to the Discussion (lines 730 - 738), but we emphasize that the current analysis already localizes the

mechanism away from global proliferation/death and toward lineage-selective and metabolic control.

2) The observation that several adult tissues are depleted and an abnormal progenitor population expands in the triple knockdown is convincing. However, I don't think the authors investigate stem cell differentiation rates in sufficient depth or breadth to make definitive mechanistic insights. Is decreased differentiation across all lineages? Are rates of stem cell symmetric vs. asymmetric divisions altered to increase stem cell numbers while decreased progenitors? Or, are the lineage progenitors specified at similar rates, but dying or being cleared at higher rates to alter final animal composition. These questions should be possible to answer using methods available in planaria. Determining which might be true will provide insight into the function of the m6A modification in planarian cell type specification.

We thank the reviewer for recognizing that the evidence for expansion of the abnormal progenitor population is convincing, and for highlighting the need to distinguish between global differentiation defects and lineage-specific changes. In the revision, we performed new and targeted analyses: FACS cell-cycle profiling, scRNAseq cell-cycle scoring, H3P mitotic counts, SMEDWI-1⁺ cell quantification, lineage-resolved FISH/FISH+IF, and TUNEL assays to dissect these alternatives. Collectively, these data indicate that (i) differentiation is **not globally reduced**, (ii) there is **no detectable shift in neoblast division dynamics** at the whole-animal level, and (iii) the phenotype reflects **cell-type-restricted changes**: depletion of defined parenchymal progenitors and expansion of an abnormal progenitor-like state primarily within neural and *mag-1*⁺ parenchymal (gland) lineages. Below we provide a detailed point-by-point response to the questions raised in this reviewer's comment:

(i) No change to global neoblast dynamics. To determine whether differentiation is broadly diminished across all lineages, we quantified the cell cycle states and mitotic activity using complementary assays. Flow cytometry and scRNAseq cell-cycle scoring revealed no shift in the proportions of G0/G1, S, or G2/M phases between controls and co-*ythdf* (RNAi) animals. Moreover, H3P labeling showed unchanged mitotic counts (Fig 7A-B, Fig EV10C-D). In parallel, the number of SMEDWI-1+ cells, representing neoblasts and their recent progeny, remained unaltered (Fig 7C). Together, these findings argue against a global reduction in differentiation or a shift in symmetric/asymmetric divisions. Therefore, decreased differentiation is not a pan-lineage phenomenon.

- **New data was included. It is found in the response to the previous comment.**
- **The text was modified in several places to describe the results and their interpretation. In particular, the following text was added (lines 635 - 639):**

“Taken together, these results demonstrated that YTHDF proteins were not required for overall stem cell maintenance or proliferation, but were essential for the proper production of specific progenitor populations. They further show, for the first time in planarians, that the production of specific progenitor populations is regulated by the combined activity of YTHDFs, likely co-expressed in the same cell, revealing a hidden layer of gene-expression regulation”.

(ii) Cell-type-resolved fate outputs. Following the reviewer’s comment, we asked whether specific cell types deviate from normal production or maturation. Guided by RNAseq and new scRNAseq, produced during the revision, we performed FISH/FISH+IF for lineages predicted to be stable (e.g., epidermal progenitors; *cathepsin*⁺/*dd_234*⁺) and for those predicted to change (decrease or increase). In agreement with the gene expression data, FISH/FISH+IF analysis demonstrated that the sizes of the epidermal and *cathepsin*⁺/*dd_234*⁺ progenitor population remained unchanged (new panels Fig 7F-G). Specific cell types had an altered number of progenitors. The number of progenitors of the parenchymal subpopulation (SMED-1⁺/*dd_940*⁺) was significantly reduced (Fig 7E), in agreement with gene expression data. Conversely, SMEDI-1⁺ cells expressing *dd_1837* were expanded and localized primarily to neural clusters and the *mag-1*⁺ parenchymal cells (Fig 7D; new panels Fig 6D and Fig EV10A-B). Altogether, this data demonstrates that the phenotype reflects lineage-restricted alterations rather than a uniform decline (or increase) to output of the neoblast compartment.

- Relevant text was modified in the Results section describing Figure 7 and in the Discussion.
- New panels 7F-G were shown in response to a previous Reviewer’s comment
- New Figure Fig 6D shows upregulation of *dd_1837* in two lineages following *co-ythdf* suppression:

(iii) Division symmetry and proliferative mode. If YTHDF loss broadly favored symmetric self-renewal at the expense of differentiation, we would expect elevated H3P⁺ counts, and an expanded SMEDWI-1⁺ pool. Neither outcome was observed: mitosis counting by H3P labeling was unchanged, flow cytometry analysis of cell cycle state distribution was stable, and SMEDWI-1⁺ counts did not increase (Fig 7A-C). While subtle changes below detection cannot be entirely excluded, the aggregate measurements do not support a whole-animal shift in symmetric versus asymmetric division.

- New figure panels (7A and B) were presented in response to a previous Reviewer’s comment.

(iv) Progenitor loss versus clearance. We distinguished between reduced progenitor production and increased cell loss by quantifying apoptosis using whole-mount TUNEL. We detected no global change in TUNEL⁺ cells in co-*ythdf* (RNAi) animals relative to controls (Fig EV11). Given technical constraints on TUNEL/FISH co-labeling, we obtained the TUNEL measurements independent of cell type lineage counts. The combination of unchanged apoptosis counts, depletion of specific parenchymal progenitors, and expansion of the dd_1837⁺ population best fits a model of altered lineage output/state rather than accelerated clearance.

- **New Figure EV11 was presented in response to a previous Reviewer's comment.**
- **Relevant text was added, in particular, lines 601 - 604:**

“We counted apoptotic cells by whole-mount TUNEL in control and co-*ythdf* (RNAi) animals (Materials and Methods), and detected no difference (Fig EV11). Therefore, neither general neoblast population dynamics nor cell turnover explained the co-*ythdf* suppression phenotype, pointing instead to subtler, lineage-specific alterations”.

In summary, the convergent evidence from cell cycle analysis, mitotic counts, neoblast abundance, apoptosis measurements, and lineage-resolved markers supports a model in which combined YTHDF activity enforces proper progenitor fate selection in specific contexts. In the absence of YTHDFs, normal production of specific cell types (e.g., dd_940⁺ parenchymal cells) is reduced, while an abnormal progenitor-like transcriptional state (dd_1837⁺/dd_422⁺) accumulates, particularly within the neural and *mag-1*⁺ parenchymal (gland) compartments. This provides mechanistic insight into how YTHDF/m⁶A-dependent regulation influences planarian cell-type specification without invoking a global defect in differentiation or a systemic change in division symmetry.

3) The dramatic accumulation of the kias1429-specific cells is very interesting, and suggests that this cell state accumulates when m6a modifications are not properly regulated. But, what does the cell state do? What genes are expressed in this cell type? Do any of them imply a cellular function that might explain the decreased size and altered cellular composition of the worms? The distinct spatial localization of the cell type suggests it may not be a random abnormal progenitor. Connecting the function or existence of this cell type to the other phenotypes observed would deepen the insight of the paper and the impact of its' findings.

We thank the reviewer for this insightful comment. We agree that the distinct spatial pattern (i.e., parapharyngeal) of dd_1837⁺ cells argues that it represents a distinct cell type. Yet, importantly, following co-*ythdf* (RNAi) the expression of dd_1837 becomes widespread well-beyond the parapharyngeal region. Evidence from confocal microscopy combined with scRNAseq indicated that normally the expression of dd_1837 was primarily localized to a subset of parenchymal cells. The co-suppression of *ythdfs* resulted in the emergence of dd_1837⁺ predominantly in neural and *mag-1*⁺ cells, which are distributed across the body. This expanded dd_1837⁺ population is

molecularly distinct from the normal *dd_1837*⁺ parenchymal subset. For example, 11 of the top 20 upregulated genes having cell type annotation following RNAi are characteristic of neuronal lineages (Dataset EV5). By contrast, parapharyngeal *dd_1837*⁺ cells (cluster 6 in the scRNAseq produced here) lack a neuronal bias. Together with the co-localization of *dd_1837*⁺ expression to *mag-1*⁺, these data are more compatible with the interpretation that the accumulated *kiaa1429*-specific cells have aberrant, neural-biased (and partially gland-associated) progenitor-like program, rather than an expansion of a normal parenchymal *dd_1837*⁺ identity or state (Fig 6C–D, Fig EV10A-B).

Importantly, we detected both SMEDWI-1⁺ and SMEDWI-1⁻ *dd_1837*⁺ cells in the *co-ythdf* (RNAi) condition. This is consistent with ongoing production followed by loss of SMEDWI-1 expression during maturation. However, analysis of global mitotic numbers (H3P) and neoblast/progenitor pool size (SMEDWI-1 counts) argued against a whole-animal shift in divisions as the driver of *dd_1837*⁺ accumulation (Fig 7A,B,F,G). Taken together, the most parsimonious interpretation is that progenitors fail to suppress this transcriptional state, allowing it to accumulate over time.

Notably, our GLORI analysis showed that markers of this state (e.g., *dd_422*) are heavily m⁶A-modified (Fig 6E), suggesting that under normal conditions YTHDFs repress these genes, which map to repetitive genomic regions (Dagan et al., 2022). Given these observations, it is unlikely that this cell state has a defined physiological function. Rather, it likely represents an aberrant transcriptional program that persists when m⁶A-dependent regulation is disrupted.

- **New panels (Fig 6C) showing high resolution images of the emerging *dd_1837*⁺ cells (top), compared to the cells that normally expressed this marker (bottom).**

- **New panels Figure EV10A-B: UMAP visualization of the cells expressing markers associated with the abnormal transcriptional state (i.e., *kiaa1429* (RNAi)-associated):**

- The text was edited to describe these results and their interpretation, including the following (lines 540-547):

“To pinpoint the cell types expressing markers of the *kiaa1429* (RNAi)-specific cells after *co-ythdf* (RNAi), we analyzed the scRNAseq dataset, and examined the expression of the markers *dd_1837* and *dd_422* (Figs 6D and EV10A-B). A striking induction was limited to clusters 23 and 24 (Figs 6D and EV10A and B), corresponding to subsets of *mag-1*⁺ parenchymal and neural cells, respectively (Fig EV9C; Dataset EV6). Moreover, induction was also detectable in several additional clusters (Fig EV10A and B), arguing against overproduction of a normal cell type and instead pointing to an aberrant, or normally transient or rare cell state that may arise from altered maturation or a change in transcriptional regulation.”

Minor Criticisms:

1) It would be more informative if the authors used box plots with data points to allow interpretations of animal number and replicates

We thank the reviewer for the comment. We will provide the source data for the figures and groups sizes in the figure legend, as we found that adding the individual datapoints frequently reduced readability.

2) The authors state that cellular quantitation of microscopy images were performed manually, so were the analysts blinded to the conditions they were quantitating?

Quantitation of microscopy images was performed by several analysts, but not blinded. Imaging, image processing, and quantitation were reviewed independently by the senior author.

Referee #2:

This study resolves m6A sites in planarians at single-nucleotide positions across the transcriptome and comprehensively determines roles for the ythdc class of m6A reader proteins. The authors find ythdc factors have overlapping but tissue-specific expression, evoking a possible code of m6A regulation important in cell-type specific gene regulation. Surprisingly, knockdown of a combination of three different ythdf factors led to a novel phenotype of reduced animal size without loss of tissue homeostatic integrity. This is particularly interesting because most perturbations of planarian growth described to date cause a wholesale failure of stem-cell-dependent tissue maintenance. A cross-species analysis of m6A site conservation surprisingly indicates widespread rapid evolutionary changes to m6A sites. Analysis of genes misregulated after ythdc RNAi identified several classes of cell-type specific markers that m6A detection is important for specifying the abundance of several classes of tissue-specific progenitors, particularly in the parenchymal compartment. Therefore, a combination of ythdc factors exerts control over cell type specific gene regulation relevant for distinct tissue renewal programs.

We thank the reviewer for the excellent summary of the manuscript.

The work is clear and compelling, and it makes a novel contribution to understanding both gene regulation and genomics in planarians, as well as the general properties of m6A. As such the work would be very interesting to a broad audience. I believe the work is already comprehensively conducted and does not require additional experiments or analysis to make the key points.

We thank the reviewer for highlighting the novelty of the manuscript, and for acknowledging the contributions of the study.

I have some minor suggestions or ideas for future work below.

Minor comments:

P9 clarify what is meant by "nuclear ythdc-1", do authors mean that they are testing only the nuclear component of YTHDC-1 activity or that they predict YTHDC to be mainly a nuclear protein, or some other interpretation?

We thank the reviewer for the opportunity to clarify. In our usage, “nuclear *ythdc-1*” refers to the planarian ortholog of the nuclear YTH family member YTHDC1. Both YTHDFs and YTHDC1 contain a YTH domain and both recognize m⁶A, but they are paralogous clades with distinct subcellular roles and evolutionary histories (see Fig EV5B). YTHDC1, based on analyses in other organisms, predominantly functions in the nucleus, whereas YTHDF proteins are mainly cytoplasmic. In this

manuscript, we focus on YTHDF contributions, as we analyzed the functions of the single planarian *ythdc-1* in previous work (Dagan et al, 2022).

- We clarified the text to avoid ambiguity and it now reads (lines 253 - 254):

“Our recent functional analysis of the planarian MTC and *ythdc-1* (predicted to encode a nuclear YTH family member)...”

The co-expression analysis is a creative application of the single-cell data and interesting to see that individual cell types are enriched by different combinations of the ythdc homologs. I wonder, could mRNAs expressed uniquely in different cell types provide a way to test whether distinct ythdc complexes might have a preference for different target m6A sites? Or would the authors predict that ythdc factors all have the same specificity? If there were differential specificity I wonder if this could help to explain the diverse tissue-specific effects observed after inhibition of these genes.

We thank the reviewer for this thoughtful suggestion. In principle, cell-type-restricted mRNAs can serve as natural reporters of paralog-specific reader activity. Using Dataset EV5 (cell-type expression, per-gene m⁶A sites, and differential expression), we intersected differentially expressed genes with those uniquely expressed in a single cell type, identifying 91 candidates (FDR < 0.01; 503 additional genes were expressed in multiple cell types). This small set limited statistical power. Moreover, only a minor fraction of these genes changed upon single-paralog inhibition, even in contexts with clear *ythdf* enrichment (e.g., *ythdf-A* in the intestine by FISH). Consequently, we did not detect robust paralog-specific targeting. However, this may reflect limited sampling rather than an absence of specificity. The accompanying plot summarizes these outcomes (bar color denotes RNAi condition).

The size control phenotype is quite interesting. Could this be due to an inability to grow upon feeding or instead be a result of faster turnover? It may be beyond the scope of this report, but would be interesting to know whether the triple ythdc-A;-B;-C RNAi delivered without food would

cause an acceleration of degrowth. Could any of the target mRNAs misregulated after triple-gene *ythdc* RNAi potentially help explain the failed growth phenotype?

We thank the reviewer for this interesting comment. We addressed the comment by delivering dsRNA without feeding, by using microinjection and examining the sizes of the *co-ythdf* (RNAi) animals compared to controls. Importantly, because size differences increase over time, a key concern was that starvation-induced degrowth could dominate over RNAi effects. We therefore performed the injections over 3 weeks, where starvation-induced degrowth is negligible. Following 3 weeks of injections, we observed no difference in animal sizes between the treatment and control groups, indicating that the suppression of *ythdfs*, at this time window, did not induce an accelerated tissue turnover.

- **We added this result (Figure EV6D-E). The text reads (lines 339 - 345):**

“To distinguish whether the size reduction after *co-ythdf* inhibition reflected impaired growth upon feeding or accelerated tissue turnover, we microinjected dsRNA targeting the three *ythdfs* into unfed animals over the course of three weeks (Materials and Methods). *Co-ythdf* (RNAi) animals did not differ in size from controls (Fig EV6D), despite efficient suppression of the *ythdf* targets (Fig EV6E). These results indicated that the size reduction was not due to increased tissue turnover but rather reduced growth in response to feeding, even though food uptake itself remained intact”.

- **New Figure panels (EV6D-E):**

The size reduction phenotype is very clear. I wonder if a jittered scatterplot overlaying the box plots could even more strongly show the weight of evidence (in terms of n) that Figure 3D depicts?

We thank the reviewer for this comment. We tested this option across several plots but found that displaying all individual observations reduced readability. For instance, in Figure 3D there are more than 150 data points, which obscure the overall pattern. To ensure transparency, we will submit the source data for the charts together with the manuscript, allowing full replication and independent examination of the data.

Relatedly and as a point perhaps for future work to understand the developmental mechanisms of ythdc factors, the failed growth phenotype is indeed rather rare in the planarian literature I believe. Miller and Newmark reported a similar phenotype after inhibition of an insulin-like peptide or an insulin receptor. Perhaps that gene is targeted by m6A and the ythdc-a/-b/-c factors or could signal upstream of downstream m6A deposition/detection? (doi: 10.1387/ijdb.113443cm). To clarify, I think the current manuscript is fully complete and do not think this needs to be explored in this paper but could be an interesting idea for future research.

Thank you for the comment, the phenotype described by Miller and Newmark indeed resembles the size reduction we observed. We inspected our data, whether the expression of *inr-1* (dd_9272) or *ilp-1* (dd_13874) is perturbed following co-*ythdf* suppression, or whether they have m6A installed. We did not find evidence for either, which suggests that m⁶A is not an upstream gene expression regulator of *inr-1* or *ilp-1*. We of course cannot rule out the opposite (that *inr-1* or *ilp-1* are regulators of *ythdfs*), however there is no data available for that, and we consider it rather unlikely. Finally, we think that suggestion for further analysis of the functions of genes targeted by *ythdfs*, for example by performing an RNAi screen targeting candidates identified from the transcriptional changes observed during degrowth, could be a very promising approach to uncover novel regulators of tissue growth and degrowth.

The work identifies and characterizes several novel progenitor populations regulated by m6A. Are any previously identified planarian progenitors (eg eyes, pharynx, epidermis, etc) also identified through the RNAseq analysis? It is interesting to consider how/why m6A may be critical for some lineages but not others.

This is a great question. Re-inspection of our RNAseq did not reveal additional progenitor lineages (e.g., epidermis) affected beyond those reported. Consistently, quantification of epidermal progenitors (*prog-2*⁺ cells) by confocal microscopy showed no change (new panel Fig 7F). In the revision, we added several approaches to examine changes to the progenitor population: (i) flow-cytometry analysis following co-suppression of *ythdfs* (new panel Fig 7A), which showed no detectable change to the proportion of the progenitor or neoblast populations; (ii) measurements of two additional progenitor populations (epidermis and dd_234⁺ cathepsin⁺ cells; new panels Fig 7F-G); and (iii) mitotic activity by H3P labeling (new panel Fig 7B). Across these assays, we find no evidence for a global progenitor defect. To explain lineage-specific sensitivity, we consider several explanations: (1) paralog redundancy/reader dosage: overlapping *ythdf* expression can be variable between lineages. In the epidermal lineage this overlap may buffer perturbations to the expression of the *ythdfs*; (2) detection limits in rare lineages (e.g., eye), where subtle effects may be masked by low cell numbers; and (3) target architecture: some lineages may be less dependent on YTHDF/m⁶A if key fate determinants lack functionally positioned m⁶A sites.

- In addition to the new data described above, we added the following Discussion text (lines 730-738):

“The effects observed after *co-ythdf* suppression were lineage-restricted, not a global disruption of neoblast differentiation. They likely reflect two factors. First, reader-dosage redundancy across progenitor lineages, with YTHDFs operating at different levels in distinct lineages. Second, lineage-specific architecture of m⁶A targets, with transcripts methylated differently across cell types. Consistent with this model, epidermal and *dd_234*⁺ *cathepsin*⁺ progenitors were unaffected by *co-ythdf* suppression, whereas other lineages (e.g., *dd_940*⁺ parenchymal) showed pronounced m⁶A dependence. Taken together, these findings indicate no global progenitor defect, but instead context-dependent sensitivity to YTHDF activity and m⁶A regulation.”

- New flow cytometry and H3P data (Fig 7A-B).

- New figure panel 7F-G examining populations of lineage-specific progenitors

Referee #3:

The study by Yesharim et al. uses flat worms to study the regulatory system relying on N6-methyladenosine (m6A) in mRNA and m6A-binding proteins in the YTHDF phylogenetic clade. The eukaryotic systems with the most advanced knowledge on molecular mechanisms and biological functions of this very important regulatory system are currently vertebrates (with important contributions mainly from human cell culture, mouse and zebrafish) and the plant Arabidopsis thaliana. Investigations on m6A-YTHDF in flat worms are, therefore, a very welcome addition to the objective of understanding eukaryotic m6A function, not least because considerable controversy on the matter exists in the mammalian literature in particular, but even the plant literature is not completely spared for this problem either.

We thank the reviewer for this thoughtful analysis. We fully agree that studying m6A biology in diverse organisms is essential for advancing our understanding of this important modification. Moreover, the very recent emergence of single-resolution m6A detection approaches (e.g., GLORI, m6A-sac-seq, m6A nanopore) have so far yielded very few publications, which focused on mammals, especially cell lines, and recently in plants (Wang et al., 2024). Planarians are strategically positioned within the animal phylogeny, therefore enabling comparative studies with vertebrates and, as the reviewer notes, offer valuable opportunities for broader comparisons, including with plant systems.

The current study rests on the previous finding that the m6A methyl transferase in flatworms is essential because knockdown of its subunits causes intestinal malfunction and inability to take up food, a phenotype explained largely by defective function of the nuclear m6A-binding protein YTHDC (Dagan et al., 2022). This study follows up by addressing two key questions:
(1) Which transcripts and which sequence contexts are methylated in flat worms, as addressed by single-nucleotide resolution transcriptome-wide m6A mapping?
(2) What are the biological functions planarian YTHDF proteins, and do they have specialised or redundant functions?

To address the first question, the authors successfully implement the single-nucleotide resolution GLORI-seq technique. These results are high-quality and provide a very useful basis also for future studies. The most interesting finding is that m6A in planarians does not only occur in the canonical DRACH context, but also in GAU-type motifs, often with a U at the +4 position. This suggests that the methyl transferase itself has been shaped a bit differently by evolution in planarians compared to mammals.

We thank the reviewer for acknowledging the quality of our results, and that they provide a very useful basis for future research. We believe several key observations from our analysis are

important, unexpected, and contribute to a deeper understanding of m6A regulation in planarians, offering intriguing points for cross-organism comparisons:

(1) The m6A/A stoichiometry is dramatically higher in planarians than in mammals (Liu et al., Nat Biotech, 2023) and in plants (Wang et al., Nat Comm, 2024), exhibiting a much more ‘switch-like’ behavior than in other profiled species.

(2) The rapid sequence divergence of m6A sites between planarian species suggests that the simple planarian motif enables rapid gain and loss of methylated sequences.

(3) Leveraging the single-base, single-molecule resolution of GLORI-seq, we performed an m6A site-pair co-occurrence analysis, representing, to our knowledge, the first such analysis outside of mammalian cell lines. The absence of linkage between methylation site-pairs suggests that the writer complex operates in a non-processive manner in planarians.

- **In the revision, we added a Discussion of the ‘switch-like’ installation stoichiometry of m6A in planarians, and potential consequences (lines 685 - 690, and 693 - 694)**

- “Notably, per-site m6A:A stoichiometry was substantially higher in planarians than in mammals (Liu *et al*, 2023) or plants (Wang *et al*, 2024), indicating a more ‘switch-like’ modification regime than reported in other profiled taxa. Analysis of the detected m⁶A sites revealed that m⁶A deposition in planarians is governed by relatively simple sequence determinants, with strict requirements that distinguish compatible from incompatible sequences”.
- “This pattern suggests that, in planarians, evolution acts chiefly by gaining or losing compatible m⁶A sites, not by adjusting per-site methylation levels”.

However, very similar results, in particular methylation of (G)GAU motifs and enrichment of pyrimidines just downstream of methylated As (often ACUCU) have been reported for the plants Arabidopsis thaliana and Oryza sativa (Arribas-Hernández et al., 2021a; Wang et al., 2024), so it is not a genuinely new observation that the methyltransferase can evolve to methylate this type of motif in addition to the canonical DRACH motif.

We thank the reviewer for this important comment. It is indeed intriguing to observe such a similarity in the m6A installation motif between plants and planarians, which contrasts with analyses in vertebrates. Plants and planarians occupy distant branches of the eukaryotic tree, thus pointing the planarian-plant similarity represents either convergence or a deeply conserved feature. We therefore find that this planarian-plant similarity strengthens – rather than undermines – the significance and general relevance of our findings, which otherwise could have been perceived as a phylum-specific curiosity.

- **In the revised manuscript, we added the following text to the discussion (lines 697 - 703):**

“These results echo plant studies reporting m6A enrichment at GAU motifs (Wang *et al*, 2024; Arribas-Hernández *et al*, 2021a). Given the evolutionary distance between planarians and plants, broader taxon sampling is needed to determine whether this similarity represents convergent evolution or deep conservation. Either way, the shared motif highlights the MTC flexible substrate recognition across distant lineages, a property, to our knowledge, not yet detected in animals beyond planarians”.

- **In the Results section we added the following text (lines 205 - 206):**

“Our observation that the planarian MTC efficiently modified GAU motifs is reminiscent of findings in *Arabidopsis* (Arribas-Hernández *et al*, 2021a; Wang *et al*, 2024)”.

What would be expected for a high-impact publication now would be identification of the methyltransferase properties that allow GAU methylation in addition to DRACH methylation. At the very least, an analysis of alignments of methyl transferase subunit sequences with including planarian/plant and mammalian sequences would be expected to see if it is possible to pinpoint candidate residues implicated in this important expansion of target sequences.

We appreciate this suggestion and agree that identifying MTC features that determine motif preference is important. In the revision, we added focused multiple sequence alignments of two MTC regions previously implicated in sensing and recognizing RNA substrates (new Figure EV3A-B). First, we aligned the sequences of METTL3 and METTL14 at regions shown to be involved in altered RNA-sequence preference in cancer-related mutations (Zhang *et al*, Science Advances, 2024, PMID: 39705353; Qi *et al*, ResearchSquare, 2024, PMID: 37609305) . Specifically, human METTL14 R298P or METTL3 R471H substitutions are implicated in altered RNA motif specificity where the mutation increases the preference for GGAU over GGAC. Our multiple sequence alignment of these regions show a complete conservation of these residues between mammals, vertebrates, planarians, *Arabidopsis*, and several invertebrates. Therefore, mutations that are known to alter the MTC specificity in cancer are not naturally present in either planarians or *Arabidopsis*.

Second, we analyzed the METTL3 ZnF1/2 RNA interfacing module across planarian species, human, mouse, and three *Arabidopsis* species (new Figure EV3C). The function of the residues in generating sequence-specificity is largely unknown (Huang *et al*, Protein Cell., 2018, PMID: 29542011). However, we detected several ZnF1/2 residues that are only partially conserved between organisms on the RNA-interaction surface. Most notably, human R301 (conserved in *Arabidopsis*) is substituted for lysine (K) in planarians. Given our GLORI motif analysis showing that -1 G is not required in planarians, especially when +1 C is present, this substitution could potentially contribute to the broader -1 tolerance we observe in planarians. R301 is located on the ZFD RNA-interaction surface: charge-reversal mutation at this site (R301D) sharply reduced enzymatic activity (Huang *et al*, 2018).

The planarian substitution (R301K) does not alter the charge, and we speculate that due to its likely interaction with the RNA substrate it is likely more consistent with alteration in binding specificity.

Importantly, strong residue-to-nucleotide assignments are not feasible because RNA-bound, sequence-resolved structures of the MTC have yet to be published or deposited. Existing structures, ZFD by NMR or MTase core with product/bisubstrate analogs, do not resolve base-specific contacts for the incoming RNA, especially at positions $-1/+1/+2$. This limits the mechanistic conclusions that could be derived from sequence and structure comparisons.

We underscore that the planarian and plant motif similarity at +1, alongside the relaxed -1 requirements in planarians, is interesting, and as the reviewer notes, opens promising avenues for deeper mechanistic work. We limit the addition to this revision to inclusion of sequence-based hypotheses. However, biochemical reconstitution and true RNA-substrate co-structures will be essential for strong, residue-level mechanistic conclusions.

- **In the revision, we added the following text to the Results (lines 205 - 218):**

“Our observation that the planarian MTC efficiently modified GAU motifs is reminiscent of findings in *Arabidopsis* (Arribas-Hernández *et al*, 2021a; Wang *et al*, 2024). Yet the molecular basis for the extended sequence preference was unclear. We therefore compared METTL3 and METTL14 sequences from representative organisms and focused on two regions implicated in substrate specificity. First, we examined residues linked to a shift in preference toward GGAU over GGAC, which is observed in cancer-associated substitutions: METTL14 R298P and METTL3 R471H (Zhang *et al*, 2024; Qi *et al*, 2024). These residues were invariant across organisms and therefore variation at these sites cannot explain the altered sequence preference in planarians (Fig EV3A and B). Second, we compared the METTL3 ZnF1/2 RNA-recognition module (Fig EV3C). The basic RNA-contact surface was broadly conserved, but a notable difference was at the position corresponding to human R301 (conserved in *Arabidopsis*; Fig EV3C), which was substituted by a lysine in planarians. Given that -1 G is not required in planarians, especially when +1 C is present, this substitution might contribute to broader -1 tolerance. However, no single decisive residue emerged from the alignments”.

- **We modified the Discussion (lines 697 - 703).**
- **We added Figure EV3, which describes the focused multiple sequence alignment. Shown below is only panel A of the alignment, because of the size of the full figure:**

To address the second question, a variety of approaches are used to study the five planarian YTHDF proteins. The most important observation is that only combined knockdown of the three most highly expressed YTHDFs produce phenotypes and that these YTHDFs are largely co-expressed, suggesting (but not proving, as the authors correctly and carefully note) redundant molecular function. The study also includes valuable insight into what the cellular defects of the combined YTHDF knockdowns are, including clear demonstration that the intestinal defects seen in methyltransferase and YTHDC knockdowns are not detected in the YTHDF knockdowns.

We thank the reviewer for the analysis of our findings, and considering our insights valuable.

The study is well executed and adds important new insights into the m6A-YTHDF system in planarians, but it does not reach the level where concrete, genuinely new insights into eukaryotic m6A-YTHDF function are obtained.

We thank the reviewer for recognizing the significance of our findings in advancing the understanding of m6A-YTHDF function in planarians, but consider these findings impactful beyond the planarian system. The ability to perturb YTH protein activity and analyze their gene expression at an organismal scale is critical for tackling the highly controversial question pertaining to the redundancy of these different proteins in animals. Prior studies in mammalian systems have reached diametrically opposing conclusions: some reporting that each YTH protein regulates strikingly different aspects of gene expression (e.g., Wang et al, 2015: 10.1016/j.cell.2015.05.014; Zou et al., 2023: 10.1186/s13059-023-02862-8), whereas others have suggested that the three human YTH proteins are highly redundant (e.g., Zaccara & Jaffrey, 2020). Given that our phylogenetic analysis of planarian YTHDFs reveals striking divergence from their vertebrate counterparts, using planarians to examine whether there is potential redundancy in YTHDF activity is particularly intriguing.

Our analysis of planarian m6A-YTHDF activity supports a nuanced model that, following the reviewer's critique, we recognize aligns more closely with observations in plants (e.g., Flores-Téllez et al., 2023). Phenotypically, planarian YTHDFs exhibit substantial, though not complete, functional redundancy, which is likely explained by the partial overlap in their gene expression. Analysis of gene expression changes following systematic *ythdf* suppression supports this model indirectly. These findings suggest a potential evolutionary convergence in mechanisms of redundancy within the m6A-YTHDF system, which, given the significant divergence of the YTHDF gene family within metazoans and between metazoans and non-metazoans, represents an important broader insight. In the following point-by-point responses, we detail the specific revisions made to address this issue.

Points of criticism to address in a revised version of the manuscript are listed below:

1. In its present form, the manuscript does not frame the study correctly within the current understanding of eukaryotic m6A function. Nearly all of the plant literature is entirely overlooked which for this study in particular is quite serious for the following reasons:

(i) The GAU context of m6A methylation has already been described in plants (Arribas-Hernández et al., 2021a; Wang et al., 2024).

Thank you for the comment. We acknowledge this issue and, as noted above, have included a new analysis in the Results section (Fig EV3) and text in the Discussion, as described above.

(ii) The matter of YTHDF redundancy/specialization is better described in Arabidopsis than in any other eukaryotic system.

(A) It is the first system where clear genetic redundancy between divergent YTHDFs was described (Arribas-Hernández et al., 2018; Arribas-Hernandez et al., 2020).

(B) Evidence for co-expression, target overlap, and in vivo competition for target binding between genetically redundant YTHDFs has been obtained (Arribas-Hernández et al., 2021b).

(C) Clear evidence for redundant molecular function of 8/11 YTHDFs, but specialized functions of 3/11 YTHDFs has been obtained (Flores-Téllez et al., 2023), with some understanding of molecular elements needed for redundant functions (Flores-Téllez et al., 2023; Tankmar et al., 2023).

Hence, thorough mention of that knowledge is mandatory when the question of YTHDF specialization is brought up, and comparisons to that frame of reference is appropriate in the discussion.

We agree with the reviewer's comment, and acknowledge that we originally focused our framing and comparison with animal systems. In the revision, we now incorporated a broader perspective that includes key insights from the extensive plant literature. This was integrated both to the introduction, results and discussion.

- **The Introduction now reads (lines 70 - 75):**

“Whether these factors function in a redundant manner, as suggested by several studies in vertebrates (Zaccara & Jaffrey, 2020; Kontur *et al*, 2020; Lasman *et al*, 2020b), exert specialized, state or lineage-specific regulatory roles (Liu *et al*, 2020b), or follow a hybrid model, where some YTHDFs are redundant and others are specialized, as established in *Arabidopsis* (Arribas-Hernández *et al*, 2018, 2020, 2021b; Flores-Téllez *et al*, 2023), remains unknown”.

- **The Results now read (lines 315 - 317):**

“Importantly, in plants, it is well-established that many, but not all, YTHDF paralogs exhibit redundant activity (Arribas-Hernández *et al*, 2018, 2020, 2021b, 2021a; Flores-Téllez *et al*, 2023)”.

- **The Discussion now reads (lines 739-747):**

“Our observations are compatible with the possibility that planarian YTHDFs may be molecularly redundant, but because of the distinct evolutionary histories (e.g., gene duplications) of planarian and vertebrate YTHDFs, such redundancy is likely the consequence of independent processes. Importantly, YTHDF functions show both redundancy and specialization across eukaryotes. In *Arabidopsis*, for instance, some YTHDF paralogs act redundantly, while others perform specialized roles (Arribas-Hernández *et al*, 2021a, 2021b; Flores-Téllez *et al*, 2023). We therefore suggest that the balance between redundancy and specialization is shaped by species-specific duplication histories and expression programs, rather than being hardwired to particular YTHDF identities.”

2. A phylogenetic analysis of YTHDFs, including the five planarian YTHDFs, is of course mandatory for this study. But the analysis shown in Figure 3B is too simplistic and superfluous. It is relevant to know how many clades these YTHDF proteins represent, how different they really are from more studied YTHDFs in other organisms (at least vertebrate and fly YTHDFs), whether they are planarian peculiarities, or found more widely in metazoans? For inspiration on how such analyses can be done, Figure 1 in (Flores-Téllez *et al*, 2023) can be consulted.

We thank the reviewer for the comment. A detailed phylogenetic tree and its associated multiple sequence alignment are provided in Figure EV5. Figure 3B is intended as a simplified schematic to orient the reader. Our analysis indicates that there is no clear one-to-one orthology between vertebrate paralogs and planarian YTHDFs, and additionally, the vertebrate *ythdf* duplication events appear more recent. These differences make the comparison of YTHDF-m6A between planarians and vertebrates even more intriguing.

3. The mRNA-seq data in YTHDF knockdowns are not sufficiently exploited. We should know (i) whether differentially expressed genes tend to be m⁶A targets or not to have an idea of whether expression differences could be a direct consequence of YTHDF knockdown or not

We thank the reviewer for this suggestion. Following the comment, we tested whether differentially expressed (DE) genes tend to be m⁶A targets. Because detectability of m⁶A sites increases with expression level, we used an expression-matched bootstrap (n=1,000): in each iteration we sampled genes from the same expression bins with or without mapped m⁶A sites and compared the fraction that were DE after co-ythdf RNAi. Across iterations, we did not detect an enrichment of DE among m⁶A-containing genes relative to expression-matched genes lacking sites. This result is consistent with our experience analyzing whole-organism RNAseq, where indirect effects and cell-composition changes often dominate as perturbation effects accumulate over time (e.g., loss of a cell type reduces expression of that cell type's markers irrespective of direct reader targeting).

- **The Results section describe the analysis (lines 445 - 446):**

“Genes showing altered expression were not enriched for m⁶A sites (Fig EV8B-D; Materials and Methods)”.

- **The Methods section detail the bootstrapping approach applied (lines 942 - 955).**
- **Figure panels EV8B-D were added to show the results of this analysis.**

(ii) not only where differentially expressed genes are expressed, but also whether they encode proteins acting in particular pathways or with shared molecular function - a gene ontology-type analysis, as simple as it is, is still informative.

We thank the reviewer for this suggestion. Using Gene Ontology annotations from the planarian genomic resource, PlanMine, enrichment analysis was performed on the differentially expressed gene set. Differentially expressed genes were enriched for small-molecule metabolic processes, consistent with perturbation of core metabolism following co-ythdf suppression. We also observed a more modest enrichment for drug transport, suggesting altered transporter activity often associated with detoxification responses. We speculate these shifts are compatible with changes in nutrient utilization, which could provide a molecular correlate of the body-size reduction we observed.

- **A summary was added to the Results section (lines 458 - 464):**

“Furthermore, we performed gene ontology (GO) enrichment analysis (Materials and Methods) on all differentially expressed genes following co-*ythdf* suppression. Several biological processes were overrepresented (Fig EV8E), including small-molecule metabolic process (GO:0044281) and metabolic process (GO:0008152), with modest enrichment for drug transport (GO:0015893) and drug transmembrane transport (GO:0006855). This enrichment pattern was consistent with perturbed core metabolism and altered transporter activity, which may relate to the observed size reduction”.

- **Figure panel showing this result was added (Figure EV8E):**

4. It is a major weakness of the paper that it does not contain information on transcripts bound by the YTHDFs (at least YTHDF A/B/C). The authors state in the discussion that antibodies are not available for these proteins, but this is hardly a good reason. The proteins, or parts thereof, can be produced heterologously and antibodies can be raised.

We thank the reviewer for the comment. We completely agree on the importance of obtaining high-quality maps of YTH binding, and as mentioned by the reviewer, we explicitly state this limitation. However, we respectfully disagree that the absence of CLIP/RIP data constitutes a major weakness for the current manuscript, and we believe, based on efforts that we already made, that it is outside the scope of this manuscript. First, our conclusions rest on multiple orthogonal lines of evidence (e.g., m⁶A maps, YTHDF functional analysis, transcriptomic/in situ analyses) that together form the basis for our model for YTHDF-dependent biology.

Second, we extensively evaluated an antibody-independent in vitro approach during the work of this project. Specifically, we produced recombinant planarian YTHDF proteins and performed RNA pulldown assays following Atanasoai et al. (bioRxiv, 2021). In our hands, binding to unmethylated control RNAs was considerable even after extensive optimization, and the assays did not yield robust discrimination that would support confident conclusions about sequence or modification preferences. Given these technical limitations and the known context dependence of YTH-RNA interactions *in vivo*, we considered these data insufficiently informative and did not include them.

Third, while we agree that raising antibodies is a conceivable path, obtaining CLIP/RIP-grade antibodies that work in planarian tissue is a substantial, dedicated effort. I consider developing such reagents a separate project to pursue, but it is beyond the scope of the present study and revision.

In summary, we concur that mapping YTHDF-RNA interactions will be informative, but we maintain that our current data strongly support the central claims, while acknowledging this limitation.

In conclusion, I find this study to have much scientific merit, and find it publishable (if not necessarily in The EMBO Journal) provided that

(1) the necessary manuscript edits be done to properly present the findings in the context of current knowledge on eukaryotic m⁶A-YTHDF function

(2) additional analyses be added to fully exploit the obtained data.

The work adds interesting knowledge on the m⁶A-YTHDF system in planarians, which is important, because knowledge of how this system has evolved in different organisms is key to understand what its origins are and what it really does. However, the study does not reveal any genuinely new molecular principles not already reported in other systems.

We thank the reviewer for the positive assessment and for finding much scientific merit in our work. In the revision, we addressed both points: (1) we substantially edited the manuscript to place our findings in the context of current m⁶A-YTHDF literature, and (2) we added targeted analyses to more fully leverage the datasets. We also highlight the novel aspects of this study, including the role of YTHDF activity in planarian physiology and the utility of planarians as a comparative model for m⁶A installation.

REFERENCES

- Arribas-Hernandez, L., Simonini, S., Hansen, M.H., Paredes, E.B., Bressendorff, S., Dong, Y., Ostergaard, L., and Brodersen, P. (2020). Recurrent requirement for the m(6)A-ECT2/ECT3/ECT4 axis in the control of cell proliferation during plant organogenesis. *Development* 147, dev189134.
- Arribas-Hernández, L., Bressendorff, S., Hansen, M.H., Poulsen, C., Erdmann, S., and Brodersen, P. (2018). An m⁶A-YTH Module Controls Developmental Timing and Morphogenesis in Arabidopsis. *The Plant cell* 30, 952-967.
- Arribas-Hernández, L., Rennie, S., Köster, T., Porcelli, C., Lewinski, M., Staiger, D., Andersson, R., and Brodersen, P. (2021a). Principles of mRNA targeting via the Arabidopsis m⁶A-binding protein ECT2. *eLife* 10, e72375.
- Arribas-Hernández, L., Rennie, S., Schon, M., Porcelli, C., Enugutti, B., Andersson, R., Nodine, M.D., and Brodersen, P. (2021b). The YTHDF proteins ECT2 and ECT3 bind largely overlapping

target sets and influence target mRNA abundance, not alternative polyadenylation. *eLife* 10, e72377.

Dagan, Y., Yesharim, Y., Bonneau, A.R., Frankovits, T., Schwartz, S., Reddien, P.W., and Wurtzel, O. (2022). m6A is required for resolving progenitor identity during planarian stem cell differentiation. *The EMBO journal* 41, e109895.

Flores-Téllez, D., Tankmar, M.D., von Bülow, S., Chen, J., Lindorff-Larsen, K., Brodersen, P., and Arribas-Hernández, L. (2023). Insights into the conservation and diversification of the molecular functions of YTHDF proteins. *PLoS genetics* 19, e1010980.

Tankmar, M.D., Reichel, M., Arribas-Hernández, L., and Brodersen, P. (2023). A YTHDF-PABP interaction is required for m6A-mediated organogenesis in plants. *EMBO reports* 24, e57741.

Wang, G., Li, H., Ye, C., He, K., Liu, S., Jiang, B., Ge, R., Gao, B., Wei, J., Zhao, Y., Li, A., Zhang, D., Zhang, J., and He, C. (2024). Quantitative profiling of m6A at single base resolution across the life cycle of rice and Arabidopsis. *Nature communications* 15, 4881.

Dear Dr Wurtzel,

Thank you for submitting your revised manuscript (EMBOJ-2025-120652R) to The EMBO Journal, as well for your patience with our feedback. Your amended study was sent back to the three referees for their scientific reassessment, and we have received re-reports from two of them, which I enclose below. Please note that while referee #1 was not able at this time to reassess your work, we have evaluated your response to the issues raised by this expert editorially and found them to be addressed satisfactorily. As you will see, the other reviewers state that the work has been substantially enhanced by the revisions and they are now in favour of publication, pending minor amendments.

Thus, we are pleased to inform you that your manuscript has been accepted in principle for publication in The EMBO Journal.

Please carefully consider the remaining minor points raised by referee #3 by adjusting the discussion of the findings, and relativising claims where appropriate.

Also, we now need you to take care of a number of issues related to formatting and data presentation as detailed below, which should be addressed at re-submission.

Please contact me at any time if you have additional questions related to below points.

As you might remember from previous experience, every paper at the EMBO Journal now includes a 'Synopsis', displayed on the html and freely accessible to all readers. The synopsis includes a 'model' figure that summarizes the article findings. I would appreciate if you could provide this figure.

Thank you for giving us the chance to consider your manuscript for The EMBO Journal. I look forward to your final revision.

Again, please contact me at any time if you need any help or have further questions.

Best regards,

Daniel Klimmeck

>> Please add up to five keywords to your study.

>> Enter a 'Disclosure and Competing Interests Statement'.

>> Section order should be corrected as follows: title page with complete author information, abstract, keywords, introduction, results, discussion, methods, data availability section, acknowledgements, disclosure and competing interests statement, references, main figure legends, tables, expanded figure legends.

>> "Summary" should be renamed to "Abstract".

>> Manuscript format: both main and EV figures need to be removed from the manuscript and their legends listed below the References.

>> Figures in separate files: main and EV figures need to be uploaded as individual, high-resolution Figure files; "Supplementary Figures" should be renamed to Figure EV1-EV6 with the updated callouts instead of Figure S1-S6.

>> Please recheck the reference for the bioRxiv entry Cui et al. (2021) and update the citation if in the meantime published as regular article.

>> Appendix file with ToC: the first page title of the file with suppl. information should be renamed to "Appendix" and uploaded as a PDF.

>> Funding: please enter the following funding information in the list of funders in our online system: "European Research Council 913/21; Israel Science Foundation (ISF) 543165; missing in eJP: Zuckerman Faculty Scholar".

>> Dataset EV legends: dataset legends should be included as a separate tab/sheet in each Excel file.

>> Data availability section: remove referee token and make sure the datasets are made publicly accessible.

>> Add a separate 'Statistical Analysis' section to the Methods part, detailing the algorithms and statistical tests applied.

>> Consider additional changes and comments from our production team as indicated below:

****Data Check:**

>> Please note that the specific URL for PRJNA1226107 dataset is not provided in the data availability statement.

****Figure Legends (main + EV):**

1. Please note that the exact p values are not provided in the legends of figures 3D, 5D, 6B, 7D, EV6, EV8 F, H
2. Please indicate the statistical test used for data analysis in the legends of figures EV6 D, EV8 E, EV8 H.
3. Please note that the box plots need to be defined in terms of minima, maxima, centre, bounds of box and whiskers, and percentile in the legends of figures 1E, 3D, 5D, 7B, C, D, E; EV1 B, C, J; EV6 D, EV7, EV8 F, G; EV11 B
4. Please note that information related to n is missing in the legends of figures 1E, EV1 B, J; EV8 C
5. Please note that the error bars are not defined in the legend of figure EV6 E
6. Please note that the measure of center for the error bars needs to be defined in the legend of figure EV6 C

Referee #2:

This revised version nicely addresses all of my comments, and I congratulate the authors on an outstanding study.

Referee #3:

In their revised manuscript, the authors have added several new analyses and placed their findings much better within the

context of the relevant literature. It is a much better read in its current form, and I think it is close to being publishable.

For me, the big question is whether the degree of novelty is sufficient for The EMBO Journal. Normally, I would expect to see more generally applicable mechanistic insight for a paper at this level. But I am 100% with the authors on the point that analyses of the m6A pathway in animal taxa other than mammals (especially mammalian cell culture) are of the essence to uncover the fundamental functions of this important gene regulatory system.

It is a judgment call that ultimately lies with the editors, not me. The science is good, and the results and interpretations are well described and clearly of interest to the m6A field.

Remaining minor points of criticism/points to consider are

- The writing style is a little heavy and repetitive. I think it should be possible to clarify, perhaps shorten a bit in the process, and to improve the flow and readability to maximize impact. This is particularly true for the Results section.
- In the introduction, I would prefer if the authors would pronounce their opinions on the mammalian YTHDF redundancy/specialization debate more clearly, backed up by facts. They describe the two models as nearly equally well supported in the introduction, but later lean more towards support for largely redundant functions, if I understand well (I agree by the way, although the pattern of YTHDF1/2/3 conservation across vertebrates almost has to imply some small degree of specialization difficult to discern by knockout analysis). But this is clearly up to the authors how they want to frame their paper, I just do not think that their treatment of this very matter is completely consistent throughout the paper.
- I am not sure that this argument really holds: "These planarian ythdfs are lowly but broadly expressed (Fincher et al, 2018; Plass et al, 2018; 737 Dagan et al, 2022), and their overlapping expression patterns hint at functional redundancy." For example, Arabidopsis ECT1 has nearly exactly the same expression pattern as ECT2/3/4, yet clearly not the same molecular functions.

The authors addressed the remaining editorial issues.

Dear Dr Wurtzel,

Thank you for submitting the revised version of your manuscript. I have now evaluated your amended manuscript and concluded that the remaining minor concerns have been sufficiently addressed.

I am thus pleased to inform you that your manuscript has been accepted for publication in the EMBO Journal.

Best regards,

Daniel Klimmeck

Daniel Klimmeck, PhD
Senior Editor
The EMBO Journal
EMBO
Postfach 1022-40
Meyerhofstrasse 1
D-69117 Heidelberg
contact@embojournal.org

Please note that it is The EMBO Journal policy for the transcript of the editorial process (containing referee reports and your response letters) to be published as an online supplement to each paper. If you should prefer removal of any referee-only figures included in the point-by-point response(s), e.g. because they may still be used for future publication or because they have been reproduced from published work by others, please do let us know immediately via response email.

More information is available here: https://www.embopress.org/transparent-process#Review_Process